# Transition to Linearity of General Neural Networks with Directed Acyclic Graph Architecture

**Libin Zhu**[*]      **Chaoyue Liu**[†]      **Mikhail Belkin**[‡]

## Abstract

In this paper we show that feedforward neural networks corresponding to arbitrary directed acyclic graphs undergo transition to linearity as their "width" approaches infinity. The width of these general networks is characterized by the minimum in-degree of their neurons, except for the input and first layers. Our results identify the mathematical structure underlying transition to linearity and generalize a number of recent works aimed at characterizing transition to linearity or constancy of the Neural Tangent Kernel for standard architectures.

## 1 Introduction

A remarkable property of wide neural networks, first discovered in [10] in terms of the constancy of the Neural Tangent Kernel along the optimization path, is that they transition to linearity (using the terminology from [14]), i.e., are approximately linear in a ball of a fixed radius. There has been an extensive study of this phenomenon for different types of standard neural networks architectures including fully-connected neural networks (FCNs), convolutional neural networks (CNNs), ResNets [12, 4, 3, 7]. Yet the scope of the transition to linearity and the underlying mathematical structure has not been made completely clear.

In this paper, we show that the property of transition to linearity holds for a much broader class of neural networks – *feedforward neural networks*. The architecture of a feedforward neural network can generically be described by a DAG [25, 24, 16]: the vertices and the edges correspond to the neurons and the trainable weight parameters of a neural network, respectively. This DAG structure includes standard network architectures e.g., FCNs, CNNs, ResNets, as well as DenseNets [9], whose property of transition to linearity has not been studied in literature. This generalization shows that the transition to linearity, or the constant Neural Tangent Kernel, does not depend on the specific designs of the networks, and is a more fundamental and universal property.

We define the width of a feedforward neural network as the minimum in-degree of all neurons except for the input and first layers, which is a natural generalization of the the minimum number of neurons in hidden layers which is how the width is defined for standard architectures. For a feedforward neural network, we show it transitions to linearity if its width goes to infinity as long as the in-degrees of individual neurons are bounded by a polynomial of the network width. Specifically, we control the deviation of the network function from its linear approximation by the spectral norm of the Hessian of the network function, which, as we show vanishes in a ball of fixed radius, in the infinite width limit. Interestingly, we observe that not only the output neurons, but any pre-activated neuron in the hidden layers of a feedforward neural network can be regarded as a function with respect to its parameters, which will also transition to linearity as the width goes to infinity.

---

[*]Department of Computer Science & Halicioğlu Data Science Institute, University of California, San Diego. E-mail: `l5zhu@ucsd.edu`

[†]Halicioğlu Data Science Institute, University of California, San Diego. E-mail: `chl212@ucsd.edu`

[‡]Halicioğlu Data Science Institute, University of California, San Diego. E-mail: `mbelkin@ucsd.edu`

36th Conference on Neural Information Processing Systems (NeurIPS 2022).

The key technical difficulty is that all existing analyses for transition to linearity or constant NTK do not apply to this general DAG setting. Specifically, those analyses assume in-degrees of neurons are either the same or proportional to each other up to a constant ratio [5, 12, 3, 26, 14, 2]. However, the general DAG setting allows different scales of neuron in-degrees, for example, the largest in-degree can be polynomially large in the smallest in-degree. In such scenarios, the $(2, 2, 1)$-norm in [14] and the norm of parameter change in [5, 12] scales with the maximum of in-degrees which causes a trivial bound on the NTK change. Instead, we introduce a different set of tools based on the tail bound for the norm of matrix Gaussian series [22]. Specifically, we show that the Hessian of the network function takes the form of matrix Gaussian series, whose matrix variance relies on the Hessian of connected neurons. Therefore, we reconcile the in-degree difference by building a recursive relation between the Hessian of neurons, which exactly cancels out the in-degree with the scaling factor.

Transition to linearity helps understand the training dynamics of wide neural networks and plays an important role in developing the optimization theory for them, as has been shown for certain particular wide neural networks [6, 5, 4, 12, 27, 26]. While transition to linearity is not a necessary condition for successful optimization, it provides a powerful tool for analyzing optimization for many different architectures. Specifically, transition to linearity in a ball of sufficient radius combined with a lower bound on the norm of the gradient at its center is sufficient to demonstrate the $\text{PL}^*$ condition [13] (a version of the Polyak-Łojasiewicz condition [19, 15]) which ensures convergence of optimization. We discuss this connection and provide one such lower bound in Section 4.

**Summary of contributions.** We show the phenomenon of transition to linearity in general feedforward neural networks corresponding to a DAG with large in-degree. Specifically, under the assumption that the maximum in-degree of its neurons is bounded by a polynomial of the width $m$ (the minimum in-degree), we prove that the spectral norm of the Hessian of a feedforward neural network is bounded by $\tilde{O}(1/\sqrt{m})$ in an $O(1)$ ball. Our results generalize the existing literature on the linearity of wide feedforward neural networks. We discuss connections to optimization. Under additional assumptions we show that the norm of the gradient of a feedforward neural network is bounded away from zero at initialization. Together with the Hessian bound this implies convergence of gradient descent for the loss function.

## 1.1 Notations

We use bold lowercase letters, e.g., $\mathbf{w}$, to denote vectors, capital letters, e.g., $A$, to denote matrices, and bold capital letters, e.g., $\mathbf{H}$, to denote higher order tensors or matrix tuples. For a matrix $A$, we use $A_{[i,:]}$ to denote its $i$-th row and $A_{[:,i]}$ to denote its $j$-th column.

We use $\nabla_{\mathbf{w}} f(\mathbf{w}_0)$ to denote the gradient of $f$ with respect to $\mathbf{w}$ at $\mathbf{w}_0$, and $H_f(\mathbf{w})$ to denote Hessian matrix (second derivative) of $f$ with respect to $\mathbf{w}$. For vectors, we use $\|\cdot\|$ to denote Euclidean norm. For matrices, we use $\|\cdot\|$ to denote spectral norm and $\|\cdot\|_F$ to denote Frobenius norm. We use $\|\cdot\|_\infty$ to denote function $L_\infty$ norm. For a set $\mathcal{S}$, we use $|\mathcal{S}|$ to denote the cardinality of the set. For $n > 0$, $[n]$ denotes the set $\{1, 2, ..., n\}$.

We use big-$O$ notation to hide constant factors, and use big-$\tilde{O}$ notation to additionally hide logarithmic factors. In this paper, the argument of $O/\tilde{O}(\cdot)$ is always with respect to the network width.

Given a vector $\mathbf{w}$ and a constant $R > 0$, we define a Euclidean ball $\mathsf{B}(\mathbf{w}, R)$ as:

$$\mathsf{B}(\mathbf{w}, R) := \{\mathbf{v} : \|\mathbf{v} - \mathbf{w}\| \le R\}. \tag{1}$$

## 2 Neural networks with acyclic graph architecture

In this section, we provide a definition and notation for general feedforward neural networks with an arbitrary DAG structure. This definition includes standard feedforward neural network architectures, such as FCNs, DenseNet and CNNs.

## 2.1 Defining feedforward neural networks

**Graph Structure.** Consider a directed acyclic graph (DAG) $\mathcal{G} = (\mathcal{V}, \mathcal{E})$, where $\mathcal{V}$ and $\mathcal{E}$ denote the sets of vertices and edges, respectively. See the left panel of Figure 1, for an illustrative example.

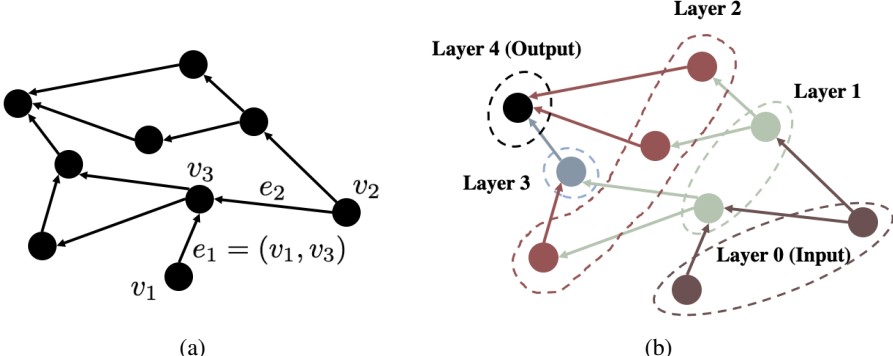

(a)                                                    (b)

Figure 1: **(a): An example of directed acyclic graph.** $v_1$, $v_2$ and $v_3$ are three vertices and $e_1$, $e_2$ are two edges of the graph. $v_3$ has two incoming edges $e_1$ and $e_2$ which connects to $v_1$ and $v_2$ respectively. **(b): Organizing the vertices into layers.** The vertices with $0$ in-degree are in $0$-th layer (or input layer), and last layer are called output layer in which the vertices have $0$ out-degree. Note that the layer index is determined by the longest path from the inputs $\mathcal{V}_{\text{input}}$, for example, the neuron in layer 3.

For a directed edge $e \in \mathcal{E}$, we may also use the notation $e = (v_1, v_2)$ to explicitly write out the start vertex $v_1$ and end vertex $v_2$.

For a vertex $v \in \mathcal{V}$, we denote its in-degree, $\text{in}(v)$, by the number of incoming edges (edges that end with it):

$$\text{in}(v) = |\mathcal{S}_{\text{in}}(v)|, \quad \text{with } \mathcal{S}_{\text{in}}(v) := \{u \in \mathcal{V} : (u, v) \in \mathcal{E}\}.$$

Similarly, for a vertex $v \in \mathcal{V}$, we denote its out-degree $\text{out}(v)$ by the number of outgoing edges (edges that start from it):

$$\text{out}(v) = |\mathcal{S}_{\text{out}}(v)|, \quad \text{with } \mathcal{S}_{\text{out}}(v) := \{u \in \mathcal{V} : (v, u) \in \mathcal{E}\}.$$

We call the set of vertices with zero in-degrees *input*: $\mathcal{V}_{\text{input}} = \{v \in \mathcal{V} : \text{in}(v) = 0\}$, and the set of vertices with zero out-degrees *output* $\mathcal{V}_{\text{output}} = \{v \in \mathcal{V} : \text{out}(v) = 0\}$.

**Definition 2.1.** For each vertex $v \in \mathcal{V} \backslash \mathcal{V}_{\text{input}}$, its distance $p(v)$, to the input $\mathcal{V}_{\text{input}}$, is defined to be the maximum length of all paths that start from a vertex within $\mathcal{V}_{\text{input}}$ and end with $v$.

It is easy to check that $p(v) = 0$ if $v \in \mathcal{V}_{\text{input}}$.

**Feedforward neural network.** Based on a given DAG architecture, we define the feedforward neural network. Each individual vertex corresponds to a neuron additionally equipped with a scalar function (also called activation function). Each edge is associated with a real-valued weight, a trainable parameter. Each neuron is defined as a function of the weight parameters and the adjacent neurons connected by its incoming edges. The feedforward neural network is considered as the output neurons, corresponding to the output $\mathcal{V}_{\text{output}}$, of all weight parameters and input neurons which correspond to the input $\mathcal{V}_{\text{input}}$. Formally, we define the feedforward neural network as follows.

**Definition 2.2** (Feedforward neural network). Consider a DAG $\mathcal{G} = (\mathcal{V}, \mathcal{E})$. For each vertex $v \in \mathcal{V} \backslash \mathcal{V}_{\text{input}}$, we associate it with an activation function $\sigma_v(\cdot) : \mathbb{R} \to \mathbb{R}$ and each of its incoming edges $e = (u, v) \in \mathcal{E}$ with a weight variable $w_e = w_{(u,v)}$. Then we define the following functions:

$$f_v = \sigma_v(\tilde{f}_v), \quad \tilde{f}_v = \frac{1}{\sqrt{\text{in}(v)}} \sum_{u \in \mathcal{S}_{\text{in}}(v)} w_{(u,v)} f_u. \tag{2}$$

When $v \in \mathcal{V}_{\text{input}}$, $f_v$ is prefixed as the input data, and we denote $f_{\text{input}} := \{f_v : v \in \mathcal{V}_{\text{input}}\}$. For $v \notin \mathcal{V}_{\text{input}}$, we call $f_v$ *neurons* and $\tilde{f}_v$ *pre-activations*. With necessary composition of functions, each $f_v$, and $\tilde{f}_v$, can be regarded as a function of all related weight variables and inputs $f_{\text{input}}$. The *feedforward neural network* is defined to be the function corresponding to the output $\mathcal{V}_{\text{output}}$:

$$f(\mathcal{W}; f_{\text{input}}) := f_{\text{output}} = \{f_v : v \in \mathcal{V}_{\text{output}}\}, \tag{3}$$

where $\mathcal{W} := \{w_e : e \in \mathcal{E}\}$ denotes the set of all the weight variables.

*Remark* 2.3. The validity of the definition is guaranteed by the fact that the DAG is acyclic. It makes sure that the dependence of each function $f_v$ on other neurons can pass all the way down to the input $f_{\text{input}}$, through Eq. (2).

*Remark* 2.4. For $v \in \mathcal{V}_{\text{input}} \bigcup \mathcal{V}_{\text{output}}$, we use the identity function $\mathbb{I}(\cdot)$ as the activation functions.

**Weight initialization and inputs.**  Each weight parameter $w_e \in \mathcal{W}$ is initialized i.i.d. following the standard normal distribution i.e., $\mathcal{N}(0, 1)$. The inputs are considered given, usually determined by datasets. Under this initialization, we introduce the scaling factor $1/\sqrt{\text{in}(v)}$ in Eq. (5) to control the value of neurons to be of order $O(1)$. Note that this initialization is an extension of the NTK initialization [10], which was defined for FCNs therein.

**Generality of DAG architecture.**  Including FCNs and DenseNets [9] as special examples, the class of feedforward neural networks allows much more choices of architectures, for example, neural networks with randomly dropped edges. Please see detailed discussions about these specific examples in Appendix A. We note that our definition of feedforward neural networks does not directly include networks with nont-trainable skip connections, e.g., ResNets, and networks with shared weights, e.g., CNNs. However, with a slight modification of the analysis, the property of transition to linearity still holds. See the detailed discussion in Appendix D and E.

## 2.2  Organizing feedforward networks into layers

The architecture of the feedforward neural network is determined by the DAG $\mathcal{G}$. The complex structures of DAGs often lead to complicated neural networks, which are hard to analyze.

For the ease of analysis, we organize the neurons of the feedforward neural network into *layers*, which are sets of neurons.

**Definition 2.5** (Layers).  Consider a feedforward neural network $f$ and its corresponding graph structure $\mathcal{G}$. A layer of the network is defined to be the set of neurons which have the same distance $p$ to the inputs. Specifically, the $\ell$-th layer, denoted by $f^{(\ell)}$, is

$$f^{(\ell)} = \{ f_v : p(v) = \ell, v \in \mathcal{V}, \ell \in \mathbb{N} \}. \tag{4}$$

It is easy to see that the layers are mutually exclusive, and the layer index $\ell$ is labeled from 0 to $\ell$, where $L + 1$ is the total number of layers in the network. As $p(v) = 0$ if and only if $v \in \mathcal{V}_{\text{input}}$, the 0-th layer $f^{(0)}$ is exactly the input layer $f_{\text{input}}$. The right panel of Figure 1 provides an illustrative example of the layer structures.

In general, the output neurons $f_{\text{output}}$ (defined in Eq. (3)) do not have to be in the same layer. For the convenience of presentation and analysis, we assume that all the output neurons are in the last layer, i.e., layer $\ell$, which is the case for most of commonly used neural networks, e.g., FCNs and CNNs. Indeed, our analysis applies to every output neuron (see Theorem 3.8), even if they are not in the same layer.

With the notion of network layers, we rewrite the neuron functions Eq. (2), as well as related notations, to reflect the layer information.

For $\ell$-layer, $\ell = 0, 1, \cdots, L$, we denote the total number of neurons as $d_\ell$, and rewrite the layer function $f^{(\ell)}$ into a form of vector-valued function

$$f^{(\ell)} = \left( f_1^{(\ell)}, f_2^{(\ell)}, ..., f_{d_\ell}^{(\ell)} \right)^T,$$

where we use $f_i^{(\ell)}$ with index $i = 1, 2, \cdots, d_\ell$ to denote each individual neuron. Correspondingly, we denote its vertex as $v_i^{(\ell)}$, and $\mathcal{S}_i^{(\ell)} := \mathcal{S}_{\text{in}}(v_i^{(\ell)})$. Hence, the in-degree $\text{in}(v_i^{(\ell)})$, denoted as $m_i^{(\ell)}$ here, is equivalent to the cardinality of the set $\mathcal{S}_i^{(\ell)}$.

*Remark* 2.6. Note that $m_i^{(\ell)}$, with the superscript $\ell$, denotes an in-degree, i.e., the number of neurons that serve as direct inputs to the current neuron in $\ell$-th layer. In the context of FCNs, $m_i^{(\ell)}$ is equivalent to the size of its previous layer, i.e., $(\ell - 1)$-th layer, and is often denoted as $m^{(\ell-1)}$ in literature.

To write the summation in Eq. (2) as a matrix multiplication, we further introduce the following two vectors: (a), $f_{\mathcal{S}_i^{(\ell)}}$ represents the vector that consists of neuron components $f_v$ with $v \in \mathcal{S}_i^{(\ell)}$; (b), $\mathbf{w}_i^{(\ell)}$ represents the vector that consists of weight parameters $w_{(u,v_i^{(\ell)})}$ with $u \in \mathcal{S}_i^{(\ell)}$. Note that both vectors $f_{\mathcal{S}_i^{(\ell)}}$ and $\mathbf{w}_i^{(\ell)}$ have the same dimension $m_i^{(\ell)}$.

With the above notation, the neuron functions Eq. (2) can be equivalently rewritten as:

$$f_i^{(\ell)} = \sigma_i^{(\ell)}(\tilde{f}_i^{(\ell)}), \ \ \tilde{f}_i^{(\ell)} = \frac{1}{\sqrt{m_i^{(\ell)}}} \left(\mathbf{w}_i^{(\ell)}\right)^T f_{\mathcal{S}_i^{(\ell)}}. \tag{5}$$

For any $\ell \in [L]$, we denote the weight parameters corresponding to all incoming edges toward neurons at layer $\ell$ by

$$\mathbf{w}^{(\ell)} := \left((\mathbf{w}_1^{(\ell)})^T, ..., (\mathbf{w}_{d_\ell}^{(\ell)})^T\right)^T \ \ \ell \in [L]. \tag{6}$$

Through the way we define the feedforward neural network, the output of the neural network is a function of all the weight parameters and the input data, hence we denote it by

$$f(\mathbf{w}; \boldsymbol{x}) := f^{(\ell)} = \left(f_1^{(\ell)}, ..., f_{d_\ell}^{(\ell)}\right)^T, \tag{7}$$

where $\mathbf{w}$ is the collection of all the weight parameters, i.e., $\mathbf{w} := \left((\mathbf{w}^{(1)})^T, ..., (\mathbf{w}^{(\ell)})^T\right)^T \in \mathbb{R}^{\sum_\ell \sum_i m_i^{(\ell)}}$.

With all the notations, for a feedforward neural network, we formally define the width of it:

**Definition 2.7** (Network width). The width $m$ of a feedforward neural network is the minimum in-degree of all the neurons except those in the input and first layers:

$$m := \inf_{\ell \in \{2,...,L\}, i \in [d_\ell]} m_i^{(\ell)}. \tag{8}$$

*Remark* 2.8. Note that, the network width $m$ is determined by the in-degrees of neurons except for the input and first layers, and not necessarily relates the number of neurons in hidden layers. But for certain architectures e.g., FCNs, these two coincide that the minimum in-degree after the first layer is the same as the minimum hidden layer size.

We say a feedforward neural network is *wide* if its width $m$ is large enough. In this paper, we consider wide feedforward neural networks with a fixed number of layers.

## 3 Transition to linearity of feedforward neural networks

In this section, we show that the feedforward neural networks exhibit the phenomenon of transition to linearity, which was previously observed in specific types of neural networks.

Specifically, we prove that a feedforward neural network $f(\mathbf{w}; \boldsymbol{x})$, when considered as a function of its weight parameters $\mathbf{w}$, is arbitrarily close to a *linear* function in the ball $\mathsf{B}(\mathbf{w}_0, R)$ given constant $R > 0$, where $\mathbf{w}_0$ is randomly initialized, as long as the width of the network is sufficiently large.

First, we make the following assumptions on the input $\boldsymbol{x}$ and the activation functions:

**Assumption 3.1.** The input is uniformly upper bounded, i.e., $\|\boldsymbol{x}\|_\infty \le C_{\boldsymbol{x}}$ for some constant $C_{\boldsymbol{x}} > 0$.

**Assumption 3.2.** All the activation functions $\sigma(\cdot)$ are twice differentiable, and there exist constants $\gamma_0, \gamma_1, \gamma_2 > 0$ such that, for all activation functions, $|\sigma(0)| \le \gamma_0$ and the following Lipschitz continuity and smoothness conditions are satisfied

$$|\sigma'(z_1) - \sigma'(z_2)| \le \gamma_1 |z_1 - z_2|,$$
$$|\sigma''(z_1) - \sigma''(z_2)| \le \gamma_2 |z_1 - z_2|, \ \ \forall z_1, z_2 \in \mathbb{R}.$$

We note that the above two assumptions are very common in literature. Although ReLU does not satisfy Assumption 3.2 due to non-differentiability at point 0, we believe our main claims still hold as ReLU can be approximated arbitrarily closely by some differentiable function which satisfies our assumption.

*Remark* 3.3. By assuming all the activation functions are twice differentiable, it is not hard to see that the feedforward neural network i.e., Eq. (7) is also twice differentiable.

**Taylor expansion.** To study the linearity of a general feedforward neural network, we consider its Taylor expansion with second order Lagrange remainder term. Given a point $\mathbf{w}_0$, we can write the network function $f(\mathbf{w})$ (omitting the input argument for simplicity) as

$$f(\mathbf{w}) = \underbrace{f(\mathbf{w}_0) + (\mathbf{w} - \mathbf{w}_0)^T \nabla_{\mathbf{w}} f(\mathbf{w}_0)}_{f_{\mathrm{lin}}(\mathbf{w})} + \underbrace{\frac{1}{2}(\mathbf{w} - \mathbf{w}_0)^T H_f(\xi)(\mathbf{w} - \mathbf{w}_0)}_{\mathcal{R}(\mathbf{w})}, \tag{9}$$

where $\xi$ is a point on the line segment between $\mathbf{w}_0$ and $\mathbf{w}$. Above, $f_{\mathrm{lin}}(\mathbf{w})$ is a linear function and $\mathcal{R}(\mathbf{w})$ is the Lagrange remainder term.

In the rest of the section, we will show that in a ball $\mathsf{B}(\mathbf{w}_0, R)$ of any constant radius $R > 0$,

$$|\mathcal{R}(\mathbf{w})| = \tilde{O}\left(1/\sqrt{m}\right) \tag{10}$$

where $m$ is the network width (see Definition 2.7). Hence, $f(\mathbf{w})$ can be arbitrarily close to its linear approximation $f_{\mathrm{lin}}(\mathbf{w})$ with sufficiently large $m$. Note that in Eq. (9), we consider a single output of the network function. The same analysis can be applied to multiple outputs (see Corollary C.1).

*Remark* 3.4. For a general function, the remainder term $\mathcal{R}(\mathbf{w})$ is not expected to vanish at a finite distance from $\mathbf{w}_0$. Hence, the transition to linearity in the ball $\mathsf{B}(\mathbf{w}_0, R)$ is a non-trivial property. On the other hand, the radius $R$ can be set to be large enough to contain the whole optimization path of GD/SGD for various types of wide neural networks (see [13, 27], also indicated in [6, 5, 26, 12]). In Section 4, we will see that such a ball is also large enough to cover the whole optimization path of GD/SGD for the general feedforward neural networks. Hence, to study the optimization dynamics of wide feedforward neural networks, this ball is large enough.

To prove Eq. (10), we make an assumption on the width $m$:

**Assumption 3.5.** The maximum in-degree of any neuron is at most polynomial in the network width $m$:

$$\sup_{\ell \in \{2, \dots, L\}, i \in [d_\ell]} m_i^{(\ell)} = O(m^c),$$

where $c > 0$ is a constant.

This assumption puts a constraint on the neurons with large in-degrees such that the in-degrees cannot be super-polynomially large compared to $m$. A natural question is whether this constraint is necessary, for example, do our main results still hold in cases some in-degrees are exponentially large in $m$? While we believe the answer is positive, we need this assumption to apply the proof techniques. Specifically, we apply the tail bound for the norm of matrix Gaussian series [22], where there is a dimension factor equivalent to the number of weight parameters. Thus an exponentially large dimension factor would result in useless bounds. It is still an open question whether the dimension factor in the bound can be removed or moderated (see the discussion after Theorem 4.1.1 in [22]).

With these assumptions, we are ready to present our main result:

**Theorem 3.6** (Scaling of the Hessian norm). *Suppose Assumption 3.1, 3.2 and 3.5 hold. Given a fixed $R > 0$, for all $\mathbf{w} \in \mathsf{B}(\mathbf{w}_0, R)$, with probability at least $1 - \exp(-\Omega(\log^2 m))$ over the random initialization $\mathbf{w}_0$, each output neuron $f_k$ of a feedforward neural network satisfies*

$$\|H_{f_k}(\mathbf{w})\| = O\left((\log m + R)^{L^2}/\sqrt{m}\right) = \tilde{O}\left(R^{L^2}/\sqrt{m}\right), \quad k \in [d_\ell]. \tag{11}$$

This theorem states that the Hessian matrix, as the second derivative with respect to weight parameters $\mathbf{w}$, of any output neuron can be arbitrarily small, if the network width is sufficient large.

Note that Eq. (11) holds for all $\mathbf{w} \in \mathsf{B}(\mathbf{w}_0, R)$ with high probability over the random initialization $\mathbf{w}_0$. The basic idea is that, the spectral norm of Hessian can be bounded at the center of the ball, i.e., $\mathbf{w}_0$, though probability bounds due to the randomness of $\mathbf{w}_0$. For all other points $\mathbf{w} \in \mathsf{B}(\mathbf{w}_0, R)$, the distance $\|\mathbf{w} - \mathbf{w}_0\|$, being no greater than $R$, controls $\|H(\mathbf{w}) - H(\mathbf{w}_0)\|$ such that it is no larger than the order of $\|H(\mathbf{w}_0)\|$, hence $\|H(\mathbf{w})\|$ keeps the same order. See the proof details in Subsection 3.1.

Using the Taylor expansion Eq. (9), we can bound the Lagrange remainder and have transition to linearity of the network:

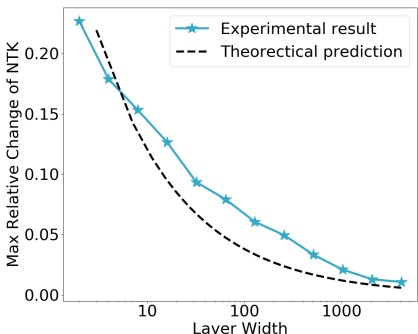

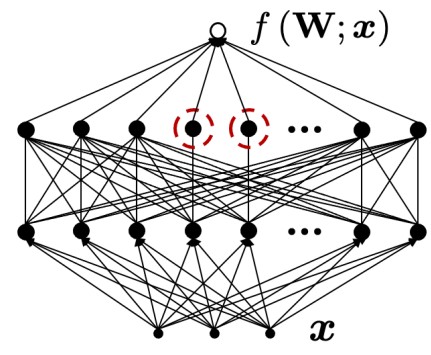

Figure 2: **Transition to linearity of DAG network.** The experimental result approximates well the theoretical prediction of relative change of tangent kernel from initialization to convergence, as a function of the network width. Each point on the solid curve is the average of independent 5 runs.

Figure 3: **An example of DAG network with bottleneck neurons.** The DAG network $f(\mathbf{W}; \boldsymbol{x})$ has two bottleneck neurons (in red dashed circles) with in-degree 1, while the rest of neurons except for the input and the first layer have large in-degree. In this case, $f(\mathbf{W}; \boldsymbol{x})$ will still transition to linearity with respect to $\mathbf{W}$ as the number of neurons goes to infinity.

**Corollary 3.7** (Transition to linearity). *Suppose Assumption 3.1, 3.2 and 3.5 hold. Given a fixed $R > 0$, for all $\mathbf{w} \in \mathsf{B}(\mathbf{w}_0, R)$, with probability at least $1 - \exp(-\Omega(\log^2 m))$ over the random initialization $\mathbf{w}_0$, each $f_k$ will be closely approximated by a linear model:*

$$|f_k(\mathbf{w}) - (f_k)_{\mathrm{lin}}(\mathbf{w})| \leq \frac{1}{2} \sup_{\mathbf{w} \in \mathsf{B}(\mathbf{w}_0, R)} \|H_{f_k}(\mathbf{w})\| R^2 = \tilde{O}\left(R^{L^2+2}/\sqrt{m}\right).$$

For feedforward neural networks with multiple output neurons, the property of transition to linearity holds with high probability, if the number of output neurons is bounded, i.e., $d_\ell = O(1)$. See the result in Appendix C.

Furthermore, as defined in Definition 2.2, each pre-activation, as a function of all related weight parameters and inputs, can be viewed as a feedforward neural network. Therefore, we can apply the same techniques used for Theorem 3.6 to show that each pre-activation can transition to linearity:

**Theorem 3.8.** *Suppose Assumption 3.1, 3.2 and 3.5 hold. Given a fixed radius $R > 0$, for all $\mathbf{w} \in \mathsf{B}(\mathbf{w}_0, R)$, with probability at least $1 - \exp(-\Omega(\log^2 m))$ over the random initliazation of $\mathbf{w}_0$, any pre-activation in a feedforward neural network i.e., $\tilde{f}_k^{(\ell)}(\mathbf{w})$ satisfies*

$$\left\|H_{\tilde{f}_k^{(\ell)}}(\mathbf{w})\right\| = O\left((\log m + R)^{\ell^2}/\sqrt{m}\right) = \tilde{O}\left(R^{\ell^2}/\sqrt{m}\right), \quad \ell \in [L], \; k \in [d_\ell]. \tag{12}$$

*Remark* 3.9. Note that pre-activations in the input layer i.e., the input data and in the first layer are constant and linear functions respectively, hence the spectral norm of their Hessian is zero.

**Experimental verification.** To verify our theoretical result on the scaling of the Hessian norm, i.e., Theorem 3.6, we train a DAG network built from a 3-layer DenseNet with each weight removed i.i.d. with probability $1/2$, on 10 data points of CIFAR-2 (2-class subset of CIFAR-10 [11]) using GD. We compute the maximum relative change of the tangent kernel (definition is deferred to Eq. (17)) during training, i.e., $\max_t \|K_t - K_0\|/\|K_0\|$ to simulate the scaling of the spectral norm of the Hessian. We observe the convergence of loss for all widths $\{2, 2^2, ..., 2^{12}\}$, and the scaling of the Hessian follows close to the theoretical prediction of $\Theta(1/\sqrt{m})$. See Figure 2.

**Non-linear activation on output neurons breaks transition to linearity.** In the above discussions, the transition to linearity of networks are under the assumption of identity activation function on every output neuron. In fact, the activation function on output neurons is critical to the linearity of neural networks. Simply, composing a non-linear function with a linear function will break the linearity. Consistently, as shown in [13] for FCNs, with non-linear activation function on the output, transition to linearity does not hold any more.

**"Bottleneck neurons" do not necessarily break transition to linearity.** We have seen that if *all* neurons have sufficiently large in-degree, the network will transition to linearity. Does transition to linearity still hold, if neurons with small in-degree exist? We call neurons with small in-degree *bottleneck neurons*, as their in-degrees are smaller than rest of neurons hence forming "bottlenecks". As we show in Appendix E, existence of these bottleneck neurons does not break the transition to linearity, as long as the number of such neurons is significantly smaller than that of non-bottleneck neurons. Figure 3 shows an example with two bottleneck neurons, whose in-degree is 1. This network still transitions to linearity as the number of neurons goes to infinity.

### 3.1 Proof sketch of Theorem 3.6

By Lemma K.1, the spectral norm of $H_{f_k}$ can be bounded by the summation of the spectral norm of all the Hessian blocks, i.e., $\|H_{f_k}\| \leq \sum_{\ell_1, \ell_2} \|H_{f_k}^{(\ell_1, \ell_2)}\|$, where $H_{f_k}^{(\ell_1, \ell_2)} := \frac{\partial^2 f_k}{\partial \mathbf{w}^{(\ell_1)} \partial \mathbf{w}^{(\ell_2)}}$. Therefore, it suffices to bound the spectral norm of each block. Without lose of generality, we consider the block with $1 \leq \ell_1 \leq \ell_2 \leq L$.

By the chain rule of derivatives, we can write the Hessian block into:

$$\frac{\partial^2 f_k}{\partial \mathbf{w}^{(\ell_1)} \partial \mathbf{w}^{(\ell_2)}} = \sum_{\ell'=\ell_2}^{L} \sum_{i=1}^{d_{\ell'}} \frac{\partial^2 f_i^{(\ell')}}{\partial \mathbf{w}^{(\ell_1)} \partial \mathbf{w}^{(\ell_2)}} \frac{\partial f_k}{\partial f_i^{(\ell')}} := \sum_{\ell'=\ell_2}^{L} G_k^{L,\ell'}. \tag{13}$$

For each $G_k^{L,\ell'}$, since $f_i^{(\ell')} = \sigma\left(\tilde{f}_i^{(\ell')}\right)$, again by the chain rule of derivatives, we have

$$G_k^{L,\ell'} = \sum_{i=1}^{d_{\ell'}} \frac{\partial^2 \tilde{f}_i^{(\ell')}}{\partial \mathbf{w}^{(\ell_1)} \partial \mathbf{w}^{(\ell_2)}} \frac{\partial f_k}{\partial \tilde{f}_i^{(\ell')}} + \frac{1}{\sqrt{m_k^{(L)}}} \sum_{i:f_i^{(\ell')} \in \mathcal{F}_{\mathcal{S}_k^{(L)}}} \left(\mathbf{w}_k^{(L)}\right)_{\mathsf{id}_{\ell',i}^{L,k}} \sigma''\left(\tilde{f}_i^{(\ell')}\right) \frac{\partial \tilde{f}_i^{(\ell')}}{\partial \mathbf{w}^{(\ell_1)}} \left(\frac{\partial \tilde{f}_i^{(\ell')}}{\partial \mathbf{w}^{(\ell_2)}}\right)^T$$

$$= \frac{1}{\sqrt{m_k^{(L)}}} \sum_{r=\ell'}^{L-1} \sum_{i:f_i^{(r)} \in \mathcal{F}_{\mathcal{S}_k^{(L)}}} \left(\mathbf{w}_k^{(L)}\right)_{\mathsf{id}_{r,i}^{L,k}} \sigma'\left(\tilde{f}_s^{(r)}\right) G_i^{r,\ell'}$$

$$+ \frac{1}{\sqrt{m_k^{(L)}}} \sum_{i:f_i^{(\ell')} \in \mathcal{F}_{\mathcal{S}_k^{(L)}}} \left(\mathbf{w}_k^{(L)}\right)_{\mathsf{id}_{\ell',i}^{L,k}} \sigma''\left(\tilde{f}_i^{(\ell')}\right) \frac{\partial \tilde{f}_i^{(\ell')}}{\partial \mathbf{w}^{(\ell_1)}} \left(\frac{\partial \tilde{f}_i^{(\ell')}}{\partial \mathbf{w}^{(\ell_2)}}\right)^T, \tag{14}$$

where $\mathcal{F}_{\mathcal{S}_k^{(L)}} := \{f : f \in f_{\mathcal{S}_k^{(L)}}\}$ and $\mathsf{id}_{\ell',i}^{L,k} := \{p : \left(f_{\mathcal{S}_k^{(L)}}\right)_p = f_i^{(\ell')}\}$.

The first quantity on the RHS of the above equation, $\sum \left(\mathbf{w}_k^{(L)}\right)_{\mathsf{id}_{\ell',i}^{L,k}} \sigma'\left(\tilde{f}_i^{(r)}\right) G_i^{r,\ell'}$, is a matrix Gaussian series with respect to random variables $\mathbf{w}_k^{(L)}$, conditioned on fixed $\sigma'\left(\tilde{f}_i^{(r)}\right) G_i^{r,\ell'}$ for all $i$ such that $f_i^{(r)} \in \mathcal{F}_{\mathcal{S}_k^{(\ell)}}$. We apply the tail bound for matrix Gaussian series, Theorem 4.1.1 from [22], to bound this quantity. To that end, we need to bound its matrix variance, which suffices to bound the spectral norm of $\sum_i G_i^{r,\ell'}$ since $\sigma'(\cdot)$ is assumed to be uniformly bounded by Assumption 3.2. There is a recursive relation that the norm bound of $G_k^{L,\ell'}$ depends on the norm bound of $G_i^{r,\ell'}$ which appears in the matrix variance. Therefore, we can recursively apply the argument to bound each $G$.

Similarly, the second quantity on the RHS of the above equation is also a matrix Gaussian series with respect to $\mathbf{w}_k^{(L)}$, conditioned on fixed $\sigma''\left(\tilde{f}_i^{(\ell')}\right) \frac{\partial \tilde{f}_i^{(\ell')}}{\partial \mathbf{w}^{(\ell_1)}} \left(\frac{\partial \tilde{f}_i^{(\ell')}}{\partial \mathbf{w}^{(\ell_2)}}\right)^T$ for all $i$ such that $f_i^{(\ell')} \in \mathcal{F}_{\mathcal{S}_k^{(L)}}$. As $\sigma''(\cdot)$ is assumed to be uniformly bounded by Assumption 3.2, we use Lemma B.1 to bound its matrix variance, hence the matrix Gaussian series can be bounded.

Note that such tail bound does not scale with the largest in-degree of the networks, since the in-degree of $f_k$, i.e., $m_k^{(L)}$, will be cancelled out with the scaling factor $1/\sqrt{m_k^{(L)}}$ in the bound of matrix variance.

See the complete proof in Appendix B.

## 4 Relation to optimization

While transition to linearity is a significant and surprising property of wide networks in its own right, it also plays an important role in building the optimization theory of wide feedforward neural networks. Specifically, transition to linearity provides a path toward showing that the corresponding loss function satisfies the PL* condition in a ball of a certain radius, which is sufficient for exponential convergence of optimization to a global minimum by gradient descent or SGD [13].

Consider a supervised learning task. Given training inputs and labels $\{(\boldsymbol{x}_i, y_i)\}_{i=1}^n$, we use GD/SGD to minimize the square loss:

$$\mathcal{L}(\mathbf{w}) = \frac{1}{2} \sum_{i=1}^n (f(\mathbf{w}; \boldsymbol{x}_i) - y_i)^2, \tag{15}$$

where $f(\mathbf{w}; \cdot)$ is a feedforward neural network.

The loss $\mathcal{L}(\mathbf{w})$ is said to satisfy $\mu$-PL* condition, a variant of the well-known Polyak-Łojasiewicz condition [19, 15], at point $\mathbf{w}$, if

$$\|\nabla_{\mathbf{w}} \mathcal{L}(\mathbf{w})\|^2 \geq 2\mu \mathcal{L}(\mathbf{w}), \quad \mu > 0.$$

Satisfaction of this $\mu$-PL* condition in a ball $B(\mathbf{w}_0, R)$ with $R = O(1/\mu)$ around the starting point $\mathbf{w}_0$ of GD/SGD guarantees a fast converge of the algorithm to a global minimum in this ball [13].

In the following, we use the transition to linearity of wide feedforward networks to establish the PL* condition for $\mathcal{L}(\mathbf{w})$. Taking derivative on Eq. (15), we get

$$\|\nabla_{\mathbf{w}} \mathcal{L}(\mathbf{w})\|^2 \geq 2\lambda_{\min}(K(\mathbf{w}))\mathcal{L}(\mathbf{w}), \tag{16}$$

where matrix $K(\mathbf{w})$, with elements

$$K_{i,j}(\mathbf{w}) = \nabla_{\mathbf{w}} f(\mathbf{w}; \boldsymbol{x}_i)^T \nabla_{\mathbf{w}} f(\mathbf{w}; \boldsymbol{x}_j) \text{ for } i, j \in [n], \tag{17}$$

is called Neural Tangent Kernel (NTK) [10], and $\lambda_{\min}(\cdot)$ denotes the smallest eigenvalue of a matrix. Note that, by definition, the NTK matrix is always positive semi-definite, i.e., $\lambda_{\min}(K(\mathbf{w})) \geq 0$.

Directly by the definition of PL* condition, at a given point $\mathbf{w}$, if $\lambda_{\min}(K(\mathbf{w}))$ is strictly positive, then the loss function $\mathcal{L}(\mathbf{w})$ satisfies PL* condition.

To establish convergence of GD/SGD, it is sufficient to verify that PL* condition is satisfied in a ball $B(\mathbf{w}_0, R)$ with $R = O(1/\mu)$. Assuming that $\lambda_{\min}(K(\mathbf{w}_0))$ is bounded away from zero, transition to linearity extends the satisfaction of the PL* condition from one point $\mathbf{w}_0$ to all points in $B(\mathbf{w}_0, R)$.

**PL* condition at $\mathbf{w}_0$.** For certain neural networks, e.g., FCNs, CNNs and ResNets, strict positiveness of $\lambda_{\min}(K(\mathbf{w}_0))$ can be shown, see for example, [6, 5]. We expect same holds more generally, in the case of general feedforward neural networks. Here we show that $\lambda_{\min}(K(\mathbf{w}_0))$ can be bounded from 0 for one data point under certain additional assumptions. Since there is only one data point, $\lambda_{\min}(K(\mathbf{w}_0)) = K(\mathbf{w}_0) = \|\nabla_{\mathbf{w}} f(\mathbf{w}_0)\|^2$. We also assume the following on activation functions and the input.

**Assumption 4.1.** The input $\mathbf{x}$ satisfies $\mathbf{x} \sim \mathcal{N}(0, I_{d_0})$.

**Assumption 4.2.** The activation function is homogeneous, i.e. $\sigma_i^{(\ell)}(az) = a^r \sigma_i^{(\ell)}(z), r > 0$ for any constant $a$. And $\inf_{\ell \in [L-1], i \in [d_\ell]} \mathbb{E}_{z \sim \mathcal{N}(0,1)} \left[ \sigma_i^{(\ell)}(z)^2 \right] = C_\sigma > 0$.

*Remark* 4.3. Here for simplicity we assume the activation functions are homogeneous with the same $r$. It is not hard to extend the result to the case that each activation function has different $r$.

**Proposition 4.4.** *With Assumption 4.1 and 4.2, we have for any $k \in [d_\ell]$,*

$$\mathbb{E}_{\mathbf{x}, \mathbf{w}_0}[\|\nabla_{\mathbf{w}} f_k(\mathbf{w}_0)\|] \geq \sqrt{\min \left( 1, \min_{1 \leq j \leq L} C_\sigma^{\sum_{l'=0}^{j-1} r^{l'}} \right)} = \Omega(1). \tag{18}$$

The proof can be found in Appendix G.

The above proposition establishes a positive lower bound on $\|\nabla_{\mathbf{w}} f(\mathbf{w}_0)\|$, hence also on $\lambda_{\min}(K(\mathbf{w}_0))$. Using Eq. (16), we get that the loss function $\mathcal{L}(\mathbf{w})$ satisfies PL* at $\mathbf{w}_0$.

**Extending PL* condition to $\mathsf{B}(\mathbf{w}_0, R)$.** Now we use transition to linearity to extend the satisfaction of PL* condition to the ball $\mathsf{B}(\mathbf{w}_0, R)$. In Theorem 3.6, we see that, a feedforward neural network $f(\mathbf{w})$ transitions to linearity, i.e., $\|H_f(\mathbf{w})\| = \tilde{O}(1/\sqrt{m})$ in this ball. An immediate consequence is that, for any $\mathbf{w} \in \mathsf{B}(\mathbf{w}_0, R)$,

$$|\lambda_{\min}(K(\mathbf{w})) - \lambda_{\min}(K(\mathbf{w}_0))| \leq O \left( \sup_{\mathbf{w} \in \mathsf{B}(\mathbf{w}_0, R)} \|H_f(\mathbf{w})\| \right).$$

Since $\lambda_{\min}(K(\mathbf{w}_0))$ is bound from 0 and $\|H_f(\mathbf{w})\|$ can be arbitrarily small as long as $m$ is large enough, we have $\lambda_{\min}(K(\mathbf{w}))$ is lower bounded from 0 in the whole ball. Specifically, there is a $\mu > 0$ such that

$$\inf_{\mathbf{w} \in \mathsf{B}(\mathbf{w}_0, R)} \lambda_{\min}(K(\mathbf{w})) \geq \mu.$$

Moreover, the radius $R$ can be set to be $O(1/\mu)$, while keeping the above inequality hold. Then, applying the theory in [13], existence of global minima of $\mathcal{L}(\mathbf{w})$ and convergence of GD/SGD can be established.

For the case of multiple data points, extra techniques are needed to lower bound the minimum eigenvalue of the tangent kernel. Since we focus more on the transition to linearity of feedforward neural networks in this paper, we leave it as a future work.

**Non-linear activation function on outputs and transition to linearity.** In this paper, we mainly discussed feedforward neural networks with linear activation function on output neurons. In most of the literature also considers this setting [10, 17, 18, 6, 5, 27, 26]. In fact, as pointed out in [13] for FCNs, while this linearity of activation function on the outputs is necessary for transition to linearity, it is not required for successful optimization. Specifically, simply adding a nonlinear activation function on the output layer causes the Hessian norm to be $O(1)$, independently of the network width. Thus transition to linearity does not occur. However, the corresponding square loss can still satisfy the PL* condition and the existence of global minimum and efficient optimization can still be established.

## 5 Discussion and future directions

In this work, we showed that transition to linearity arises in general feedforward neural networks with arbitrary DAG architectures, extending previous results for standard architectures [10, 12, 14]. We identified the minimum in-degree of all neurons except for the input and first layers as the key quantity to control the transition to linearity of general feedforward neural networks.

We showed that the property of transition to linearity is flexible to the choice of the neuron function Eq. (2). For example, skip connections Eq. (30) and shared weights Eq. (32) do not break the property. Therefore, we believe our framework can be extended to more complicated neuron functions, e.g., attention layers [8]. For non-feedforward networks, such as RNN, recent work [1] showed that they also have a constant NTK. For this reason, we expect transition to linearity also to occur for of non-feedforward networks.

Another direction of future work is better understanding of optimization for DAG networks, which requires a more delicate analysis of the NTK at initialization. Specifically, with multiple training examples, a lower bound on the minimum eigenvalue of the NTK of the DAG networks is sufficient for the PL* condition to hold, thus guaranteeing the convergence of GD/SGD.

## Acknowledgements

We are grateful for support of the NSF and the Simons Foundation for the Collaboration on the Theoretical Foundations of Deep Learning[4] through awards DMS-2031883 and #814639. We also acknowledge NSF support through IIS-1815697 and the TILOS institute (NSF CCF-2112665). We thank Nvidia for the donation of GPUs. This work used the Extreme Science and Engineering Discovery Environment (XSEDE, [21]), which is supported by National Science Foundation grant number ACI-1548562 and allocation TG-CIS210104.

---

[4] https://deepfoundations.ai/

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
