## Appendix

**Notations for set of neurons.**   We extra define the following notations for the proof. For $0 \leq \ell \leq L - 1$, $i \in [d_\ell]$, we use $\mathcal{F}_{\mathcal{S}_i^{(\ell)}}$ to denote the set of all the elements in the vector $f_{\mathcal{S}_i^{(\ell)}}$ (Eq. (5)):

$$\mathcal{F}_{\mathcal{S}_i^{(\ell)}} := \{f : f \in f_{\mathcal{S}_i^{(\ell)}}\}. \tag{19}$$

And we use $\mathcal{P}^{(\ell)}$ to denote the set of all neurons in $\ell'$-th layer i.e., $f^{(\ell')}$ defined in Eq. (4), with $0 \leq \ell' \leq \ell$:

$$\mathcal{P}^{(\ell)} := \{f : f \in f^{(\ell')}, \ell' \leq \ell\}. \tag{20}$$

**Activation functions.**   In Assumption 3.2, we assume the Lipschitz continuity and smoothness for all the activation functions. In the proof of lemmas, e.g., Lemma B.1 and B.2, we only use the fact that they are Lipschitz continuous and smooth, as well as bounded by a constant $\gamma_0 > 0$ at point 0, hence we use $\sigma(\cdot)$ to denote all the activation functions like what we do in Assumption 3.2 for simplicity.

**Notations for derivatives.**   Additionally, in the following we introduce notations of the derivatives, mainly used in the proof of Lemma B.1 and Lemma B.2.

By definition of feedforward neural networks in Section 2, different from the standard neural networks such as FCNs and CNNs in which the connection between neurons are generally only in adjacent layers, the neurons in feedforward neural networks can be arbitrarily connected as long as there is no loop.

To that end, we define $\partial f_{\mathcal{S}_i^{(\ell)}} / \partial f^{(\ell')}$ to be a mask matrix for any $\ell' < \ell$, $i \in [d_\ell]$ to indicate whether the neurons $f_{\mathcal{S}_i^{(\ell)}}$ appear in $f^{(\ell')}$:

$$\left( \frac{\partial f_{\mathcal{S}_i^{(\ell)}}}{\partial f^{(\ell')}} \right)_{j,k} = \mathbb{I}\left\{ \left( f_{\mathcal{S}_i^{(\ell)}} \right)_k \in f_j^{(\ell')} \right\}. \tag{21}$$

And $\partial f_i^{(\ell)} / \partial f_{\mathcal{S}_i^{(\ell)}}$ and $\partial f_i^{(\ell)} / \partial \mathbf{w}_i^{(\ell)}$ are standard derivatives according to Eq. (5):

$$\frac{\partial f_i^{(\ell)}}{\partial f_{\mathcal{S}_i^{(\ell)}}} = \frac{1}{\sqrt{m_i^{(\ell)}}} \left( \mathbf{w}_i^{(\ell)} \right)^T (\sigma_i^{(\ell)})'(\tilde{f}_i^{(\ell)}),$$

$$\frac{\partial f_i^{(\ell)}}{\partial \mathbf{w}_i^{(\ell)}} = \frac{1}{\sqrt{m_i^{(\ell)}}} \left( f_{\mathcal{S}_i^{(\ell)}} \right)^T (\sigma_i^{(\ell)})'(\tilde{f}_i^{(\ell)}).$$

We give a table of notations that will be frequently used (See Table 1). The same notations will be used for ResNets and CNNs with extra subscripts res and cnn respectively.

## A   Examples of feedforward neural networks

Here we show that many common neural networks are special examples of the feedforward neural networks in Definition 2.2.

**Fully-connected neural networks.**   Given an input $\boldsymbol{x} \in \mathbb{R}^d$, an $\ell$-layer fully-connected neural network is defined as follows:

$$f^{(0)} = \boldsymbol{x},$$
$$f^{(\ell)} = \sigma \left( \frac{1}{\sqrt{m_{\ell-1}}} W^{(\ell)} f^{(\ell-1)} \right), \quad \forall \ell \in [L - 1], \tag{22}$$
$$f(\mathbf{W}; \boldsymbol{x}) := f^{(\ell)} = \frac{1}{\sqrt{m_{\ell-1}}} W^{(\ell)} f^{(\ell-1)},$$

Table 1: Table of notations

| Symbol | Meaning |
|---|---|
| $f^{(\ell)}$ | Vector of neurons in $\ell$-th layer |
| $d_\ell$ | Number of neurons in $\ell$-th layer, i.e., length of $f^{(\ell)}$ |
| $f_{\mathcal{S}_i^{(\ell)}}$ | Vector of in-coming neurons of $f_i^{(\ell)}$ |
| $\mathbf{w}_i^{(l)}$ | Weight vector corresponding to in-coming edges of $f_i^{(\ell)}$ |
| $m_i^{(l)}$ | Number of in-coming neurons of $f_i^{(\ell)}$, i.e., length of $f_{\mathcal{S}_i^{(\ell)}}$ and $\mathbf{w}_i^{(l)}$ |
| $\sigma_i^{(l)}$ | Activation function on $\tilde{f}_i^{(\ell)}$ |
| $\mathbf{w}^{(l)}$ | Weight vector corresponding to all incoming edges toward neurons at layer $\ell$ |
| $\mathcal{F}_{\mathcal{S}_i^{(\ell)}}$ | Set of all the elements in the vector $f_{\mathcal{S}_i^{(\ell)}}$ (Eq. (19)) |
| $\mathcal{P}^{(\ell)}$ | Set of all neurons in $f^{(\ell')}$ with $0 \le \ell' \le \ell$ (Eq. (20)) |
| $\mathrm{id}_{\ell_2,j}^{\ell_1,i}$ | Index of $f_j^{(\ell_2)}$ in the vector $f_{\mathcal{S}_i^{(\ell_1)}}$ |

where each $f^{(\ell)}$ is a $m_\ell$-dimensional vector-valued function, and $\mathbf{W} := \left(W^{(1)}, ..., W^{(\ell)}\right)$, $W^{(\ell)} \in \mathbb{R}^{m_{\ell+1} \times m_\ell}$, is the collection of all the weight matrices. Here $\sigma(\cdot)$ is an element-wise activation function, e.g., sigmoid function.

For FCNs, the inputs are the 0-th layer neurons $f^{(0)} = \boldsymbol{x}$ and the outputs are the $\ell$-th layer neurons $f^{(\ell)}$, which have zero in-degrees and zero out-degrees, respectively. For each non-input neuron, its in-degree is the number of neurons in its previous layer, $m_{\ell-1}$; the summation in Eq. (2) turns out to be over all the neurons in the previous layer, which is manifested in the matrix multiplication of $W^{(\ell)} f^{(\ell-1)}$. For this network, the activation functions are the same, except the ones on input and output neurons, where identity functions are used in the definition above.

**DenseNets [9].**  Given an input $\boldsymbol{x} \in \mathbb{R}^d$, an $\ell$-layer DenseNet is defined as follows:

$$f^{(0)} = f_{\text{temp}}^{(0)} = \boldsymbol{x},$$

$$f^{(\ell)} = \sigma \left( \frac{1}{\sqrt{\sum_{l'=0}^{\ell-1} m_{\ell'}}} W^{(\ell)} f_{\text{temp}}^{(\ell-1)} \right), \tag{23}$$

$$f_{\text{temp}}^{(\ell)} = \left[ \left( f_{\text{temp}}^{(\ell-1)} \right)^T, \left( f^{(\ell)} \right)^T \right]^T, \quad \forall \ell \in [L-1],$$

$$f(\mathbf{W}; \boldsymbol{x}) := f^{(\ell)} = \frac{1}{\sqrt{\sum_{\ell'=0}^{L-1} m_{\ell'}}} W^{(\ell)} f_{\text{temp}}^{(\ell-1)}, \tag{24}$$

where $\mathbf{W} = \left(W^{(1)}, ..., W^{(\ell)}\right)$ is the collection of all the weight matrices. Here $\sigma(\cdot)$ is an element-wise activation function and for each $\ell \in [L]$, $W^{(\ell)} \in \mathbb{R}^{m_\ell \times \sum_{\ell'=0}^{\ell-1} m_{\ell'}}$.

The DenseNet shares much similarity with the fully-connected neural network, except that each non-input neuron depends on all the neurons in previous layers. This difference makes the in-degree of the neuron be $\sum_{\ell'=0}^{\ell-1} m_{\ell'}$.

**Neural networks with randomly dropped edges.**  Given a network $f$ built from a DAG, for any neuron $f_v$, where $v \in \mathcal{V} \backslash \mathcal{V}_{\text{input}}$, according to Eq. (22), it is defined by

$$f_v = \sigma_v(\tilde{f}_v), \quad \tilde{f}_v = \frac{1}{\sqrt{\mathsf{in}(v)}} \sum_{u \in \mathcal{S}_{\text{in}}(v)} w_{(u,v)} f_u.$$

If each edge $(u, v)$ is randomly dropped with parameter $p \in (0, 1)$, then the above equation becomes

$$f_v = \sigma_v(\tilde{f}_v), \quad \tilde{f}_v = \frac{1}{\sqrt{\mathsf{in}(v)}} \sum_{u \in \mathcal{S}_{\text{in}}(v)} w_{(u,v)} f_u \cdot \mathbb{I}_{\{\xi_{u,v} \ge p\}},$$

where $\xi_{u,v}$ is i.i.d. drawn from Bernoulli($p$).

To interpret such an architecture, we can simply remove the edges $(u, v)$ in the DAG where $\xi_{u,v} < p$. Then it is not hard to see that the new DAG network corresponds to the network with randomly dropped edges.

Similarly, for a neural network with randomly dropped edges in multiple layers, we can remove all the edges whose corresponding $\xi$ is less than $p$. Then the resulting DAG can describe this network architecture.

We note the similarity of this network with the popularly used dropout layer [20], both of which have a mechanism of randomly dropping out neurons/edges. However, the major difference is that, neural networks with dropout layers dynamically remove (or put mask on) neurons/edges during training, while the networks we considered only here drop edges and are fixed during training.

## B  Proof of Theorem 3.6

We will first compute the Hessian matrix of the network function then show how to bound the spectral norm of it.

We denote for each $\ell \in [L]$,

$$\underline{m}_\ell := \inf_{i \in [d_\ell]} m_i^{(\ell)}, \quad \overline{m}_\ell := \sup_{i \in [d_\ell]} m_i^{(\ell)}. \tag{25}$$

By Assumption 3.5, it is not hard to infer that $\overline{m}_\ell$ and $\underline{m}_\ell$ are also polynomial in $m$.

Fixing $k \in [d_\ell]$, to bound $\|H_{f_k}\|$, we will first bound the spectral norm of the each Hessian block $H_{f_k}^{(\ell_1,\ell_2)}$, which takes the form

$$H_{f_k}^{(\ell_1,\ell_2)} := \frac{\partial^2 f_k}{\partial \mathbf{w}^{(\ell_1)} \partial \mathbf{w}^{(\ell_2)}}, \quad k \in [d_\ell], \;\; \ell_1, \ell_2 \in [L].$$

Without lose of generality, we assume $1 \le \ell_1 \le \ell_2 \le L$ and we start with the simple case when $\ell_2 = L$.

If $\ell_1 = \ell_2 = L$, $H_{f_k}^{(L,L)}$ is simply a zero matrix since $f_k(\mathbf{w})$ is linear in $\mathbf{w}^{(\ell)}$.

If $1 \le \ell_1 < \ell_2 = L$, we will use the following Lemma:

**Lemma B.1.** *Given $\ell' \ge 1$, for any $\ell' + 1 \le \ell \le L$, $\mathbf{w} \in \mathsf{B}(\mathbf{w}_0, R)$, and $j \in [d_\ell]$, we have, with probability at least $1 - \exp(-C_{\ell,\ell'}^f \log^2 m)$,*

$$\left\| \frac{\partial f_{S_j^{(\ell)}}}{\partial \mathbf{w}^{(\ell')}} \right\| = O\left( \max_{\ell'+1 \le p \le \ell} \frac{\sqrt{m_j^{(\ell)}}}{\sqrt{\underline{m}_p}} (\log m + R)^{\ell'} \right) = \tilde{O}\left( \max_{\ell'+1 \le p \le \ell} \frac{\sqrt{m_j^{(\ell)}}}{\sqrt{\underline{m}_p}} R^{\ell'} \right), \tag{26}$$

$$\left\| \frac{\partial f_{S_j^{(\ell)}}}{\partial \mathbf{w}^{(\ell')}} \right\|_F = O\left( \sqrt{m_j^{(\ell)}} (\log m + R)^{\ell-1} \right) = \tilde{O}\left( \sqrt{m_j^{(\ell)}} R^{\ell-1} \right), \tag{27}$$

*where $C_{\ell,\ell'}^f > 0$ is a constant.*

See the proof in Appendix H.

By Lemma B.1, with probability at least $1 - \exp(-\Omega(\log^2 m))$,

$$\left\| H_{f_k}^{(\ell_1,L)} \right\| = \left\| \frac{1}{\sqrt{m_k^{(\ell)}}} \frac{\partial f_{S_k^{(\ell)}}}{\partial \mathbf{w}^{(\ell_1)}} \right\| = O\left( \max_{\ell_1+1 \le \ell \le L} \frac{1}{\sqrt{\underline{m}_\ell}} (\log m + R)^{\ell_1} \right) = \tilde{O}(R^{\ell_1}/\sqrt{m}).$$

For the rest of blocks that $1 \le \ell_1 \le \ell_2 \le L - 1$, we will use the following lemma to bound their spectral norms:

**Lemma B.2.** *Given $1 \leq \ell_1 \leq \ell_2 \leq L - 1$, for any $\ell_2 < \ell \leq L$, $\mathbf{w} \in \mathsf{B}(\mathbf{w}_0, R)$, and $j \in [d_\ell]$, we have, with probability at least $1 - \exp(-\Omega(\log^2 m))$,*

$$\left\| \frac{\partial^2 \tilde{f}_j^{(\ell)}}{\partial \mathbf{w}^{(\ell_1)} \partial \mathbf{w}^{(\ell_2)}} \right\| = O\left( \max_{\ell_1+1 \leq p \leq \ell} \frac{1}{\sqrt{m}\, p} (\log m + R)^{\ell^2} \right) = \tilde{O}\left( \max_{\ell_1+1 \leq p \leq \ell} \frac{R^{\ell^2}}{\sqrt{m}\, p} \right). \tag{28}$$

See the proof in Appendix I.

*Remark* B.3. Note that the above results hold for any $\ell \leq L$. When $\ell = L$, $\tilde{f}_j^{(\ell)} = f_j$ which is what we need to show the transition to linearity of $f_j$. When $\ell < L$, as discussed before, we can regard $\tilde{f}_j^{(\ell)}$ as a function of its parameters. We note that $\tilde{f}_j^{(\ell)}$ with $\ell < L$ will also transition to linearity by applying the same analysis for $f_j$, which is the result of Theorem 3.8.

By letting $\ell = L$ in Lemma B.2, for any $1 \leq \ell_1 \leq \ell_2 \leq L - 1$, with probability at least $1 - \exp(-\Omega(\log^2 m))$,

$$\left\| H_{f_k}^{(\ell_1, \ell_2)} \right\| = O\left( \max_{\ell_1 \leq \ell \leq L} \frac{1}{\sqrt{m}\, \ell} (\log m + R)^{\ell^2} \right) = O((\log m + R)^{L^2}/\sqrt{m}) = \tilde{O}(R^{L^2}/\sqrt{m}).$$

Finally by Lemma K.1, the spectral norm of $H_{f_k}$ can be bounded by the summation of the spectral norm of all the Hessian blocks, i.e., $\|H_{f_k}\| \leq \sum_{\ell_1, \ell_2} \|H_{f_k}^{(\ell_1, \ell_2)}\|$. Applying the union bound over the indices of layers $\ell_1, \ell_2$, we finish the proof.

## C Feedforward neural networks with multiple output

In cases of multiple output neurons, the network function is vector-valued and its Hessian is a three-order tensor. The spectral norm of Hessian is defined in a standard way, i.e.,

$$\|\mathbf{H}_f(\mathbf{w})\| := \sup_{\|\mathbf{v}\|=\|\mathbf{u}\|=\|\mathbf{s}\|=1} \sum_{i,j,k} (\mathbf{H}_f(\mathbf{w}))_{i,j,k}\, v_i u_j s_k,$$

where $\mathbf{s} \in \mathbb{R}^{d_\ell}$ and $\mathbf{v}, \mathbf{u}$ have the same dimension with $\mathbf{w}$. It is not hard to see that $\|\mathbf{H}_f(\mathbf{w})\| \leq d_\ell \max_{k \in [d_\ell]} \|H_{f_k}(\mathbf{w})\|$.

If the number of output neurons $d_\ell$ is bounded (as in most practical cases), the spectral norm of the Hessian of $f$ is also of the order $\tilde{O}(1/\sqrt{m})$, with high probability, as a direct consequence of Theorem 3.6.

**Corollary C.1.** *Suppose Assumption 3.1, 3.2 and 3.5 hold. Given a fixed radius $R > 0$, for all $\mathbf{w} \in \mathsf{B}(\mathbf{w}_0, R)$, with probability at least $1 - \exp(-\Omega(\log^2 m))$ over the random initialization $\mathbf{w}_0$, a vector-valued feedforward neural network $f$ satisfies*

$$\|\mathbf{H}_f(\mathbf{w})\| = \tilde{O}\left( \frac{R^{L^2}}{\sqrt{m}} \right). \tag{29}$$

## D Feedforward neural networks with skip connections

In this section, we discuss the property of transition to linearity holds for networks with skip connection.

We formally define the skip connection in the following. We add a skip connection to each neuron then the neuron functions Eq. (5) become

$$f_{i,\text{res}}^{(\ell)} = \sigma_i^{(\ell)}\left(\tilde{f}_{i,\text{res}}^{(\ell)}\right) + f_{B(\ell,i),\text{res}}^{(A(\ell,i))}, \quad \tilde{f}_{i,\text{res}}^{(\ell)} = \frac{1}{\sqrt{m_i^{(\ell)}}} \left(\mathbf{w}_i^{(\ell)}\right)^T f_{\mathcal{S}_i^{(\ell)},\text{res}}, \tag{30}$$

where $1 \leq \ell \leq L - 1$ and $i \in [d_\ell]$. Here $A(\ell, i) \in \{0, \cdots, \ell - 1\}$ denotes the layer index of the connected neuron by skip connection with respect to $f_{i,\text{res}}^{(\ell)}$ and $B(\ell, i) \in [d_{A(\ell,i)}]$.

And for the output layer $\ell = L$, we define

$$f_{i,\mathrm{res}}^{(L)} = \tilde{f}_{i,\mathrm{res}}^{(L)} = \frac{1}{\sqrt{m_i^{(L)}}} \left( \mathbf{w}_i^{(L)} \right)^T f_{\mathcal{S}_i^{(L)},\mathrm{res}},$$

where $i \in [d_L]$.

The following theorem shows the property of transition to linearity holds for networks with skip connections. The proof of the theorem follows the almost identical idea with the proof of Theorem 3.6, hence we present the proof sketch and focus on the arguments that are new for $f_{\mathrm{res}}$.

**Theorem D.1** (Scaling of the Hessian norm for $f_{\mathrm{res}}$). *Suppose Assumption 3.1, 3.2 and 3.5 hold. Given a fixed radius $R > 0$, for all $\mathbf{w} \in \mathsf{B}(\mathbf{w}_0, R)$, with probability at least $1 - \exp(-\Omega(\log^2 m))$ over the random initliazation of $\mathbf{w}_0$, each output neuron $f_{k,\mathrm{res}}$ satisfies*

$$\left\| H_{f_{k,\mathrm{res}}}(\mathbf{w}) \right\| = \tilde{O}\left( \frac{R^{L^2}}{\sqrt{m}} \right), \quad \ell \in [L], \ \ k \in [d_\ell]. \tag{31}$$

*Proof sketch of Theorem D.1.* For each output $f_{k,\mathrm{res}}$, where $k \in [d_\ell]$, similar to the proof of Theorem 3.6, we bound the spectral norm of each Hessian block, i.e., $\frac{\partial^2 f_{k,\mathrm{res}}}{\partial \mathbf{w}^{(\ell_1)} \partial \mathbf{w}^{(\ell_2)}}$. Without loss of generality, we assume $1 \le \ell_1 \le \ell_2 \le L$.

Similar to Eq.(13), we derive the expression of the Hessian block by definition:

$$\frac{\partial^2 f_{k,\mathrm{res}}}{\partial \mathbf{w}^{(\ell_1)} \partial \mathbf{w}^{(\ell_2)}} = \sum_{\ell'=\ell_2}^{L} \sum_{i=1}^{d_{\ell'}} \frac{\partial^2 f_{i,\mathrm{res}}^{(\ell')}}{\partial \mathbf{w}^{(\ell_1)} \partial \mathbf{w}^{(\ell_2)}} \frac{\partial f_{k,\mathrm{res}}}{\partial f_{i,\mathrm{res}}^{(\ell')}} := \sum_{\ell'=\ell_2}^{L} G_{k,\mathrm{res}}^{L,\ell'}.$$

And again by chain rule of derivatives, each $G_{k,\mathrm{res}}^{L,\ell'}$ can be written as

$$G_{k,\mathrm{res}}^{L,\ell'} = \underbrace{\frac{1}{\sqrt{m_k^{(L)}}} \sum_{r=\ell'}^{L-1} \sum_{s:f_{s,\mathrm{res}}^{(r)} \in \mathcal{F}_{\mathcal{S}_k^{(L)}}} \left( \mathbf{w}_k^{(L)} \right)_{\mathrm{id}_{r,s}^{L,k}} \sigma'\left( \tilde{f}_{s,\mathrm{res}}^{(r)} \right) G_{s,\mathrm{res}}^{r,\ell'}}_{T_1}$$

$$+ \underbrace{\frac{1}{\sqrt{m_k^{(L)}}} \sum_{r=\ell'}^{L-1} \sum_{s:f_{s,\mathrm{res}}^{(r)} \in \mathcal{F}_{\mathcal{S}_k^{(L)}}} \left( \mathbf{w}_k^{(L)} \right)_{\mathrm{id}_{r,s}^{L,k}} \sigma'\left( \tilde{f}_{B(\ell',s),\mathrm{res}}^{(A(\ell',s))} \right) G_{B(\ell',s),\mathrm{res}}^{A(\ell',s),\ell'}}_{T_2}$$

$$+ \underbrace{\frac{1}{\sqrt{m_k^{(L)}}} \sum_{i:f_{i,\mathrm{res}}^{(\ell')} \in \mathcal{F}_{\mathcal{S}_{k,\mathrm{res}}^{(L)}}} \left( \mathbf{w}_k^{(L)} \right)_{\mathrm{id}_{\ell',i}^{L,k}} \left( \sigma''\left( \tilde{f}_{i,\mathrm{res}}^{(\ell')} \right) \frac{\partial \tilde{f}_{i,\mathrm{res}}^{(\ell')}}{\partial \mathbf{w}^{(\ell_1)}} \left( \frac{\partial \tilde{f}_{i,\mathrm{res}}^{(\ell')}}{\partial \mathbf{w}^{(\ell_2)}} \right)^T \right)}_{T_3}$$

$$+ \underbrace{\frac{1}{\sqrt{m_k^{(L)}}} \sum_{i:f_{i,\mathrm{res}}^{(\ell')} \in \mathcal{F}_{\mathcal{S}_{k,\mathrm{res}}^{(L)}}} \left( \mathbf{w}_k^{(L)} \right)_{\mathrm{id}_{\ell',i}^{L,k}} \left( \sigma''\left( \tilde{f}_{B(\ell',i),\mathrm{res}}^{(A(\ell',i))} \right) \frac{\partial \tilde{f}_{B(\ell',i),\mathrm{res}}^{(A(\ell',i))}}{\partial \mathbf{w}^{(\ell_1)}} \left( \frac{\partial \tilde{f}_{B(\ell',i),\mathrm{res}}^{(A(\ell',i))}}{\partial \mathbf{w}^{(\ell_2)}} \right)^T \right)}_{T_4},$$

where $\mathcal{F}_{\mathcal{S}_{k,\mathrm{res}}^{(L)}} := \{ f : f \in f_{\mathcal{S}_{k,\mathrm{res}}^{(L)}} \}$ and $\mathrm{id}_{\ell',i}^{L,k} := \{ p : \left( f_{\mathcal{S}_k^{(L)},\mathrm{res}} \right)_p = f_{i,\mathrm{res}}^{(\ell')} \}$.

Note that the new terms which are induced by the skip connection in the above equation are

$$T_2 = \frac{1}{\sqrt{m_k^{(L)}}} \sum_{r=\ell'}^{L-1} \sum_{s:f_{s,\mathrm{res}}^{(r)} \in \mathcal{F}_{\mathcal{S}_k^{(L)}}} \left( \mathbf{w}_k^{(L)} \right)_{\mathrm{id}_{r,s}^{L,k}} \sigma'\left( \tilde{f}_{B(\ell',s),\mathrm{res}}^{(A(\ell',s))} \right) G_{B(\ell',s),\mathrm{res}}^{A(\ell',s),\ell'},$$

and

$$T_4 = \frac{1}{\sqrt{m_k^{(L)}}} \sum_{i: f_{i,\text{res}}^{(\ell')} \in \mathcal{F}_{\mathcal{S}_{k,\text{res}}^{(L)}}} \left(\mathbf{w}_k^{(L)}\right)_{\text{id}_{\ell',i}^{L,k}} \left(\sigma'' \left(\tilde{f}_{B(\ell',i),\text{res}}^{(A(\ell',i))}\right) \frac{\partial \tilde{f}_{B(\ell',i),\text{res}}^{(A(\ell',i))}}{\partial \mathbf{w}^{(\ell_1)}} \left(\frac{\partial \tilde{f}_{B(\ell',i),\text{res}}^{(A(\ell',i))}}{\partial \mathbf{w}^{(\ell_2)}}\right)^T \right).$$

These two new terms take the same form with the original two terms i.e., $T_1$ and $T_3$, which are matrix Gaussian series with respect to the random variables $\mathbf{w}_k^{(L)}$. Therefore, we can use the same method as $T_1$ and $T_3$ to bound the spectral norm of $T_2$ and $T_4$.

As $A(\ell', i) < \ell'$ by definition, the bound on $T_2$ and $T_4$ will be automatically included in our recursive analysis. Then the rest of the proof is identical to the one for feedforward neural networks, i.e., the proof of Theorem 3.6.

$\square$

# E  Feedforward neural networks with shared weights, e.g., convolutional neural networks

In this section, we consider the feedforward neural networks where weight parameters are shared, e.g., convolutional neural networks, as an extension to our result where we assume each weight parameter $w_e \in \mathcal{W}$ is initialized i.i.d. We will show that such feedforward neural networks in which the weight parameters are shared constant times, i.e., independent of the width $m$, the property of transition to linearity still holds.

We formally define the networks with shared weights in the following:

$$f_{i,j,\text{cnn}}^{(\ell)} = \sigma_i^{(\ell)}\left(\tilde{f}_{i,j,\text{cnn}}^{(\ell)}\right), \quad \tilde{f}_{i,j,\text{cnn}}^{(\ell)} = \frac{1}{\sqrt{m_{i,j}^{(\ell)}}} \left(\mathbf{w}_i^{(\ell)}\right)^T f_{\mathcal{S}_{i,j}^{(\ell)},\text{cnn}}, \tag{32}$$

where $1 \leq \ell \leq L$, $i \in [d_\ell]$. We introduce new index $j \in [D(\ell, i)]$ where $D(\ell, i)$ denotes the number of times that weights $\mathbf{w}_i^{(\ell)}$ are shared. Note that the element in $f_{\mathcal{S}_{i,j}^{(\ell)},\text{cnn}}$ is allowed to be 0, corresponding to the zero padding which is commonly used in CNNs.

We similarly denote the output of the networks $f_{i,j,\text{cnn}}^{(L)}$ by $f_{i,j,\text{cnn}}$.

To see how CNNs fit into this definition, we consider a CNN with 1-D convolution as a simple example.

**Convolutional neural networks**  Given input $\boldsymbol{x} \in \mathbb{R}^d$, an $\ell$-layer convolutional neural network is defined as follows:

$$f^{(0)} = \boldsymbol{x},$$

$$f^{(\ell)} = \sigma\left(\frac{1}{\sqrt{m_{\ell-1} \times p}} W^{(\ell)} * f^{(\ell-1)}\right), \quad \forall l \in [L-1],$$

$$f_i(\mathbf{W}; \boldsymbol{x}) = \frac{1}{\sqrt{m_{\ell-1} \times d}} \left\langle W_{[i,:,:]}^{(\ell)}, f^{(\ell-1)} \right\rangle, \quad \forall i \in [d_\ell], \tag{33}$$

where $\mathbf{W} = \left(W^{(1)}, ..., W^{(\ell)}\right)$ is the collection of all the weight matrices.

We denote the size of the window by $p \times 1$, hence $W^{(\ell)} \in \mathbb{R}^{m_\ell \times m_{\ell-1} \times p}$ for $\ell \in [L-1]$. We assume the stride is 1 for simplicity, and we do the standard zero-padding to each $f^{(\ell)}$ such that for each $\ell \in [L-1]$, $f^{(\ell)} \in \mathbb{R}^{m_\ell \times d}$. At the last layer, as $f^{(\ell-1)} \in \mathbb{R}^{m_{\ell-1} \times d}$ and $W^{(\ell)} \in \mathbb{R}^{m_L \times m_{\ell-1} \times d}$, we do the matrix inner product for each $i \in [d_\ell]$.

Now we show how Eq. (33) fits into Eq. (32). For $\ell \in [L-1]$, in Eq. (33), each component of $f^{(\ell)} \in \mathbb{R}^{m_\ell \times d}$ is computed as

$$f_{i,j}^{(\ell)} = \sigma\left(\frac{1}{\sqrt{m_{\ell-1} \times p}} \left\langle W_{[i,:,:]}^{(\ell)}, f_{[:,j-\lceil \frac{p-1}{2} \rceil : j+\lceil \frac{p-1}{2} \rceil]}^{(\ell-1)} \right\rangle\right).$$

Therefore, $m_{i,j}^{(\ell)}$, $\mathbf{w}_i^{(\ell)}$ and $f_{\mathcal{S}_{i,j}^{(\ell)},\mathrm{cnn}}$ in Eq. (32) correspond to $m_{\ell-1} \times p$, $W_{[i,:,:]}^{(\ell)}$ and $f_{[:,j-\lceil\frac{p-1}{2}\rceil:j+\lceil\frac{p-1}{2}\rceil]}^{(\ell-1)}$ respectively. For $\ell = L$, $m_{i,j}^{(L)}$ corresponds to $m_{L-1} \times d$ and $f_{\mathcal{S}_{i,j}^{(L)},\mathrm{cnn}}$ corresponds to $f^{(L-1)}$. Then we can see our definition of networks with shared weights, i.e., Eq. (32) includes standard CNN as an special example.

Similar to Theorem 3.6, we will show that the spectral norm of its Hessian can be controlled, hence the property of transition to linearity will hold for $f_{\mathrm{cnn}}^{(\ell)}$. The proof of the following theorem follows the almost identical idea with the proof of Theorem 3.6, hence we present the proof sketch and focus on the arguments that are new for $f_{\mathrm{cnn}}$.

**Theorem E.1.** *Suppose Assumption 3.1, 3.2 and 3.5 hold. Given a fixed radius $R > 0$, for all $\mathbf{w} \in \mathsf{B}(\mathbf{w}_0, R)$, with probability at least $1 - \exp(-\Omega(\log^2 m))$ over the random initliazation of $\mathbf{w}_0$, each output neuron $f_{i,j,\mathrm{cnn}}(\mathbf{w})$ satisfies*

$$\left\| H_{f_{i,j,\mathrm{cnn}}}(\mathbf{w}) \right\| = O\left( (\log m + R)^{\ell^2}/\sqrt{m} \right) = \tilde{O}\left( R^{\ell^2}/\sqrt{m} \right), \quad \ell \in [L], \ i \in [d_\ell], \ j \in [D(i,\ell)].$$

$$(34)$$

*Proof sketch of Theorem E.1.* Similar to the proof of Theorem 3.6, by Lemma K.1, the spectral norm of $H_{f_{i,j,\mathrm{cnn}}}$ can be bounded by the summation of the spectral norm of all the Hessian blocks, i.e., $\|H_{f_{i,j,\mathrm{cnn}}}\| \le \sum_{\ell_1,\ell_2} \|H_{f_{i,j,\mathrm{cnn}}}^{(\ell_1,\ell_2)}\|$, where $H_{f_{i,j,\mathrm{cnn}}}^{(\ell_1,\ell_2)} := \frac{\partial^2 f_k}{\partial \mathbf{w}^{(\ell_1)}\partial \mathbf{w}^{(\ell_2)}}$. Therefore, it suffices to bound the spectral norm of each block. Without lose of generality, we consider the block with $1 \le \ell_1 \le \ell_2 \le L$.

By the chain rule of derivatives, we can write the Hessian block into:

$$\frac{\partial^2 f_{i,j,\mathrm{cnn}}}{\partial \mathbf{w}^{(\ell_1)}\partial \mathbf{w}^{(\ell_2)}} = \sum_{\ell'=\ell_2}^{L} \sum_{k=1}^{d_{\ell'}} \sum_{t=1}^{D(k,\ell')} \frac{\partial^2 f_{k,t,\mathrm{cnn}}^{(\ell')}}{\partial \mathbf{w}^{(\ell_1)}\partial \mathbf{w}^{(\ell_2)}} \frac{\partial f_{i,j,\mathrm{cnn}}}{\partial f_{k,t,\mathrm{cnn}}^{(\ell')}} := \sum_{\ell'=\ell_2}^{L} G_{i,j,\mathrm{cnn}}^{L,\ell'}. \quad (35)$$

For each $G_{i,j,\mathrm{cnn}}^{L,\ell'}$, since $f_{i,j,\mathrm{cnn}}^{(\ell')} = \sigma\left(\tilde{f}_{i,j,\mathrm{cnn}}^{(\ell')}\right)$, again by the chain rule of derivatives, we have

$$
\begin{aligned}
G_{i,j,\mathrm{cnn}}^{L,\ell'} =& \sum_{k=1}^{d_{\ell'}} \sum_{t=1}^{D(k,\ell')} \frac{\partial^2 \tilde{f}_{k,t,\mathrm{cnn}}^{(\ell')}}{\partial \mathbf{w}^{(\ell_1)}\partial \mathbf{w}^{(\ell_2)}} \frac{\partial f_{i,j,\mathrm{cnn}}}{\partial \tilde{f}_{k,t,\mathrm{cnn}}^{(\ell')}} \\
&+ \frac{1}{\sqrt{m_{i,j}^{(L)}}} \sum_{k,t:f_{k,t,\mathrm{cnn}}^{(\ell')}\in\mathcal{F}_{\mathcal{S}_{i,j,\mathrm{cnn}}^{(L)}}} \left(\mathbf{w}_i^{(L)}\right)_{\mathrm{id}_{\ell',k,t}^{L,i,j}} \sigma''\left(\tilde{f}_{k,t,\mathrm{cnn}}^{(\ell')}\right) \frac{\partial \tilde{f}_{k,t,\mathrm{cnn}}^{(\ell')}}{\partial \mathbf{w}^{(\ell_1)}} \left(\frac{\partial \tilde{f}_{k,t,\mathrm{cnn}}^{(\ell')}}{\partial \mathbf{w}^{(\ell_2)}}\right)^T \\
=& \frac{1}{\sqrt{m_{i,j}^{(L)}}} \sum_{r=\ell'}^{L-1} \sum_{k,t:f_{k,t,\mathrm{cnn}}^{(r)}\in\mathcal{F}_{\mathcal{S}_{i,j,\mathrm{cnn}}^{(L)}}} \left(\mathbf{w}_i^{(L)}\right)_{\mathrm{id}_{r,k,t}^{L,i,j}} \sigma'\left(\tilde{f}_{k,t,\mathrm{cnn}}^{(r)}\right) G_{k,t,\mathrm{cnn}}^{r,\ell'} \\
&+ \frac{1}{\sqrt{m_{i,j}^{(L)}}} \sum_{k,t:f_{k,t,\mathrm{cnn}}^{(\ell')}\in\mathcal{F}_{\mathcal{S}_{i,j,\mathrm{cnn}}^{(L)}}} \left(\mathbf{w}_i^{(L)}\right)_{\mathrm{id}_{\ell',k,t}^{L,i,j}} \sigma''\left(\tilde{f}_{k,t,\mathrm{cnn}}^{(\ell')}\right) \frac{\partial \tilde{f}_{k,t,\mathrm{cnn}}^{(\ell')}}{\partial \mathbf{w}^{(\ell_1)}} \left(\frac{\partial \tilde{f}_{k,t,\mathrm{cnn}}^{(\ell')}}{\partial \mathbf{w}^{(\ell_2)}}\right)^T,
\end{aligned}
$$

where $\mathcal{F}_{\mathcal{S}_{i,j,\mathrm{cnn}}^{(L)}} := \{f : f \in f_{\mathcal{S}_{i,j}^{(L)}}\}$ and $\mathrm{id}_{\ell',k,t}^{L,i,j} := \{p : \left(f_{\mathcal{S}_{i,j}^{(L)},\mathrm{cnn}}\right)_p = f_{k,t,\mathrm{cnn}}^{(\ell')}\}$.

Compared to the derivation for standard feedforward neural networks, i.e., Eq. (14), there is an extra summation over the index $t$, whose carnality is at most $D(k,\ell')$. Recall that $D(k,\ell')$ denotes the number of times that the weight parameters $\mathbf{w}_k^{(\ell')}$ is shared. Therefore, as we assume $D(k,\ell')$ is independent of the width $m$, the norm bound will have the same order of $m$. Consequently, the spectral norm of each $G_{i,j,\mathrm{cnn}}^{L,\ell'}$ can be recursively bounded then Eq. (34) holds.

$\square$

# F  Feedforward neural networks with bottleneck neurons

In this section, we show that constant number of bottleneck neurons which serve as incoming neurons will not break the linearity.

We justify this claim based on the recursive relation in Eq. (13), which is used to prove the small spectral norm of the Hessian of the network function, hence proving the transition to linearity.

Recall that each Hessian block can be written into:

$$\frac{\partial^2 f_k}{\partial \mathbf{w}^{(\ell_1)} \partial \mathbf{w}^{(\ell_2)}} = \sum_{\ell'=\ell_2}^{L} \sum_{i=1}^{d_{\ell'}} \frac{\partial^2 f_i^{(\ell')}}{\partial \mathbf{w}^{(\ell_1)} \partial \mathbf{w}^{(\ell_2)}} \frac{\partial f_k}{\partial f_i^{(\ell')}} := \sum_{\ell'=\ell_2}^{L} G_k^{L,\ell'}. \tag{36}$$

For each $G_k^{L,\ell'}$, we have a recursive form

$$G_k^{L,\ell'} = \sum_{i=1}^{d_{\ell'}} \frac{\partial^2 \tilde{f}_i^{(\ell')}}{\partial \mathbf{w}^{(\ell_1)} \partial \mathbf{w}^{(\ell_2)}} \frac{\partial f_k}{\partial \tilde{f}_i^{(\ell')}} + \frac{1}{\sqrt{m_k^{(L)}}} \sum_{i:f_i^{(\ell')} \in \mathcal{F}_{\mathcal{S}_k^{(L)}}} \left(\mathbf{w}_k^{(L)}\right)_{\mathrm{id}_{\ell',i}^{L,k}} \sigma'' \left(\tilde{f}_i^{(\ell')}\right) \frac{\partial \tilde{f}_i^{(\ell')}}{\partial \mathbf{w}^{(\ell_1)}} \left(\frac{\partial \tilde{f}_i^{(\ell')}}{\partial \mathbf{w}^{(\ell_2)}}\right)^T$$

$$= \frac{1}{\sqrt{m_k^{(L)}}} \sum_{r=\ell'}^{L-1} \sum_{i:f_i^{(r)} \in \mathcal{F}_{\mathcal{S}_k^{(L)}}} \left(\mathbf{w}_k^{(L)}\right)_{\mathrm{id}_{r,i}^{L,k}} \sigma' \left(\tilde{f}_s^{(r)}\right) G_i^{r,\ell'}$$

$$+ \frac{1}{\sqrt{m_k^{(L)}}} \sum_{i:f_i^{(\ell')} \in \mathcal{F}_{\mathcal{S}_k^{(L)}}} \left(\mathbf{w}_k^{(L)}\right)_{\mathrm{id}_{\ell',i}^{L,k}} \sigma'' \left(\tilde{f}_i^{(\ell')}\right) \frac{\partial \tilde{f}_i^{(\ell')}}{\partial \mathbf{w}^{(\ell_1)}} \left(\frac{\partial \tilde{f}_i^{(\ell')}}{\partial \mathbf{w}^{(\ell_2)}}\right)^T, \tag{37}$$

where $\mathcal{F}_{\mathcal{S}_k^{(L)}} := \{f : f \in f_{\mathcal{S}_k^{(L)}}\}$ and $\mathrm{id}_{\ell',i}^{L,k} := \{p : \left(f_{\mathcal{S}_k^{(L)}}\right)_p = f_i^{(\ell')}\}$.

As mentioned in Section 3.1, to prove the spectral norm of $G_k^{L,\ell'}$ is small, we need to bound the matrix variance, which suffices to bound the spectral norm of

$$\frac{1}{\sqrt{m_k^{(L)}}} \sum_{i:f_i^{(r)} \in \mathcal{F}_{\mathcal{S}_k^{(L)}}} G_i^{r,\ell'} \quad \text{and} \quad \frac{1}{\sqrt{m_k^{(L)}}} \sum_{i:f_i^{(\ell')} \in \mathcal{F}_{\mathcal{S}_k^{(L)}}} \frac{\partial \tilde{f}_i^{(\ell')}}{\partial \mathbf{w}^{(\ell_1)}} \left(\frac{\partial \tilde{f}_i^{(\ell')}}{\partial \mathbf{w}^{(\ell_2)}}\right)^T.$$

For the first quantity, if all $\tilde{f}_i^{(r)}$ are neurons with large in-degree, which is the case of our analysis by Assumption 3.5, then each $\tilde{f}_i^{(r)}$ will transition to linearity by Theorem 3.8. This is manifested as small spectral norm of $G_i^{r,\ell'}$ for all $i$. If some of $\tilde{f}_i^{(r)}$ are neurons with small in-degree, their corresponding $G_i^{r,\ell'}$ can be of a larger order, i.e., $O(1)$. However, note that the cardinally of the set $\mathcal{F}_{\mathcal{S}_k^{(L)}}$ is $m_k^{(L)}$. As long as the number of such neurons is not too large, i.e., $o\left(m_k^{(L)}\right)$, the order of the summation will be not affected. Therefore, the desired bound for the matrix variance will be the same hence the recursive argument can still apply.

The same analysis works for the second quantity as well. Neurons with small in-degree can make the norm of $\frac{\partial \tilde{f}_i^{(\ell')}}{\partial \mathbf{w}^{(\ell_1)}} \left(\frac{\partial \tilde{f}_i^{(\ell')}}{\partial \mathbf{w}^{(\ell_2)}}\right)^T$ be of a larger order. However, as long as the number of such neurons is not too large, the bound still holds.

For example, for the bottleneck neural network which has a narrow hidden layer (i.e., bottleneck layer) while the rest of hidden layers are wide, all neurons in the next layer to the bottleneck layer are bottleneck neurons. Such bottleneck neural networks were shown to break transition to linearity in [14]. However, we observe that for such bottleneck neural networks, the number of bottleneck is large, a fixed fraction of all neurons. With our analysis, if we add trainable connections to the bottleneck neurons such that almost all (except a small number of) bottleneck neurons become neurons with sufficiently large in-degrees, then the resulting network can have the property of transition to linearity.

# G  Proof of Proposition 4.4

Note that for any $k \in [d_\ell]$,

$$\|\nabla_{\mathbf{w}} f_k(\mathbf{w}_0)\| \geq \|\nabla_{\mathbf{w}^{(\ell)}} f_k(\mathbf{w}_0)\| = \left\|\frac{1}{\sqrt{m_k^{(\ell)}}} f_{\mathcal{S}_k^{(\ell)}}\right\| = \frac{1}{\sqrt{m_k^{(\ell)}}} \left\|f_{\mathcal{S}_k^{(\ell)}}\right\|.$$

Since $f_{\mathcal{S}_k^{(\ell)}}$ contains neurons from $\mathcal{P}^{(\ell)}$ (defined in Eq. (20)), in the following we prove $\mathbb{E}_{\mathbf{x}, \mathbf{w}_0} \left| f_i^{(\ell)} \right|^2$ is uniformly bounded from 0 for any $\ell \in \{0, 1, ..., L-1\}$, $i \in [d_\ell]$.

Specifically, we will prove by induction that $\forall\, \ell \in \{0, 1, ..., L-1\}$, $\forall\, i \in [d_\ell]$,

$$\mathbb{E}_{\mathbf{x}} \mathbb{E}_{\mathbf{w}_0} [|f_i^{(\ell)}|^2] \geq \min\left\{1, \min_{1 \leq j \leq \ell} C_\sigma^{\sum_{\ell'=0}^{j-1} r^{\ell'}}\right\}.$$

When $\ell = 0$, $\mathbb{E}_{\mathbf{x}} \left[|x_i|^2\right] = 1$ for all $i \in [d_0]$ by Assumption 4.1.

Suppose for all $\ell \leq q-1$, $\mathbb{E}_{\mathbf{x}} \mathbb{E}_{\mathbf{w}_0} [|f_i^{(\ell)}|^2] \geq \min\left(1, \min_{1 \leq j \leq q} C_\sigma^{\sum_{\ell'=0}^{j-1} r^{\ell'}}\right)$. When $\ell = q$,

$$\mathbb{E}_{\mathbf{x}} \mathbb{E}_{\mathbf{w}_0} [|f_i^{(q)}|^2] = \mathbb{E}_{\mathbf{w}_0} \left[\left|\sigma_i^{(q)}\left(\frac{1}{\sqrt{m_i^{(\ell)}}} (\mathbf{w}_i^{(q)})^T f_{\mathcal{S}_i^{(q)}}\right)\right|^2\right] = \mathbb{E}_{\mathbf{x}} \mathbb{E}_{\mathbf{w}_0} \mathbb{E}_{z \sim \mathcal{N}(0,1)} \left[\left|\sigma_i^{(q)}\left(\frac{\|f_{\mathcal{S}_i^{(q)}}\|}{\sqrt{m_i^{(q)}}} z\right)\right|^2\right].$$

By Assumption 4.2,

$$\mathbb{E}_{\mathbf{x}} \mathbb{E}_{\mathbf{w}_0} \mathbb{E}_{z \sim \mathcal{N}(0,1)} \left[\left|\sigma_i^{(q)}\left(\frac{\|f_{\mathcal{S}_i^{(q)}}\|}{\sqrt{m_i^{(q)}}} z\right)\right|^2\right] = \mathbb{E}_{z \sim \mathcal{N}(0,1)} [|\sigma_i^{(q)}(z)|^2] \mathbb{E}_{\mathbf{x}} \mathbb{E}_{\mathbf{w}_0} \left[\left(\frac{\|f_{\mathcal{S}_i^{(q)}}\|^2}{m_i^{(q)}}\right)^r\right]$$

$$\geq C_\sigma \mathbb{E}_{\mathbf{x}} \mathbb{E}_{\mathbf{w}_0} \left[\left(\frac{\|f_{\mathcal{S}_i^{(q)}}\|^2}{m_i^{(q)}}\right)^r\right]$$

We use Jensen's inequality,

$$C_\sigma \mathbb{E}_{\mathbf{x}} \mathbb{E}_{\mathbf{w}_0} \left[\left(\frac{\|f_{\mathcal{S}_i^{(q)}}\|^2}{m_i^{(q)}}\right)^r\right] \geq C_\sigma \left(\frac{\mathbb{E}_{\mathbf{x}} \mathbb{E}_{\mathbf{w}_0} \left[\|f_{\mathcal{S}_i^{(q)}}\|^2\right]}{m_i^{(q)}}\right)^r$$

Then according to inductive assumption, we have

$$C_\sigma \left(\frac{\mathbb{E}_{\mathbf{x}} \mathbb{E}_{\mathbf{w}_0} \left[\|f_{\mathcal{S}_i^{(q)}}\|^2\right]}{m_i^{(q)}}\right)^r \geq C_\sigma \left(\min\left(1, \min_{1 \leq j \leq q} C_\sigma^{\sum_{\ell'=0}^{j-1} r^{\ell'}}\right)\right)^r$$

$$\geq \min_{1 \leq j \leq q+1} C_\sigma^{\sum_{\ell'=0}^{j-1} r^{\ell'}}.$$

Hence for all $l \leq q$, $\mathbb{E}_{\mathbf{x}} \mathbb{E}_{\mathbf{w}_0} [|f_i^{(\ell)}|^2] \geq \min\left(1, \min_{1 \leq j \leq q+1} C_\sigma^{\sum_{\ell'=0}^{j-1} r^{\ell'}}\right)$, which finishes the inductive step hence the proof.

Therefore,

$$\mathbb{E}_{\mathbf{x}, \mathbf{w}_0} \left[\|\nabla_{\mathbf{w}^{(\ell)}} f_k(\mathbf{w}_0)\|\right] = \mathbb{E}_{\mathbf{x}, \mathbf{w}_0} \left[\frac{1}{\sqrt{m_k^{(\ell)}}} \left\|f_{\mathcal{S}_k^{(\ell)}}\right\|\right] \geq \sqrt{\min\left(1, \min_{1 \leq j \leq L} C_\sigma^{\sum_{\ell'=0}^{j-1} r^{\ell'}}\right)} = \Omega(1).$$

# H  Proof of Lemma B.1

We prove the result by induction.

For the base case when $\ell = \ell' + 1$,

$$\left\| \frac{\partial f_{S_j^{(\ell)}}}{\partial \mathbf{w}^{(\ell-1)}} \right\| = \left\| \frac{\partial f^{(\ell-1)}}{\partial \mathbf{w}^{(\ell-1)}} \frac{\partial f_{S_j^{(\ell)}}}{\partial f^{(\ell-1)}} \right\|$$

$$\leq \max_{i:f_i^{(\ell-1)} \in \mathcal{F}_{S_j^{(\ell)}}} \frac{1}{\sqrt{m_i^{(\ell-1)}}} \left| \sigma'(\tilde{f}_i^{(\ell-1)}) \right| \left\| f_{S_i^{(\ell-1)}} \right\|$$

$$\leq \max_{i:f_i^{(\ell-1)} \in \mathcal{F}_{S_j^{(\ell)}}} \frac{\gamma_1}{\sqrt{m_i^{(\ell-1)}}} \left\| f_{S_i^{(\ell-1)}} \right\|.$$

and

$$\left\| \frac{\partial f_{S_j^{(\ell)}}}{\partial \mathbf{w}^{(\ell-1)}} \right\|_F = \sqrt{\sum_{i:f_i^{(\ell-1)} \in \mathcal{F}_{S_j^{(\ell)}}} \left\| \frac{\partial f_i^{(\ell-1)}}{\partial \mathbf{w}^{(\ell-1)}} \right\|^2}$$

$$\leq \sqrt{m_j^{(\ell)}} \max_{i:f_i^{(\ell-1)} \in \mathcal{F}_{S_j^{(\ell)}}} \frac{1}{\sqrt{m_i^{(\ell-1)}}} \left| \sigma'(\tilde{f}_i^{(\ell-1)}) \right| \left\| f_{S_i^{(\ell-1)}} \right\|$$

$$\leq \sqrt{m_j^{(\ell)}} \max_{i:f_i^{(\ell-1)} \in \mathcal{F}_{S_j^{(\ell)}}} \frac{\gamma_1}{\sqrt{m_i^{(\ell-1)}}} \left\| f_{S_i^{(\ell-1)}} \right\|.$$

By Lemma K.3, with probability at least $1 - m_i^{(\ell-1)} \exp(-C_{\ell-1}^{\mathcal{P}} \log^2 m)$, $\left\| f_{S_i^{(\ell-1)}} \right\| = O\left( (\log m + R)^{\ell-2} \sqrt{m_i^{(\ell-1)}} \right) = \tilde{O}\left( R^{\ell-2} \sqrt{m_i^{(\ell-1)}} \right)$.

For the maximum norm $\max_i \left\| f_{S_i^{(\ell-1)}} \right\| / \sqrt{m_i^{(\ell-1)}}$, we apply union bound over the indices $i$ such that $f_i^{(\ell-1)} \in \mathcal{F}_{S_j^{(\ell)}}$, the cardinality of which is at most $\left| \mathcal{F}_{S_j^{(\ell)}} \right| = m_j^{(\ell)}$. Hence with probability at least $1 - m_i^{(\ell-1)} m_j^{(\ell)} \exp(-C_{\ell-1}^{\mathcal{P}} \log^2 m)$,

$$\max_i \left\| f_{S_i^{(\ell-1)}} \right\| / \sqrt{m_i^{(\ell-1)}} = O\left( (\log m + R)^{\ell-2} \right) = \tilde{O}(R^{\ell-2}).$$

Since $m_i^{(\ell-1)} \leq \overline{m}_{\ell-1}$ and $m_j^{(\ell)} \leq \overline{m}_\ell$ where $\overline{m}_{\ell-1}, \overline{m}_\ell$ are polynomial in $m$, we can find a constant $C_{\ell,\ell-1}^f > 0$ such that $\exp(-C_{\ell,\ell-1}^f \log^2 m) \geq \exp(-C_{\ell-1}^{\mathcal{P}} \log^2 m) \cdot \exp(\log(\overline{m}_{\ell-1} \cdot \overline{m}_\ell))$. As a result, with probability at least $1 - \exp(-C_{\ell,\ell-1}^f \log^2 m)$,

$$\left\| \frac{\partial f_{S_j^{(\ell)}}}{\partial \mathbf{w}^{(\ell-1)}} \right\| = O\left( (\log m + R)^{\ell-1} \right) = \tilde{O}(R^{\ell-1}),$$

$$\left\| \frac{\partial f_{S_j^{(\ell)}}}{\partial \mathbf{w}^{(\ell-1)}} \right\|_F = O\left( \sqrt{m_j^{(\ell)}} (\log m + R)^{\ell-1} \right) = \tilde{O}\left( \sqrt{m_j^{(\ell)}} R^{\ell-1} \right).$$

Supposing $\ell \leq k$, Eq. (26) and (27) hold with probability at least $1 - \exp(-C_{k,\ell'}^f \log^2 m)$.

For $\ell = k + 1$, since elements of $f_{S_j^{(k+1)}}$ are from $\mathcal{P}^{(k)}$ where only $f^{(\ell')}, ..., f^{(k)}$ possibly depend on $\mathbf{w}^{(\ell')}$, we have

$$\frac{\partial f_{S_j^{(k+1)}}}{\partial \mathbf{w}^{(\ell')}} = \sum_{q=\ell'+1}^{k} \sum_{i:f_i^{(q)} \in \mathcal{F}_{S_j^{(k+1)}}} \frac{\partial f_{S_i^{(q)}}}{\partial \mathbf{w}^{(\ell')}} \frac{\partial f_i^{(q)}}{\partial f_{S_i^{(q)}}} \frac{\partial f_{S_j^{(k+1)}}}{\partial f_i^{(q)}}. \tag{38}$$

With simple computation, we know that for any $i$ s.t. $f_i^{(q)} \in \mathcal{F}_{\mathcal{S}_j^{(k+1)}}$:

$$\frac{\partial f_i^{(q)}}{\partial f_{\mathcal{S}_i^{(q)}}} \frac{\partial f_{\mathcal{S}_j^{(k+1)}}}{\partial f_i^{(q)}} = \frac{1}{\sqrt{m_i^{(q)}}} \sigma'(\tilde{f}_i^{(q)}) \mathbf{w}_i^{(q)} \frac{\partial f_{\mathcal{S}_j^{(k+1)}}}{\partial f_i^{(q)}},$$

where $\dfrac{\partial f_{\mathcal{S}_j^{(k+1)}}}{\partial f_i^{(q)}}$ is a mask matrix defined in Eq. (21).

Supposing $\partial f_{\mathcal{S}_i^{(q)}}/\partial \mathbf{w}^{(\ell')}$, $i \in [d_q]$ in Eq. (38) is fixed, for each $q$, we apply Lemma K.6 to bound the spectral norm. Choosing $t = \sqrt{m_j^{(k+1)}} \log m$, with probability at least $1 - 2\exp(-m_j^{(k+1)} \log^2 m)$, for some absolute constant $C > 0$,

$$\left\| \sum_{i:f_i^{(q)} \in \mathcal{F}_{\mathcal{S}_j^{(k+1)}}} \frac{\partial f_{\mathcal{S}_i^{(q)}}}{\partial \mathbf{w}^{(\ell')}} \frac{\partial f_i^{(q)}}{\partial f_{\mathcal{S}_i^{(q-1)}}} \frac{\partial f_{\mathcal{S}_j^{(k+1)}}}{\partial f_i^{(q)}} \right\|$$

$$\leq C\gamma_1 \left( \max_i \frac{1}{\sqrt{m_i^{(q)}}} \left\| \frac{\partial f_{\mathcal{S}_i^{(q)}}}{\partial \mathbf{w}^{(\ell')}} \right\| \left( \sqrt{m_j^{(k+1)}} + \sqrt{m_j^{(k+1)}} \log m + R \right) + \max_i \frac{1}{\sqrt{m_i^{(q)}}} \left\| \frac{\partial f_{\mathcal{S}_i^{(q)}}}{\partial \mathbf{w}^{(\ell')}} \right\|_F \right).$$

$$\tag{39}$$

To bound the Frobenious norm of Eq. (38) for each $q$, we apply Lemma K.7 and choose $t = \left\| \partial f_{\mathcal{S}_i^{(q)}}/\partial \mathbf{w}^{(\ell')} \right\| \log m$. By union bound over indices $i$ such that $f_i^{(q)} \in \mathcal{F}_{\mathcal{S}_j^{(k+1)}}$, then with probability at least $1 - 2m_j^{(k+1)} \exp(-c' \log^2 m)$, where $c' > 0$ is a constant, we have

$$\left\| \sum_{i:f_i^{(q)} \in \mathcal{F}_{\mathcal{S}_j^{(k+1)}}} \frac{\partial f_{\mathcal{S}_i^{(q)}}}{\partial \mathbf{w}^{(\ell')}} \frac{\partial f_i^{(q)}}{\partial f_{\mathcal{S}_i^{(q)}}} \frac{\partial f_{\mathcal{S}_j^{(k+1)}}}{\partial f_i^{(q)}} \right\|_F$$

$$= \sqrt{\sum_{i:f_i^{(q)} \in \mathcal{F}_{\mathcal{S}_j^{(k+1)}}} \left\| \frac{\partial f_{\mathcal{S}_i^{(q)}}}{\partial \mathbf{w}^{(\ell')}} \frac{\partial f_i^{(q)}}{\partial f_{\mathcal{S}_i^{(q)}}} \right\|^2}$$

$$\leq \sqrt{m_j^{(k+1)}} \max_i \left\| \frac{\partial f_{\mathcal{S}_i^{(q)}}}{\partial \mathbf{w}^{(\ell')}} \frac{1}{\sqrt{m_i^{(q)}}} \left( \mathbf{w}_i^{(q)} \right)^T \sigma'(\tilde{f}_i^{(q)}) \right\|$$

$$\leq \gamma_1 \sqrt{m_j^{(k+1)}} \max_i \frac{1}{\sqrt{m_i^{(q)}}} \left( \left\| \frac{\partial f_{\mathcal{S}_i^{(q)}}}{\partial \mathbf{w}^{(\ell')}} \right\| (\log m + R) + \left\| \frac{\partial f_{\mathcal{S}_i^{(q)}}}{\partial \mathbf{w}^{(\ell')}} \right\|_F \right). \tag{40}$$

To bound the maximum of $\left\| \partial f_{\mathcal{S}_i^{(q)}}/\partial \mathbf{w}^{(\ell')} \right\| / \sqrt{m_i^{(q)}}$ and $\left\| \partial f_{\mathcal{S}_i^{(q)}}/\partial \mathbf{w}^{(\ell')} \right\|_F / \sqrt{m_i^{(q)}}$ that appear in Eq. (39) and (40), with the induction hypothesis, we apply union bound over indices $i$ such that $f_i^{(q)} \in \mathcal{F}_{\mathcal{S}_j^{(k+1)}}$. Therefore, with probability at least $1 - m_j^{(k+1)} \exp(-C_{q,\ell'}^f \log^2 m)$,

$$\max_i \frac{1}{\sqrt{m_i^{(q)}}} \left\| \frac{\partial f_{\mathcal{S}_i^{(q)}}}{\partial \mathbf{w}^{(\ell')}} \right\| = O\left( \max_{\ell'+1 \leq p \leq q} \frac{1}{\sqrt{\overline{m}_p}} (\log m + R)^{\ell'} \right) = \tilde{O}\left( \max_{\ell'+1 \leq p \leq q} \frac{R^{\ell'}}{\sqrt{\overline{m}_p}} \right),$$

$$\max_i \frac{1}{\sqrt{m_i^{(q)}}} \left\| \frac{\partial f_{\mathcal{S}_i^{(q)}}}{\partial \mathbf{w}^{(\ell')}} \right\|_F = O\left( (\log m + R)^{q-1} \right) = \tilde{O}\left( R^{q-1} \right).$$

Putting them in Eq. (39) and (40), we have

$$
\left\| \sum_{i:f_i^{(q)} \in \mathcal{F}_{\mathcal{S}_j^{(k+1)}}} \frac{\partial f_{\mathcal{S}_i^{(q)}}}{\partial \mathbf{w}^{(\ell')}} \frac{\partial f_i^{(q)}}{\partial f_{\mathcal{S}_i^{(q)}}} \frac{\partial f_{\mathcal{S}_j^{(k+1)}}}{\partial f_i^{(q)}} \right\| = \tilde{O}\left( \max\left( \left( \max_{\ell'+1 \leq p \leq q} \frac{\sqrt{m_j^{(k+1)}}}{\sqrt{\underline{m}_p}} \right), 1 \right) \right),
$$

$$
\left\| \sum_{i:f_i^{(q)} \in \mathcal{F}_{\mathcal{S}_j^{(k+1)}}} \frac{\partial f_{\mathcal{S}_i^{(q)}}}{\partial \mathbf{w}^{(\ell')}} \frac{\partial f_i^{(q)}}{\partial f_{\mathcal{S}_i^{(q)}}} \frac{\partial f_{\mathcal{S}_j^{(k+1)}}}{\partial f_i^{(q)}} \right\|_F = \tilde{O}\left( \sqrt{m_j^{(k+1)}} \right),
$$

with probability at least $1 - 2\exp(-m_j^{(k+1)} \log^2 m) - m_j^{(k+1)} \exp(-C_{q,\ell'}^f \log^2 m) - 2m_j^{(k+1)} \exp(-c' \log^2 m)$.

As the current result is for fixed $q$, applying the union bound over indices $q \in \{\ell'+1, ..., k\}$, we have with probability at least $1 - 2(k-\ell')\exp(-m_j^{(k+1)}) - \sum_q m_j^{(k+1)} \exp(-C_{q,\ell'}^f \log^2 m) - 2m_j^{(k+1)} \exp(-c' \log^2 m)$,

$$
\left\| \sum_{q=\ell'+1}^{k} \sum_{i:f_i^{(q)} \in \mathcal{F}_{\mathcal{S}_j^{(k+1)}}} \frac{\partial f_{\mathcal{S}_i^{(q)}}}{\partial \mathbf{w}^{(\ell')}} \frac{\partial f_i^{(q)}}{\partial f_{\mathcal{S}_i^{(q)}}} \frac{\partial f_{\mathcal{S}_j^{(k+1)}}}{\partial f_i^{(q)}} \right\| = O\left( \max\left( \max_{\ell'+1 \leq p \leq k} \frac{\sqrt{m_j^{(k+1)}}}{\sqrt{\underline{m}_p}}, 1 \right) (\log m + R)^{\ell'} \right)
$$

$$
= \tilde{O}\left( \max_{\ell'+1 \leq p \leq k+1} \frac{\sqrt{m_j^{(k+1)}}}{\sqrt{\underline{m}_p}} R^{\ell'} \right),
$$

$$
\left\| \sum_{q=\ell'+1}^{k} \sum_{i:f_i^{(q)} \in \mathcal{F}_{\mathcal{S}_j^{(k+1)}}} \frac{\partial f_{\mathcal{S}_i^{(q)}}}{\partial \mathbf{w}^{(\ell')}} \frac{\partial f_i^{(q)}}{\partial f_{\mathcal{S}_i^{(q)}}} \frac{\partial f_{\mathcal{S}_j^{(k+1)}}}{\partial f_i^{(q)}} \right\|_F = O\left( \sqrt{m_j^{(k+1)}} (\log m + R)^k \right) = \tilde{O}\left( \sqrt{m_j^{(k+1)}} R^k \right).
$$

Since $m_j^{(k+1)}$ is upper bounded by $\overline{m}_{k+1}$ which is polynomial in $m$, we can find a constant $C_{k+1,\ell'}^f > 0$ such that for each $j$, the result holds with probability at least $1 - \exp(-C_{k+1,\ell'}^f \log^2 m)$ for $\ell \leq k+1$. Then we finish the inductive step which completes the proof.

# I  Proof of Lemma B.2

Before the proof, by Assumption 3.5, we have the following proposition which is critical in the tail bound of the norm of the matrix Gaussian series, i.e., Lemma K.8. In the bound, there will be a dimension factor which is the number of parameters (see Eq. (50)). Note that the number of parameters at each layer can be exponentially large w.r.t. the width $m$. If we naively apply the bound, the bound will be useless. However, each neuron in fact only depends on polynomial in $m$ number of parameters, which is the dimension factor we should use.

**Proposition I.1.** *Fixed $\ell' \in [L]$, we denote the maximum number of elements in $\mathbf{w}^{(\ell')}$ that $f_i^{(\ell)}$ depends on for all $\ell \in [L], i \in [d_\ell]$ by $m_{\ell'}^*$, which is polynomial in $m$.*

The proof the proposition can be found in Appendix J.

Now we start the proof of the lemma. In fact, we will prove a more general result which includes the neurons in output layer, i.e. $\ell$-th layer. And we will use the result of Lemma B.1 in the proof. Specifically, we will prove the following lemma:

**Lemma I.2.** *Given $1 \leq \ell_1 \leq \ell_2 \leq L$, for any $\ell_2 \leq \ell \leq L$, $\mathbf{w} \in \mathsf{B}(\mathbf{w}_0, R)$, and $j \in [d_\ell]$, we have, with probability at least $1 - \exp(-\Omega(\log^2 m))$,*

$$
\left\| \frac{\partial^2 \tilde{f}_j^{(\ell)}}{\partial \mathbf{w}^{(\ell_1)} \partial \mathbf{w}^{(\ell_2)}} \right\| = O\left( \max_{\ell_1+1 \leq p \leq \ell} \frac{1}{\sqrt{\underline{m}_p}} (\log m + R)^{\ell^2} \right) = \tilde{O}\left( \max_{\ell_1+1 \leq p \leq \ell} \frac{R^{\ell^2}}{\sqrt{\underline{m}_p}} \right). \tag{41}
$$

We will prove the results by induction.

For the base case that $\ell = \ell_2$,

$$\left\| \frac{\partial^2 \tilde{f}_j^{(\ell_2)}}{\partial \mathbf{w}^{(\ell_1)} \partial \mathbf{w}^{(\ell_2)}} \right\| = \left\| \frac{1}{\sqrt{m_j^{(\ell_2)}}} \frac{\partial f_{S_j^{(\ell_2)}}}{\partial \mathbf{w}^{(\ell_1)}} \right\|.$$

By Lemma B.1, we can find a constant $M_{\ell_1,\ell_2}^{(\ell_2),j} > 0$ such that with probability at least $1 - \exp\left(-M_{\ell_1,\ell_2}^{(\ell_2),j} \log^2 m\right)$,

$$\left\| \frac{1}{\sqrt{m_j^{(\ell_2)}}} \frac{\partial f_{S_j^{(\ell_2)}}}{\partial \mathbf{w}^{(\ell_1)}} \right\| = O\left( \max_{\ell_1+1 \leq p \leq \ell_2} \frac{1}{\sqrt{m}_p} (\log m + R)^{\ell_1} \right).$$

Suppose for $\ell_2 \leq \ell' \leq \ell$, with probability at least $1 - \exp\left(-M_{\ell_1,\ell_2}^{(\ell),j} \log^2 m\right)$ for some constant $M_{\ell_1,\ell_2}^{(\ell),j} > 0$, Eq. (28) holds.

When $\ell' = \ell + 1$, we have

$$\left\| \frac{\partial^2 \tilde{f}_j^{(\ell+1)}}{\partial \mathbf{w}^{(\ell_1)} \partial \mathbf{w}^{(\ell_2)}} \right\| = \left\| \sum_{\ell'=\ell_2}^{\ell} \sum_{i=1}^{d_{\ell'}} \frac{\partial^2 f_i^{(\ell')}}{\partial \mathbf{w}^{(\ell_1)} \partial \mathbf{w}^{(\ell_2)}} \frac{\partial \tilde{f}_j^{(\ell+1)}}{\partial f_i^{(\ell')}} \right\| \leq \sum_{\ell'=\ell_2}^{\ell} \left\| \sum_{i=1}^{d_{\ell'}} \frac{\partial^2 f_i^{(\ell')}}{\partial \mathbf{w}^{(\ell_1)} \partial \mathbf{w}^{(\ell_2)}} \frac{\partial \tilde{f}_j^{(\ell+1)}}{\partial f_i^{(\ell')}} \right\|.$$

$$(42)$$

We will bound every term in the above summation. For each term, by definition,

$$\sum_{i=1}^{d_{\ell'}} \frac{\partial^2 f_i^{(\ell')}}{\partial \mathbf{w}^{(\ell_1)} \partial \mathbf{w}^{(\ell_2)}} \frac{\partial \tilde{f}_j^{(\ell+1)}}{\partial f_i^{(\ell')}} = \sum_{i=1}^{d_{\ell'}} \frac{\partial^2 \tilde{f}_i^{(\ell')}}{\partial \mathbf{w}^{(\ell_1)} \partial \mathbf{w}^{(\ell_2)}} \frac{\partial \tilde{f}_j^{(\ell+1)}}{\partial \tilde{f}_i^{(\ell')}}$$

$$+ \frac{1}{\sqrt{m_j^{(\ell+1)}}} \sum_{i:f_i^{(\ell')} \in \mathcal{F}_{S_j^{(\ell+1)}}} (\mathbf{w}_j^{(\ell+1)})_{\mathrm{id}_{\ell',i}^{\ell+1,j}} \sigma''(\tilde{f}_i^{(\ell')}) \frac{\partial \tilde{f}_i^{(\ell')}}{\partial \mathbf{w}^{(\ell_1)}} \left( \frac{\partial \tilde{f}_i^{(\ell')}}{\partial \mathbf{w}^{(\ell_2)}} \right)^T.$$

$$(43)$$

For the first term in Eq. (43), we use Lemma K.10. Specifically, we view $U_i = \frac{\partial^2 \tilde{f}_i^{(\ell')}}{\partial \mathbf{w}^{(\ell_1)} \partial \mathbf{w}^{(\ell_2)}}$, hence with probability at least $1 - \sum_{k=1}^{\ell-\ell'+1} k(m_{\ell_1}^* + m_{\ell_2}^*) \exp(-\log^2 m/2)$,

$$\left\| \sum_{i=1}^{d_{\ell'}} \frac{\partial^2 \tilde{f}_i^{(\ell')}}{\partial \mathbf{w}^{(\ell_1)} \partial \mathbf{w}^{(\ell_2)}} \frac{\partial \tilde{f}_j^{(\ell+1)}}{\partial \tilde{f}_i^{(\ell')}} \right\| = O\left( \max_{i:f_i^{(\ell')} \in \mathcal{F}_{S_j^{(\ell'+1)}}} \left\| \frac{\partial^2 \tilde{f}_i^{(\ell')}}{\partial \mathbf{w}^{(\ell_1)} \partial \mathbf{w}^{(\ell_2)}} \right\| (\log m + R)^{\ell-\ell'+1} \right).$$

Here we'd like to note that from Lemma K.8, the tail bound depends on the dimension of $\mathbf{w}^{(\ell_1)}$ and $\mathbf{w}^{(\ell_2)}$ which are $\sum_{i=1}^{d_{\ell_1}} m_i^{(\ell)}$ and $\sum_{i=1}^{d_{\ell_2}} m_i^{(\ell)}$ respectively. By Assumption 3.5, for any $\ell$, $m_i^{(\ell)}$ is polynomial in $m$. Therefore, the number of elements in $\mathbf{w}^{(\ell)}$ that $f_j^{(\ell+1)}$ depends on is polynomial in $m$ by Proposition I.1. And the matrix variance $\tilde{\nu}^{(\ell')}$ in Lemma K.10 is equivalent to the matrix variance that we only consider the elements in $\mathbf{w}^{(\ell_1)}$ and $\mathbf{w}^{(\ell_2)}$ that $f_j^{(\ell+1)}$ depends on, in which case the dimension is polynomial in $m$. Therefore we can use $m_\ell^*$ here. It is the same in the following when we apply matrix Gaussian series tail bound.

Then we apply union bound over indices $i$ such that $f_i^{(\ell')} \in \mathcal{F}_{S_j^{(\ell'+1)}}$, whose cardinality is at most $m_j^{(\ell+1)}$. By the inductive hypothesis, with probability at least $1 - \sum_{k=1}^{\ell-\ell'+1} k(m_{\ell_1}^* +$

$$m^*_{\ell_2}) \exp(-\log^2 m/2) - m^{(\ell+1)}_j \exp\left(-M^{(\ell),j}_{\ell_1,\ell_2} \log^2 m\right),$$

$$\left\| \sum_{i=1}^{d_{\ell'}} \frac{\partial^2 \tilde{f}^{(\ell')}_i}{\partial \mathbf{w}^{(\ell_1)} \partial \mathbf{w}^{(\ell_2)}} \frac{\partial \tilde{f}^{(\ell+1)}_j}{\partial \tilde{f}^{(\ell')}_i} \right\| = O\left( \max_{\ell_1+1\le p\le \ell'} \frac{1}{\sqrt{\underline{m}_p}} (\log m + R)^{(\ell')^2+\ell-\ell'+1} \right).$$

For the second term in Eq. (43), we view it as a matrix Gaussian series with respect to $\mathbf{w}^{(\ell+1)}_j$. The matrix variance takes the form

$$\nu^{(\ell'),j}_{\ell_1,\ell_2} = \frac{1}{m^{(\ell+1)}_j} \max$$

$$\left\{ \left\| \sum_{i:f^{(\ell')}_i \in \mathcal{F}_{\mathcal{S}^{(\ell+1)}_j}} \left(\sigma''(\tilde{f}^{(\ell')}_i)\right)^2 \left\| \frac{\partial \tilde{f}^{(\ell')}_i}{\partial \mathbf{w}^{(\ell_1)}} \right\|^2 \frac{\partial \tilde{f}^{(\ell')}_i}{\partial \mathbf{w}^{(\ell_2)}} \left(\frac{\partial \tilde{f}^{(\ell')}_i}{\partial \mathbf{w}^{(\ell_2)}}\right)^T \right\|, \right.$$

$$\left. \left\| \sum_{i:f^{(\ell')}_i \in \mathcal{F}_{\mathcal{S}^{(\ell+1)}_j}} \left(\sigma''(\tilde{f}^{(\ell')}_i)\right)^2 \left\| \frac{\partial \tilde{f}^{(\ell')}_i}{\partial \mathbf{w}^{(\ell_2)}} \right\|^2 \frac{\partial \tilde{f}^{(\ell')}_i}{\partial \mathbf{w}^{(\ell_1)}} \left(\frac{\partial \tilde{f}^{(\ell')}_i}{\partial \mathbf{w}^{(\ell_1)}}\right)^T \right\| \right\}.$$

We use Lemma K.9. By the definition in Eq. (51), here $\nu^{(\ell'),j}_{\ell_1,\ell_2} = \max\left\{ \mu^{(\ell'),j}_{\ell_1,\ell_2}, \mu^{(\ell'),j}_{\ell_2,\ell_1} \right\}$. Hence with probability at least at least $1 - \exp\left(-C^{(\ell'),j}_{\ell_1,\ell_2} \log^2 m\right) - \exp\left(-C^{(\ell'),j}_{\ell_2,\ell_1} \log^2 m\right)$ for some constant $C^{(\ell'),j}_{\ell_1,\ell_2}, C^{(\ell'),j}_{\ell_2,\ell_1} > 0$, we have

$$\nu^{(\ell'),j}_{\ell_1,\ell_2} = O\left( \max\left(1/m^{(\ell+1)}_j, \max_{\ell_1+1\le p\le \ell} 1/\underline{m}_p\right) (\log m + R)^{4\ell'-2} \right).$$

Using Lemma K.8 again and choosing $t = \log m \sqrt{\nu^{(\ell'),j}_{\ell_1,\ell_2}}$, we have with probability at least $1 - (m^*_{\ell_2} + m^*_{\ell_1}) \exp(-\log^2 m/2)$,

$$\left\| \frac{1}{\sqrt{m^{(\ell+1)}_j}} \sum_{i:f^{(\ell')}_i \in \mathcal{F}_{\mathcal{S}^{(\ell+1)}_j}} \left(\mathbf{w}^{(\ell+1)}_j\right)_{\text{id}^{\ell+1,j}_{\ell',i}} \sigma''\left(\tilde{f}^{(\ell')}_i\right) \frac{\partial \tilde{f}^{(\ell')}_i}{\partial \mathbf{w}^{(\ell_1)}} \left(\frac{\partial \tilde{f}^{(\ell')}_i}{\partial \mathbf{w}^{(\ell_2)}}\right)^T \right\| \le (\log m + R) \sqrt{\nu^{(\ell'),j}_{\ell_1,\ell_2}}.$$

Combined the bound on $\nu^{(\ell'),j}_{\ell_1,\ell_2}$, we have with probability at least $1 - \exp\left(-C^{(\ell'),j}_{\ell_1,\ell_2} \log^2 m\right) - \exp\left(-C^{(\ell'),j}_{\ell_2,\ell_1} \log^2 m\right) - (m^*_{\ell_2} + m^*_{\ell_1}) \exp(-\log^2 m/2)$,

$$\left\| \frac{1}{\sqrt{m^{(\ell+1)}_j}} \sum_{i:f^{(\ell')}_i \in \mathcal{F}_{\mathcal{S}^{(\ell+1)}_j}} \left(\mathbf{w}^{(\ell+1)}_j\right)_{\text{id}^{\ell+1,j}_{\ell',i}} \sigma''\left(\tilde{f}^{(\ell')}_i\right) \frac{\partial \tilde{f}^{(\ell')}_i}{\partial \mathbf{w}^{(\ell_1)}} \left(\frac{\partial \tilde{f}^{(\ell')}_i}{\partial \mathbf{w}^{(\ell_2)}}\right)^T \right\|$$

$$= O\left( \max\left(1/\sqrt{m^{(\ell_2+1)}_j}, \max_{\ell_1+1\le p\le \ell} 1/\sqrt{\underline{m}_p}\right) (\log m + R)^{2\ell'} \right)$$

$$= \tilde{O}\left( \max\left(1/\sqrt{m^{(\ell_2+1)}_j}, \max_{\ell_1+1\le p\le \ell} 1/\sqrt{\underline{m}_p}\right) R^{2\ell'} \right)$$

$$= \tilde{O}\left( \max_{\ell_1+1\le p\le \ell+1} R^{2\ell'}/\sqrt{\underline{m}_p} \right).$$

Now we have bound both terms in Eq. (43). Combining the bounds, we have with probability at least $1 - \sum_{k=1}^{\ell-\ell'+1} k(m_{\ell_1}^* + m_{\ell_2}^*) \exp(-\log^2 m/2) - m_j^{(\ell+1)} \exp\left(-M_{\ell_1,\ell_2}^{(\ell),j} \log^2 m\right) - \exp\left(-C_{\ell_1,\ell_2}^{(\ell'),j} \log^2 m\right) - \exp\left(-C_{\ell_2,\ell_1}^{(\ell'),j} \log^2 m\right) - 2(m_{\ell_2}^* + m_{\ell_1}^*) \exp(-\log^2 m/2)$

$$\left\| \sum_{i=1}^{d_{\ell'}} \frac{\partial^2 f_i^{(\ell')}}{\partial \mathbf{w}^{(\ell_1)} \partial \mathbf{w}^{(\ell_2)}} \frac{\partial \tilde{f}_j^{(\ell+1)}}{\partial f_i^{(\ell')}} \right\| = O\left( \max_{\ell_1+1 \leq p \leq \ell+1} 1/\sqrt{m_p}(\log m + R)^{(\ell'+1)^2 + \ell - \ell'} \right).$$

With the above results, to bound Eq. (42), we apply the union bound over the layer indices $l' = \ell_2, ..., \ell$. We have with probability at least $1 - \sum_{\ell'=\ell_2}^{l} \sum_{k=1}^{\ell-\ell'+1} k(m_{\ell_1}^* + m_{\ell_2}^*) \exp(-\log^2 m/2) - (\ell - \ell_2 + 1) m_j^{(\ell+1)} \exp\left(-M_{\ell_1,\ell_2}^{(\ell),j} \log^2 m\right) - \sum_{\ell'=\ell_2}^{l} \exp\left(-C_{\ell_1,\ell_2}^{(\ell'),j} \log^2 m\right) - \sum_{\ell'=\ell_2}^{l} \exp\left(-C_{\ell_2,\ell_1}^{(\ell'),j} \log^2 m\right) - 2(\ell - \ell_2 + 1)(m_{\ell_2}^* + m_{\ell_1}^*) \exp(-\log^2 m/2)$

$$\left\| \frac{\partial^2 \tilde{f}_j^{(\ell+1)}}{\partial \mathbf{w}^{(\ell_1)} \partial \mathbf{w}^{(\ell_2)}} \right\| \leq \sum_{\ell'=\ell_2}^{\ell} \left\| \sum_{i=1}^{d_{\ell'}} \frac{\partial^2 f_i^{(\ell')}}{\partial \mathbf{w}^{(\ell_1)} \partial \mathbf{w}^{(\ell_2)}} \frac{\partial \tilde{f}_j^{(\ell+1)}}{\partial f_i^{(\ell')}} \right\|$$

$$= O\left( \max_{\ell_1+1 \leq p \leq \ell+1} 1/\sqrt{m_p}(\log m + R)^{(\ell+1)^2} \right)$$

$$= \tilde{O}\left( \max_{\ell_1+1 \leq p \leq \ell+1} R^{(\ell+1)^2}/\sqrt{m_p} \right).$$

By Proposition I.1, $m_{\ell_1}^*, m_{\ell_2}^*$ are also polynomial in $m$. Hence, we can find a constant $M_{\ell_1,\ell_2}^{(\ell+1),j} > 0$ such that

$$\exp\left(-M_{\ell_1,\ell_2}^{(\ell+1),j} \log^2 m\right)$$

$$> \sum_{\ell'=\ell_2}^{\ell} \sum_{k=1}^{\ell-\ell'+1} k(m_{\ell_1}^* + m_{\ell_2}^*) \exp(-\log^2 m/2) - (\ell - \ell_2 + 1) m_j^{(\ell+1)} \exp\left(-M_{\ell_1,\ell_2}^{(\ell),j} \log^2 m\right)$$

$$- \sum_{\ell'=\ell_2}^{\ell} \exp\left(-C_{\ell_1,\ell_2}^{(\ell'),j} \log^2 m\right) - \sum_{\ell'=\ell_2}^{\ell} \exp\left(-C_{\ell_2,\ell_1}^{(\ell'),j} \log^2 m\right) - 2(\ell - \ell_2 + 1)(m_{\ell_2}^* + m_{\ell_1}^*) \exp(-\log^2 m/2)$$

$$+ \exp\left(-M_{\ell_1,\ell_2}^{(\ell),j} \log^2 m\right).$$

Then Eq. (43) holds with probability at least $1 - \exp\left(-M_{\ell_1,\ell_2}^{(\ell+1),j} \log^2 m\right)$ for any $\ell_2 \leq \ell + 1 \leq L$, $j \in [d_{\ell+1}]$, which finishes the induction step hence completes the proof.

## J Proof of Proposition I.1

Fixing $\ell' \in [L]$, for any $\ell \in \{\ell', ..., L\}$, $i \in [d_\ell]$, we first show $f_i^{(\ell)}$ depends on polynomial number of elements in $\mathbf{w}^{(\ell')}$. We prove the result by induction.

If $\ell = \ell'$, then the number of elements in $\mathbf{w}^{(\ell)}$ that $f_i^{(\ell)}$ depend on is $m_i^{(\ell)}$.

Suppose $\ell' \leq \ell \leq k$ that $f_i^{(\ell)}$ depends on polynomial number of elements in $\mathbf{w}^{(\ell')}$. Then at $\ell = k+1$, we know

$$f_i^{(k+1)} = \sigma_i^{(k+1)} \left( \frac{1}{\sqrt{m_i^{(k+1)}}} \left\langle \mathbf{w}_i^{(k+1)}, f_{\mathcal{S}_i^{(k+1)}} \right\rangle \right).$$

As $f_{\mathcal{S}_i^{(k+1)}}$ contains $m_i^{(k+1)}$ neurons where each one depends on polynomial number of elements in $\mathbf{w}^{(\ell')}$ by the induction hypothesis. The composition of two polynomial functions is still polynomial, hence $f_i^{(k+1)}$ also depends on polynomial number of elements in $\mathbf{w}^{(\ell')}$.

The maximum number of elements in $\mathbf{w}^{(\ell')}$ that $f_i^{(\ell)}$ depends on among all layers $\ell$ is polynomial since it is the maximum of a finite sequence. By Assumption 3.5 that $\sup_{\ell \in \{2,...,L-1\}, i \in [d_\ell]} m_i^{(\ell)} = O(m^c)$, it is not hard to see that the maximum among all $i \in [d_\ell]$ is also polynomial.

# K    Useful Lemmas and their proofs

**Lemma K.1.** *Spectral norm of a matrix $H$ is upper bounded by the sum of the spectral norm of its blocks.*

*Proof.*

$$
\|H\| = \left\| \begin{pmatrix} H^{(1,1)} & 0 & \cdots & 0 \\ 0 & 0 & \cdots & 0 \\ \vdots & \vdots & \ddots & \vdots \\ 0 & 0 & \cdots & 0 \end{pmatrix} + \begin{pmatrix} 0 & H^{(1,2)} & \cdots & 0 \\ 0 & 0 & \cdots & 0 \\ \vdots & \vdots & \ddots & \vdots \\ 0 & 0 & \cdots & 0 \end{pmatrix} + \cdots + \begin{pmatrix} 0 & 0 & \cdots & 0 \\ 0 & 0 & \cdots & 0 \\ \vdots & \vdots & \ddots & \vdots \\ 0 & 0 & \cdots & H^{(L,L)} \end{pmatrix} \right\|
$$
$$
\leq \sum_{\ell_1,\ell_2} \|H^{(\ell_1,\ell_2)}\|.
$$

$\square$

**Lemma K.2.** *For $\ell = 0, 1, .., L$, with probability at least $1 - \exp(-C_\ell^{\mathcal{P}} \log^2 m)$ for some constant $C_\ell^{\mathcal{P}} > 0$, the absolute value of all neurons in $\mathcal{P}^{(\ell)}$ Eq. (20) is of the order $\tilde{O}(1)$ in the ball $\mathsf{B}(\mathbf{w}_0, R)$.*

*Proof.* We prove the result by induction.

When $\ell = 0$, $\mathcal{P}^{(0)} = f^{(0)} = \{x_1, ..., x_{d_0}\}$ therefore for all $i$, $|f_i^{(0)}| \leq C_{\boldsymbol{x}}$ surely by Assumption 3.1.

Suppose when $\ell = k$, with probability at least $1 - \exp(-C_k^{\mathcal{P}} \log^2 m)$, the absolute value of each neuron in $\mathcal{P}^{(k)}$ is of the order $O\left((\log m + R)^k\right)$ where $C_k^{\mathcal{P}} > 0$ is a constant. Then when $\ell = k + 1$, there will be new neurons $f^{(k+1)}$ added to $\mathcal{P}^{(k)}$, where each $f_i^{(k+1)}$ can be bounded by

$$
|f_i^{(k+1)}| = \left| \sigma \left( \frac{1}{\sqrt{m_i^{(k+1)}}} \left(\mathbf{w}_i^{(k+1)}\right)^T f_{\mathcal{S}_i^{(k+1)}} \right) \right|
$$
$$
\leq \frac{\gamma_1}{\sqrt{m_i^{(k+1)}}} \left(\mathbf{w}_i^{(k+1)}\right)^T f_{\mathcal{S}_i^{(k+1)}} + \sigma(0).
$$

By the union bound over all the elements in $f_{\mathcal{S}_i^{(k+1)}}$ which are in $\mathcal{P}^{(k)}$ and the induction hypothesis, with probability at least $1 - m_i^{(k+1)} \exp(-C_k^{\mathcal{P}} \log^2 m)$,

$$
\|f_{\mathcal{S}_i^{(k+1)}}\| = \sqrt{\sum_{j=1}^{m_i^{(k+1)}} \left(f_{\mathcal{S}_i^{(k+1)}}\right)_j} = O\left(\sqrt{m_i^{(k+1)}}(\log m + R)^k\right).
$$

By Lemma K.4, supposing $f_{\mathcal{S}_i^{(k+1)}}$ is fixed, choosing $t = \log m \left\|f_{\mathcal{S}_i^{(k+1)}}\right\|$, with probability at least $1 - 2\exp(-\log^2 m/2)$, in the ball $\mathsf{B}(\mathbf{w}_0, R)$,

$$
\left|(\mathbf{w}_i^{(k+1)})^T f_{\mathcal{S}_i^{(k+1)}}\right| \leq (\log m + R) \left\|f_{\mathcal{S}_i^{(k+1)}}\right\|.
$$

Combined with the bound on $\|f_{\mathcal{S}_i^{(k+1)}}\|$, with probability at least $1 - 2\exp(-\log^2 m/2) - m_i^{(k+1)} \exp(-C_k^{\mathcal{P}} \log^2 m)$,

$$
\left|f_i^{(k+1)}\right| \leq \frac{\gamma_1}{\sqrt{m_i^{(k+1)}}}(\log m + R) \left\|f_{\mathcal{S}_i^{(k+1)}}\right\| + \gamma_0 = O\left((\log m + R)^{k+1}\right) = \tilde{O}(R^{k+1}).
$$

Since $m_i^{(k+1)} \leq \overline{m}_{k+1}$ which is polynomial in $m$, we can find a constant $C_{k+1}^{\mathcal{P}} > 0$ such that for all $i$,

$$\exp(-C_{k+1}^{\mathcal{P}} \log^2 m) \geq 2 \exp(-\log^2 m/2) + \exp(-C_k^{\mathcal{P}} \log^2(m) + \log(\overline{m}_{k+1})) + \exp(-C_k^{\mathcal{P}} \log^2 m).$$

Hence the above results hold with probability $1 - \exp(-C_{k+1}^{\mathcal{P}} \log^2 m)$, which completes the proof. $\qquad\square$

**Lemma K.3.** *For $\ell \in [L], i \in [d_\ell]$, with probability at least $1 - m_i^{(\ell)} \exp(-C_\ell^{\mathcal{P}} \log^2 m)$, in the ball $\mathsf{B}(\mathbf{w}_0, R)$,*

$$\left\| f_{\mathcal{S}_i^{(\ell)}} \right\| = O\left( \sqrt{m_i^{(\ell)}} (\log m + R)^{\ell-1} \right) = \tilde{O}\left( \sqrt{m_i^{(\ell)}} R^{\ell-1} \right)$$

*Proof.* By Lemma K.2, each neuron is of order $\tilde{O}(1)$. Then we apply union bound over $m_i^{(\ell)}$ neurons and we get the result. $\qquad\square$

**Lemma K.4.** *Given a fixed vector $\boldsymbol{b} \in \mathbb{R}^n$ and a random vector $\mathbf{a}_0 \sim \mathcal{N}(0, I_n)$, for any $\mathbf{a}$ in the ball $\mathsf{B}(\mathbf{a}_0, R)$, we have with probability at least $1 - 2\exp(-t^2/(2\|\boldsymbol{b}\|^2))$,*

$$|\mathbf{a}^T \boldsymbol{b}| \leq t + \|\boldsymbol{b}\| R. \tag{44}$$

*Proof.* We can write $\mathbf{a}^T \boldsymbol{b} = (\mathbf{a}_0 + \Delta \mathbf{a})^T \boldsymbol{b} = \mathbf{a}_0^T \boldsymbol{b} + \Delta \mathbf{a}^T \boldsymbol{b}$. Since $\mathbf{a}_0 \sim \mathcal{N}(0, 1)$, we have $\mathbf{a}_0^T \boldsymbol{b} \sim \mathcal{N}(0, \|\mathbf{b}\|^2)$. By Proposition 2.5.2 in [23], for any $t > 0$, with probability at least $1 - 2\exp(-t^2/(2\|\boldsymbol{b}\|^2))$,

$$|\mathbf{a}_0^T \boldsymbol{b}| \leq t.$$

Therefore, with the same probability

$$|\mathbf{a}^T \boldsymbol{b}| \leq |\mathbf{a}_0^T \boldsymbol{b}| + |\Delta \mathbf{a}^T \boldsymbol{b}| \leq t + \|\boldsymbol{b}\| R.$$

$\qquad\square$

**Lemma K.5.** *For a random $m \times n$ matrix $W = [B_1 \boldsymbol{a}_1, B_2 \boldsymbol{a}_2, ..., B_n \boldsymbol{a}_n]$ where $A = [\boldsymbol{a}_1, \boldsymbol{a}_2, ..., \boldsymbol{a}_n]$ is an $N_i \times n$ random matrix whose entries i.i.d. follow $\mathcal{N}(0, 1)$ and $B_1, B_2, ..., B_n$ is a sequence of $m \times N_i$ non-random matrices, we have for some absolute constant $C > 0$, for any $t \geq 0$*

$$\|W\| \leq C\left( \max_i \|B_i\|(\sqrt{n} + t) + \max_i \|B_i\|_F \right) \tag{45}$$

*with probability at least $1 - 2\exp(-t^2)$.*

*Proof.* We prove the result using an $\epsilon$-net argument. Choosing $\epsilon = 1/4$, by Corollary 4.2.13 in [23], we can find an $\epsilon$-net $\mathcal{N}$ of the sphere $S^{n-1}$ with cardinalities $|\mathcal{N}| \leq 9^n$.

By Lemma 4.4.1 in [23], $\|W\| \leq 2 \sup_{\boldsymbol{x} \in \mathcal{N}} \|W\boldsymbol{x}\|$.

Fix $\boldsymbol{x} \in \mathcal{N}$, it is nor hard to see that

$$W\boldsymbol{x} = \sum_{i=1}^n x_i B_i \boldsymbol{a}_i \sim \mathcal{N}\left( 0, \sum_{i=1}^n x_i^2 B_i B_i^T \right),$$

which can be viewed as $B'\boldsymbol{z}$ where $B' = \sqrt{\sum_{i=1}^n x_i^2 B_i B_i^T}$ and $\boldsymbol{z} \sim \mathcal{N}(0, I_m)$.

By Theorem 6.3.2 in [23], we have

$$\left\| \|B'\boldsymbol{z}\| - \|B'\|_F \right\|_{\psi_2} \leq C K^2 \|B'\|,$$

where $K = \max_i \|z_i\|_{\psi_2}$ and $\|\cdot\|_{\psi_2}$ is the sub-guassian norm (see Definition 2.5.6 in [23]) and $C$ is an absolute constant.

By the definition of sub-gaussian norm, we can use the tail bound. For some positive absolute constant $c$ and for any $\mu > 0$,

$$\mathbb{P}\left\{ \|B'\boldsymbol{z}\| - \|B'\|_F \geq u \right\} \leq 2\exp(-cu^2/(K^4\|B'\|^2)).$$

Then we unfix $\boldsymbol{x}$ using a union bound. With probability at least $1 - 9^n 2 \exp(-cu^2/(K^4\|B'\|^2))$

$$\sup_{\boldsymbol{x} \in \mathcal{N}} \|B'\boldsymbol{z}\| - \|B'\|_F \le \mu.$$

Choose $u = CK^2\|B'\|(\sqrt{n} + t)$. If the constant $C$ is chosen sufficiently large, we can let $cu^2/K^4 \ge 3n + t^2$. Thus,

$$\mathbb{P}\left\{ \sup_{\boldsymbol{x} \in \mathcal{N}} \|B'\boldsymbol{z}\| - \|B'\|_F \ge u \right\} \le 9^n 2 \exp\left(-3n - t^2\right) \le 2\exp(-t^2).$$

Combined with $\|W\| \le 2\sup_{\boldsymbol{x} \in \mathcal{N}} \|W\boldsymbol{x}\|$, we conclude that

$$\mathbb{P}\left\{ \|W\| \ge 2CK^2\|B'\|(\sqrt{n} + t) + 2\|B'\|_F \right\} \le 2\exp(-t^2).$$

Noticing that $\|B'\| \le \max_i \|B_i\|$ and $\|B'\|_F \le \max_i \|B_i\|_F$, we have

$$\mathbb{P}\left\{ \|W\| \ge 2CK^2\max_i\|B_i\|(\sqrt{n} + t) + 2\max_i\|B_i\|_F \right\} \le 2\exp(-t^2).$$

We absorb $K$ into $C$ as $K$ is a constant. With abuse of notation of $C$ which is absolute, we have

$$\mathbb{P}\left\{ \|W\| \ge C(\max_i\|B_i\|(\sqrt{n} + t) + \max_i\|B_i\|_F) \right\} \le 2\exp(-t^2).$$

$\square$

**Lemma K.6.** *For a random $m \times n$ matrix $W = [B_1\boldsymbol{a}_1, B_2\boldsymbol{a}_2, ..., B_n\boldsymbol{a}_n]$ where $A = [\boldsymbol{a}_1, \boldsymbol{a}_2, ..., \boldsymbol{a}_n]$ and $B_1, B_2, ..., B_n$ is a sequence of $m \times N$ non-random matrices. Here $A = A_0 + \Delta A$ where $A_0$ is an $N \times n$ random matrix whose entries i.i.d. follow $\mathcal{N}(0,1)$ and $\Delta A$ is a fixed matrix with $\|\Delta A\|_F \le R$ given constant $R > 0$. We have for some absolute constant $C > 0$, for any $t \ge 0$*

$$\|W\| \le C\left( \max_i\|B_i\|(\sqrt{n} + R + t) + \max_i\|B_i\|_F \right) \tag{46}$$

*with probability at least $1 - 2\exp(-t^2)$.*

*Proof.* Comparing to Lemma K.5, we only need to bound the norm of $\Delta W$:

$$\Delta W := [B_1\Delta\boldsymbol{a}_1, B_2\Delta\boldsymbol{a}_2, , ..., B_n\Delta\boldsymbol{a}_n],$$

where $\Delta A = [\Delta\boldsymbol{a}_1, \Delta\boldsymbol{a}_2, ..., \Delta\boldsymbol{a}_n]$.

By the definition that $\|A_0\|_F = \sqrt{\sum_{i=1}^n \|\Delta\boldsymbol{a}_i\|^2}$, we have

$$\|\Delta W\| \le \|\Delta W\|_F = \sqrt{\sum_{i=1}^n \|B_i\Delta\boldsymbol{a}_i\|^2} \le \max_i\|B_i\|\|\Delta A\|_F \le \max_i\|B_i\|R.$$

Therefore, for any $t \ge 0$, with probability at least $1 - 2\exp(-t^2)$,

$$\|W\| \le \|W - \Delta W\| + \|\Delta W\| \le C\left( \max_i\|B_i\|(\sqrt{n} + R + t) + \max_i\|B_i\|_F \right).$$

$\square$

**Lemma K.7.** *Consider a fixed matrix $B \in \mathbb{R}^{m \times n}$ and a random vector $\mathbf{a}_0 \sim \mathcal{N}(0, I_n)$. For any $\mathbf{a} \in \mathbb{R}^n$ in the ball $\mathsf{B}(\mathbf{a}_0, R)$ given constant $R > 0$, for any $t > 0$, we have with probability at least $1 - 2\exp(-ct^2/\|B\|^2)$, where $c$ is an absolute constant,*

$$\|B\mathbf{a}\| \le t + \|B\|_F + \|B\|R. \tag{47}$$

*Proof.* By Theorem 6.3.2 in [23], for any $t > 0$,

$$\mathbb{P}\{|\|B\mathbf{a}_0\| - \|B\|_F| \ge t\} \le 2\exp(-ct^2/\|B\|^2),$$

where $c > 0$ is an absolute constant.

Note that $\|B\mathbf{a}\| \le \|B\mathbf{a}_0\| + \|B(\mathbf{a} - \mathbf{a}_0)\| \le \|B\mathbf{a}_0\| + \|B\|R$. With probability at least $1 - 2\exp(-ct^2/\|B\|^2)$, we have

$$\|B\mathbf{a}\| \le t + \|B\|_F + \|B\|R.$$

$\square$

**Lemma K.8** (Matrix Gaussian series). *For a sequence of fixed matrices $\{B_k\}_{k=1}^n$ with dimension $d_1 \times d_2$ and a sequence of independent standard normal variables $\{\gamma_k\}$, we define $Z = \sum_{k=1}^n (\gamma_k + \Delta\gamma_k)B_k$ where $\{\Delta\gamma_k\}_{k=1}^n$ is a fixed sequence with $\sum_{k=1}^n \Delta\gamma_k^2 \leq R^2$ given constant $R > 0$. Then we have for any $t \geq 0$, with probability at least $1 - (d_1 + d_2)\exp(-t^2/(2\nu))$,*

$$\|Z\| \leq t + R\nu, \tag{48}$$

*where*

$$\nu = \max\left\{\left\|\sum_k B_k B_k^T\right\|, \left\|\sum_k B_k^T B_k\right\|\right\}. \tag{49}$$

*Proof.* By Theorem 4.1.1 in [22], for all $t \geq 0$,

$$\mathbb{P}\left(\left\|\sum_{k=1}^n \gamma_k B_k\right\| \geq t\right) \leq (d_1 + d_2)\exp\left(\frac{-t^2}{2\nu}\right). \tag{50}$$

Since

$$\left\|Z - \sum_{k=1}^n \gamma_k B_k\right\| = \left\|\sum_{k=1}^n \Delta\gamma_k B_k\right\|$$

$$\leq \sqrt{\sum_{k=1}^n (\Delta\gamma_k)^2}\sqrt{\left\|\sum_{k=1}^n B_k B_k^T\right\|}$$

$$\leq R\sqrt{\nu}.$$

Then for $Z$, we have

$$\mathbb{P}(\|Z\| \geq t + R\sqrt{\nu}) \leq (d_1 + d_2)\exp\left(\frac{-t^2}{2\nu}\right).$$

$\square$

**Lemma K.9** (Bound on matrix variance). *For any $\ell \in [L], \ell_1, \ell_2 \in [\ell], j \in [d_{\ell+1}]$, with probability at least $1 - \exp(-C_{\ell_1,\ell_2}^{(\ell),j}\log^2 m)$ for some constant $C_{\ell_1,\ell_2}^{(\ell),j} > 0$, we have*

$$\mu_{\ell_1,\ell_2}^{(\ell),j} := \frac{1}{m_j^{(\ell+1)}}\left\|\sum_{i:f_i^{(\ell)}\in\mathcal{F}_{\mathcal{S}_j^{(\ell+1)}}}\left(\sigma''\left(\tilde{f}_i^{(\ell)}\right)\right)^2\left\|\frac{\partial\tilde{f}_i^{(\ell)}}{\partial\mathbf{w}^{(\ell_1)}}\right\|^2\frac{\partial\tilde{f}_i^{(\ell)}}{\partial\mathbf{w}^{(\ell_2)}}\left(\frac{\partial\tilde{f}_i^{(\ell)}}{\partial\mathbf{w}^{(\ell_2)}}\right)^T\right\|$$

$$= O\left(\max\left(1/m_j^{(\ell+1)}, \max_{\min(\ell_1,\ell_2)+1\leq p\leq\ell} 1/\underline{m}_p\right)(\log m + R)^{4\ell-2}\right)$$

$$= \tilde{O}\left(\max\left(1/m_j^{(\ell+1)}, \max_{\min(\ell_1,\ell_2)+1\leq p\leq\ell} 1/\underline{m}_p\right)R^{4\ell-2}\right). \tag{51}$$

*Proof.* Without lose of generality, we assume $\ell_1 \leq \ell_2 \leq \ell$.

We consider two scenarios, (a) $\ell_1 \leq \ell_2 = \ell$ and (b) $\ell_1 \leq \ell_2 < \ell$.

In the scenario (a), we analyze $\ell_1 = \ell_2 = \ell$ and $\ell_1 < \ell_2 = \ell$ respectively.

When $\ell_1 = \ell_2 = \ell$, by definition,

$$\frac{\partial\tilde{f}_i^{(\ell)}}{\partial\mathbf{w}^{(\ell_1)}} = \frac{1}{\sqrt{m_i^{(\ell)}}}f_{\mathcal{S}_i^{(\ell)}}.$$

By Lemma K.3, with probability at least $1 - m_i^{(\ell)}\exp(-C_\ell^{\mathcal{P}}\log^2 m)$, $\left\|f_{\mathcal{S}_i^{(\ell)}}\right\| = O\left(\sqrt{m_i^{(\ell)}}(\log m + R)^{\ell-1}\right) = \tilde{O}\left(\sqrt{m_i^{(\ell)}}\right)$. Applying union bound over the indices $i$ such

that $f_i^{(\ell)} \in f_{\mathcal{S}_j^{(\ell+1)}}$, the carnality of which is at most $m_j^{(\ell+1)}$, we have with probability at least $1 - m_i^{(\ell)} m_j^{(\ell+1)} \exp(-C_\ell^{\mathcal{P}} \log^2 m)$,

$$\max_{i: f_i^{(\ell)} \in \mathcal{F}_{\mathcal{S}_j^{(\ell+1)}}} \left\| \frac{\partial \tilde{f}_i^{(\ell)}}{\partial \mathbf{w}^{(\ell_1)}} \right\| = \max_{i: f_i^{(\ell)} \in \mathcal{F}_{\mathcal{S}_j^{(\ell+1)}}} \frac{\left\| f_{\mathcal{S}_i^{(\ell)}} \right\|}{\sqrt{m_i^{(\ell)}}} = O\left((\log m + R)^{\ell-1}\right) = \tilde{O}(R^{\ell-1}).$$

It is not hard to see that

$$\sum_{i: f_i^{(\ell)} \in \mathcal{F}_{\mathcal{S}_j^{(\ell+1)}}} \frac{\partial \tilde{f}_i^{(\ell)}}{\partial \mathbf{w}^{(\ell)}} \left( \frac{\partial \tilde{f}_i^{(\ell)}}{\partial \mathbf{w}^{(\ell)}} \right)^T$$

is a block diagonal matrix with $i$-th block in the form $\frac{1}{m_i^{(\ell)}} f_{\mathcal{S}_i^{(\ell)}} \left( f_{\mathcal{S}_i^{(\ell)}} \right)^T \cdot \mathbb{I}\left\{ f_i^{(\ell)} \in f_{\mathcal{S}_j^{(\ell+1)}} \right\}$.

Therefore, $\mu_{\ell,\ell}^{(\ell),j}$ can be bounded by

$$\mu_{\ell,\ell}^{(\ell),j} \leq \frac{1}{m_j^{(\ell+1)}} \gamma_2^2 \left( \max_{i: f_i^{(\ell)} \in \mathcal{F}_{\mathcal{S}_j^{(\ell+1)}}} \frac{\left\| f_{\mathcal{S}_i^{(\ell)}} \right\|}{\sqrt{m_i^{(\ell)}}} \right)^2 \left( \max_{i: f_i^{(\ell)} \in \mathcal{F}_{\mathcal{S}_j^{(\ell+1)}}} \left\| \frac{1}{m_i^{(\ell)}} f_{\mathcal{S}_i^{(\ell)}} \left( f_{\mathcal{S}_i^{(\ell)}} \right)^T \right\| \right)$$

$$= O\left((\log m + R)^{4\ell-4}/m_j^{(\ell+1)}\right) = \tilde{O}\left(R^{4\ell-4}/m_j^{(\ell+1)}\right),$$

with probability at least $1 - 2m_i^{(\ell)} m_j^{(\ell+1)} \exp(-C_\ell^{\mathcal{P}} \log^2 m)$, where we apply the union bound on $\left\| f_{\mathcal{S}_i^{(\ell)}} \right\|$ once again.

By definition Eq. (25), $m_i^{(\ell)} \leq \overline{m}_\ell$ and $m_j^{(\ell+1)} \leq \overline{m}_{\ell+1}$. By Assumption 3.5, $\overline{m}_\ell, \overline{m}_{\ell+1}$ are polynomial in $m$. If $m$ is large enough, we can find a constant $C_{\ell,\ell}^{(\ell),j} > 0$ such that

$$\exp(-C_{\ell,\ell}^{(\ell),j} \log^2 m) > 2m_i^{(\ell)} m_j^{(\ell+1)} \exp(-C_\ell^{\mathcal{P}} \log^2 m),$$

thus the bound holds with probability $1 - \exp\left(-C_{\ell,\ell}^{(\ell),j} \log^2 m\right)$.

When $\ell_1 < \ell_2 = \ell$, By Eq. (5), we compute the derivative:

$$\frac{\partial \tilde{f}_i^{(\ell)}}{\partial \mathbf{w}^{(\ell_1)}} = \frac{1}{\sqrt{m_i^{(\ell)}}} \frac{\partial f_{\mathcal{S}_i^{(\ell)}}}{\partial \mathbf{w}^{(\ell_1)}} \mathbf{w}_i^{(\ell)}. \tag{52}$$

By Lemma B.1, with probability at least $1 - \exp\left(-C_{\ell,\ell_1}^f \log^2 m\right)$, $\left\| \partial f_{\mathcal{S}_i^{(\ell)}} / \partial \mathbf{w}^{(\ell_1)} \right\| = \tilde{O}\left(\max_{\ell_1+1 \leq p \leq \ell} \sqrt{m_i^{(\ell)}} / \sqrt{\overline{m}_p}\right)$ and $\left\| \partial f_{\mathcal{S}_i^{(\ell)}} / \partial \mathbf{w}^{(\ell_1)} \right\|_F = \tilde{O}\left(\sqrt{m_i^{(\ell)}}\right)$. We use Lemma K.7 and choose $t = \log m \left\| \partial f_{\mathcal{S}_i^{(\ell)}} / \partial \mathbf{w}^{(\ell_1)} \right\|$, then with probability at least $1 - 2\exp(-c' \log^2 m) - \exp\left(-C_{\ell,\ell_1}^f \log^2 m\right)$ for some absolute constant $c' > 0$,

$$\left\| \frac{\partial \tilde{f}_i^{(\ell)}}{\partial \mathbf{w}^{(\ell_1)}} \right\| = \frac{1}{\sqrt{m_i^{(\ell)}}} \left\| \frac{\partial f_{\mathcal{S}_i^{(\ell)}}}{\partial \mathbf{w}^{(\ell_1)}} \mathbf{w}_i^{(\ell)} \right\| \tag{53}$$

$$\leq \frac{1}{\sqrt{m_i^{(\ell)}}} \left( (\log m + R) \left\| \frac{\partial f_{\mathcal{S}_i^{(\ell)}}}{\partial \mathbf{w}^{(\ell_1)}} \right\| + \left\| \frac{\partial f_{\mathcal{S}_i^{(\ell)}}}{\partial \mathbf{w}^{(\ell_1)}} \right\|_F \right) \tag{54}$$

$$= O\left((\log m + R)^\ell\right) = \tilde{O}(R^\ell). \tag{55}$$

Similar to the case when $\ell_1 = \ell_2 = \ell$,

$$\sum_{i:f_i^{(\ell)} \in \mathcal{F}_{\mathcal{S}_j^{(\ell+1)}}} \frac{\partial \tilde{f}_i^{(\ell)}}{\partial \mathbf{w}^{(\ell)}} \left( \frac{\partial \tilde{f}_i^{(\ell)}}{\partial \mathbf{w}^{(\ell)}} \right)^T$$

is a block matrix.

Therefore,

$$\mu_{\ell,\ell}^{(\ell),j} \leq \frac{1}{m_j^{(\ell+1)}} \gamma_2^2 \left( \max_{i:f_i^{(\ell)} \in \mathcal{F}_{\mathcal{S}_j^{(\ell+1)}}} \left\| \frac{\partial \tilde{f}_i^{(\ell)}}{\partial \mathbf{w}^{(\ell_1)}} \right\| \right)^2 \left( \max_{i:f_i^{(\ell)} \in \mathcal{F}_{\mathcal{S}_j^{(\ell+1)}}} \left\| \frac{1}{m_i^{(\ell)}} f_{\mathcal{S}_i^{(\ell)}} \left( f_{\mathcal{S}_i^{(\ell)}} \right)^T \right\| \right)$$

$$= O\left( (\log m + R)^{4\ell-2} / m_j^{(\ell+1)} \right) = \tilde{O}\left( R^{4\ell-2} / m_j^{(\ell+1)} \right),$$

with probability at least $1 - 2m_j^{(\ell+1)} \exp(-c' \log^2 m) - m_j^{(\ell+1)} \exp\left( -C_{\ell,\ell_1}^f \log^2 m \right) - 2m_i^{(\ell)} m_j^{(\ell+1)} \exp\left( -C_\ell^{\mathcal{P}} \log^2 m \right)$ where we apply the union bound over the indices $i$ for the maximum.

Similarly, we can find a constant $C_{\ell_1,\ell}^{(\ell),j} > 0$ such that the bound holds with probability $1 - \exp\left( -C_{\ell,\ell}^{(\ell),j} \log^2 m \right)$.

For scenario (b) that $\ell_1 \leq \ell_2 < \ell$, we apply Lemma K.6 to bound $\mu_{\ell_1,\ell_2}^{(\ell),j}$. Specifically, we view

$$B_i = \frac{1}{\sqrt{m_i^{(\ell)}}} \left| \sigma''(\tilde{f}_i^{(\ell)}) \right| \left\| \frac{\partial \tilde{f}_i^{(\ell)}}{\partial \mathbf{w}^{(\ell_1)}} \right\| \frac{\partial f_{\mathcal{S}_i^{(\ell)}}}{\partial \mathbf{w}^{(\ell_2)}}, \tag{56}$$

$$\mathbf{a}_i = \mathbf{w}_i^{(\ell)}. \tag{57}$$

Choosing $t = \log m$ and supposing $B_i$ is fixed, then with probability at least $1 - 2\exp(-\log^2 m)$, for some constant $K_{\ell_1,\ell_2}^{\ell,j} > 0$,

$$\left\| \sum_{i:f_i^{(\ell)} \in f_{\mathcal{S}_j^{(\ell+1)}}} \left( \sigma''(\tilde{f}_i^{(\ell)}) \right)^2 \left\| \frac{\partial \tilde{f}_i^{(\ell)}}{\partial \mathbf{w}^{(\ell_1)}} \right\|^2 \frac{\partial \tilde{f}_i^{(\ell)}}{\partial \mathbf{w}^{(\ell_2)}} \left( \frac{\partial \tilde{f}_i^{(\ell)}}{\partial \mathbf{w}^{(\ell_2)}} \right)^T \right\|$$

$$\leq (K_{\ell_1,\ell_2}^{\ell,j})^2 \left( \max_i \|B_i\| \left( \sqrt{m_j^{(\ell+1)}} + \log m + R \right) + \max_i \|B_i\|_F \right)^2$$

$$\leq (K_{\ell_1,\ell_2}^{\ell,j})^2 \gamma_2^2 \left( \max_i \frac{1}{\sqrt{m_i^{(\ell)}}} \left\| \frac{\partial \tilde{f}_i^{(\ell)}}{\partial \mathbf{w}^{(\ell_1)}} \right\| \left\| \frac{\partial f_{\mathcal{S}_i^{(\ell)}}}{\partial \mathbf{w}^{(\ell_2)}} \right\| \left( \sqrt{m_j^{(\ell+1)}} + \log m + R \right) + \max_i \frac{1}{\sqrt{m_i^{(\ell)}}} \left\| \frac{\partial \tilde{f}_i^{(\ell)}}{\partial \mathbf{w}^{(\ell_1)}} \right\| \left\| \frac{\partial f_{\mathcal{S}_i^{(\ell)}}}{\partial \mathbf{w}^{(\ell_2)}} \right\|_F \right)^2$$

By Eq. (55), with probability at least $1 - 2\exp(-c' \log^2 m) - \exp\left( -C_{\ell,\ell_1}^f \log^2 m \right)$ for some absolute constant $c' > 0$,

$$\left\| \frac{\partial \tilde{f}_i^{(\ell)}}{\partial \mathbf{w}^{(\ell_1)}} \right\| = \tilde{O}(R^\ell). \tag{58}$$

By Lemma B.1, with probability at least $1 - \exp(-C_{\ell,\ell_2}^f \log^2 m)$, $\left\| \partial f_{\mathcal{S}_i^{(\ell)}} / \partial \mathbf{w}^{(\ell_2)} \right\| = \tilde{O}\left( \max_{\ell_2+1 \leq p \leq \ell} \sqrt{m_i^{(\ell)}} / \sqrt{m_p} \right)$ and $\left\| \partial f_{\mathcal{S}_i^{(\ell)}} / \partial \mathbf{w}^{(\ell_2)} \right\|_F = \tilde{O}\left( \sqrt{m_i^{(\ell)}} \right)$.

Combined them together, with probability at least $1 - m_i^{(\ell)} m_j^{(\ell+1)} \exp\left( -C_\ell^{\mathcal{P}} \log^2 m \right) - 2m_j^{(\ell+1)} \exp(-c' \log^2 m) - m_j^{(\ell+1)} \exp\left( -C_{\ell,\ell_1}^f \log^2 m \right) - m_j^{(\ell+1)} \exp\left( -C_{\ell,\ell_2}^f \log^2 m \right)$,

$$\mu_{\ell_1,\ell_2}^{(\ell),j} = O\left(\max\left(1/m_j^{(\ell+1)}, \max_{\ell_1+1 \leq p \leq \ell} 1/\underline{m}_p\right)(\log m + R)^{4\ell-2}\right)$$

$$= \tilde{O}\left(\max\left(1/m_j^{(\ell+1)}, \max_{\ell_1+1 \leq p \leq \ell} 1/\underline{m}_p\right)R^{4\ell-2}\right).$$

Similarly we can find a constant $C_{\ell_1,\ell_2}^{(\ell),j} > 0$ such that with probability at least $1 - \exp\left(-C_{\ell_1,\ell_2}^{(\ell),j}\log^2 m\right)$ the above bound holds.

For $\ell_1 \geq \ell_2$, we similarly have

$$\mu_{\ell_2,\ell_1}^{(\ell),j} = \tilde{O}\left(\max\left(1/m_j^{(\ell+1)}, \max_{\ell_2+1 \leq p \leq \ell} 1/\underline{m}_p\right)R^{4\ell-2}\right),$$

with probability at least $1 - \exp(-C_{\ell_2,\ell_1}^{(\ell),j}\log^2 m)$. $\qquad\square$

**Lemma K.10.** *For any $0 < \ell' \leq \ell \leq L - 1$, given fixed matrices $U_1, ..., U_{d_{\ell'}} \in \mathbb{R}^{u_1 \times u_2}$, with probability at least $1 - \sum_{k=1}^{\ell-\ell'+1} k(u_1 + u_2)\exp(-\log^2 m/2)$*

$$\sum_{i=1}^{d_{\ell'}} U_i \frac{\partial \tilde{f}_j^{(\ell+1)}}{\partial \tilde{f}_i^{(\ell')}} = O\left(\max_{i:f_i^{(\ell')} \in \mathcal{F}_{\mathcal{S}_j^{(\ell'+1)}}} \|U_i\|(\log m + R)^{\ell-\ell'+1}\right)$$

$$= \tilde{O}\left(\max_{i:f_i^{(\ell')} \in \mathcal{F}_{\mathcal{S}_j^{(\ell'+1)}}} \|U_i\|\right).$$

*Proof.* We prove the result by induction.

For the base case that $\ell = \ell'$,

$$\sum_{i=1}^{d_{\ell'}} U_i \frac{\partial \tilde{f}_j^{(\ell'+1)}}{\partial \tilde{f}_i^{(\ell')}} = \sum_{i=1}^{d_{\ell'}} U_i \sigma'\left(\tilde{f}_i^{(\ell')}\right)\frac{\partial f_{\mathcal{S}_j^{(\ell'+1)}}}{\partial f_i^{(\ell')}}\frac{\partial \tilde{f}_j^{(\ell'+1)}}{\partial f_{\mathcal{S}_j^{(\ell'+1)}}}$$

$$= \frac{1}{\sqrt{m_j^{(\ell'+1)}}}\sum_{i:f_i^{(\ell')} \in \mathcal{F}_{\mathcal{S}_j^{(\ell'+1)}}} U_i \sigma'\left(\tilde{f}_i^{(\ell')}\right)\left(\mathbf{w}_j^{(\ell'+1)}\right)_{\mathrm{id}_{\ell',i}^{\ell'+1,j}}.$$

We view the above equation as a matrix Gaussian series with respect to $\mathbf{w}_j^{(\ell'+1)}$. Its matrix variance $\nu^{(\ell')}$ can be bounded by

$$\nu^{(\ell')} := \frac{1}{m_j^{(\ell'+1)}}\left\|\sum_{i:f_i^{(\ell')} \in \mathcal{F}_{\mathcal{S}_j^{(\ell'+1)}}} U_i \sigma'\left(\tilde{f}_i^{(\ell')}\right)\right\|^2$$

$$\leq \max_{i:f_i^{(\ell')} \in \mathcal{F}_{\mathcal{S}_j^{(\ell'+1)}}} \gamma_1^2 \|U_i\|^2.$$

Using Lemma K.8 and choosing $t = \log m\sqrt{\nu^{(\ell')}}$, we have with probability at least $1 - (u_1 + u_2)\exp(-\log^2 m/2)$,

$$\sum_{i=1}^{d_{\ell'}} U_i \frac{\partial \tilde{f}_j^{(\ell'+1)}}{\partial \tilde{f}_i^{(\ell')}} \leq (\log m + R)\sqrt{\nu^{(\ell')}} \leq \max_i(\log m + R)\gamma_1\|U_i\| = O((\log m + R)\max_i \|U_i\|).$$

Suppose with probability at least $1 - \sum_{k=1}^{\ell-\ell'+1} k(u_1 + u_2)\exp(-\log^2 m/2)$, for all $\ell' \le k \le \ell$,

$$\sum_{i=1}^{d_{\ell'}} U_i \frac{\partial \tilde{f}_j^{(k)}}{\partial \tilde{f}_i^{(\ell')}} = O\left( \max_{i:f_i^{(\ell')} \in \mathcal{F}_{\mathcal{S}_j^{(\ell'+1)}}} \|U_i\| (\log m + R)^{k-\ell'} \right).$$

Then when $k = \ell + 1$, we have

$$\sum_{i=1}^{d_{\ell'}} U_i \frac{\partial \tilde{f}_j^{(\ell+1)}}{\partial \tilde{f}_i^{(\ell')}} = \sum_{r=\ell'}^{\ell} \sum_{i=1}^{d_{\ell'}} U_i \frac{\partial f^{(r)}}{\partial \tilde{f}_i^{(\ell')}} \frac{\partial f_{\mathcal{S}_j^{(\ell+1)}}}{\partial f^{(r)}} \frac{\partial \tilde{f}_j^{(\ell+1)}}{\partial f_{\mathcal{S}_j^{(\ell+1)}}}$$

$$= \sum_{r=\ell'}^{\ell} \sum_{s=1}^{d_r} \sum_{i=1}^{d_{\ell'}} U_i \frac{\partial f_s^{(r)}}{\partial \tilde{f}_i^{(\ell')}} \frac{\partial f_{\mathcal{S}_j^{(\ell+1)}}}{\partial f_s^{(r)}} \frac{\partial \tilde{f}_j^{(\ell+1)}}{\partial f_{\mathcal{S}_j^{(\ell+1)}}}$$

$$= \sum_{r=\ell'}^{\ell} \sum_{s=1}^{d_r} \sum_{i=1}^{d_{\ell'}} U_i \frac{\partial \tilde{f}_s^{(r)}}{\partial \tilde{f}_i^{(\ell')}} \sigma'\left(\tilde{f}_s^{(r)}\right) \frac{\partial f_{\mathcal{S}_j^{(\ell+1)}}}{\partial \tilde{f}_s^{(r)}} \frac{\partial \tilde{f}_j^{(\ell+1)}}{\partial f_{\mathcal{S}_j^{(\ell+1)}}}$$

$$= \sum_{r=\ell'}^{\ell} \sum_{s:f_s^{(r)} \in \mathcal{F}_{\mathcal{S}_j^{(\ell+1)}}} \left(\sum_{i=1}^{d_{\ell'}} U_i \frac{\partial \tilde{f}_s^{(r)}}{\partial \tilde{f}_i^{(\ell')}}\right) \sigma'\left(\tilde{f}_s^{(r)}\right) \frac{1}{\sqrt{m_j^{(\ell+1)}}} \left(\mathbf{w}_j^{(\ell+1)}\right)_{\mathrm{id}_{r,s}^{\ell+1,j}}$$

For each $r \in \{\ell', ..., \ell\}$, we view $\sum_{s:f_s^{(r)} \in \mathcal{F}_{\mathcal{S}_j^{(\ell+1)}}} \left(\sum_{i=1}^{d_{\ell'}} U_i \frac{\partial \tilde{f}_s^{(r)}}{\partial \tilde{f}_i^{(\ell')}}\right) \sigma'\left(\tilde{f}_s^{(r)}\right) \frac{1}{\sqrt{m_j^{(\ell+1)}}} \left(\mathbf{w}_j^{(\ell+1)}\right)_{\mathrm{id}_{r,s}^{\ell+1,j}}$
as a matrix Gaussian series with respect to $\mathbf{w}_j^{(\ell+1)}$.
By the inductive hypothesis, for all $r$, its matrix variance can be bounded by

$$\nu^{(r)} := \frac{1}{m_j^{(\ell+1)}} \left\| \sum_{s:f_s^{(r)} \in \mathcal{F}_{\mathcal{S}_j^{(\ell+1)}}} \left(\sum_{i=1}^{d_{\ell'}} U_i \frac{\partial \tilde{f}_s^{(r)}}{\partial \tilde{f}_i^{(\ell')}}\right) \sigma'\left(\tilde{f}_s^{(r)}\right) \right\|^2$$

$$= O\left( \max_{i:f_i^{(\ell')} \in \mathcal{F}_{\mathcal{S}_j^{(\ell'+1)}}} \|U_i\|^2 (\log m + R)^{2r-2\ell'} \right).$$

Then we use Lemma K.8 and choose $t = \log m \sqrt{\nu^{(r)}}$. With probability at least $1 - (u_1 + u_2)\exp(-\log^2 m/2)$,

$$\left\| \sum_{s:f_s^{(r)} \in \mathcal{F}_{\mathcal{S}_j^{(\ell+1)}}} \left(\sum_{i=1}^{d_{\ell'}} U_i \frac{\partial \tilde{f}_s^{(r)}}{\partial \tilde{f}_i^{(\ell')}}\right) \sigma'\left(\tilde{f}_s^{(r)}\right) \frac{1}{\sqrt{m_j^{(\ell+1)}}} \left(\mathbf{w}_j^{(\ell+1)}\right)_{\mathrm{id}_{r,s}^{\ell+1,j}} \right\|$$

$$\le (\log m + R)\sqrt{\nu^{(r)}}$$
$$\le \max_{i:f_i^{(\ell')} \in \mathcal{F}_{\mathcal{S}_j^{(\ell'+1)}}} (\log m + R)\gamma_1 \|U_i\|$$

$$= O\left( \max_{i:f_i^{(\ell')} \in \mathcal{F}_{\mathcal{S}_j^{(\ell'+1)}}} \|U_i\| (\log m + R)^{r-\ell'+1} \right).$$

We apply union bound over indices $r = \ell', ..., \ell$ and add the probability from the induction hypothesis. With probability at least $1 - \sum_{k=1}^{\ell-\ell'+1} k(u_1 + u_2) \exp(-\log^2 m/2)$,

$$\left\| \sum_{i=1}^{d_{\ell'}} U_i \frac{\partial \tilde{f}_j^{(\ell+1)}}{\partial \tilde{f}_i^{(\ell')}} \right\| \leq \sum_{r=\ell'+1}^{\ell} \left\| \sum_{s:f_s^{(r)} \in \mathcal{F}_{\mathcal{S}_j^{(\ell+1)}}} \left( \sum_{i=1}^{d_{\ell'}} U_i \frac{\partial \tilde{f}_s^{(r)}}{\partial \tilde{f}_i^{(\ell')}} \right) \sigma' \left( \tilde{f}_s^{(r)} \right) \frac{1}{\sqrt{m_j^{(\ell+1)}}} \left( \mathbf{w}_j^{(\ell+1)} \right)_{\mathrm{id}_{r,s}^{\ell+1,j}} \right\|$$

$$= O \left( \max_{i:f_i^{(\ell')} \in \mathcal{F}_{\mathcal{S}_j^{(\ell'+1)}}} \|U_i\| (\log m + R)^{\ell-\ell'+1} \right)$$

$$= \tilde{O} \left( \max_{i:f_i^{(\ell')} \in \mathcal{F}_{\mathcal{S}_j^{(\ell'+1)}}} \|U_i\| R^{\ell-\ell'+1} \right).$$

Then we finish the induction step which completes the proof. $\qquad\square$