# OpenReview forum: "Transition to Linearity of General Neural Networks with Directed Acyclic Graph Architecture"
_NeurIPS.cc/2022/Conference — NeurIPS 2022 Accept_

### Official Review · Reviewer_wA2n · 2022-07-11

**Rating:** 4
**Confidence:** 2
**Soundness:** 3 good
**Presentation:** 2 fair
**Contribution:** 3 good

**Summary:**

The paper extends the key result from the neural tangent kernel on the transition to linearity as width of dense neural networks grows to infinity and shows that feedforward neural networks that can be captured as directed acyclic graphs exhibit similar property captured by large in-degree. The results in the paper are more general than existing ones on transition to linearity. The paper gives theoretical justifications for the results but is lacking on the empirical aspects.

**Questions:**

I would like to ask the following questions:
1. How does the result actually apply to CNN, ResNets, and DenseNets that the author claims? May consider adding related experiments (may be simple) for this in the next revision.
2. Would the constant in (12) depend on the c in Assumption 3.5?

**Limitations:**

See main review.

**Strengths And Weaknesses:**

[Strength] The paper has the following strength.
- The paper works on top of the neural tangent kernel and extends its key property of transition to linearity under infinite width to more general network structures captured by directed acyclic graphs, which is a more general result from existing ones.
- The paper contextualizes well by offering reference to existing works and discussions.
- The paper provides a view from optimization of the loss via Polyak-Lojasiewicz theories which sees potential convergence rate arguments under the framework.
- The paper is well written overall except for the lack of experiments.

[Weakness] However, it has the following weakness which stops me from recommending the paper for publication in NeurIPS.
- The paper has focused solely on theoretical justifications and offers limited empirical evidence. Though I agree the overall atmosphere in the community focusing on empirical evidence may be detrimental, but I still believe carefully designed experiments would enable a wider audience to understand the work better (myself included). For example, the claims on linearity for the DAG-captured networks can be empirically shown like the experiments in the original neural tangent kernel paper.
- The scope of the work is incremental in its nature by extending the property of transition to linearity to other explicit neural networks. I am very curious on whether these results extend to attention based structures and implicit models. Though the reviewer fully recognizes that the difficulty would be very high in this direction (already considerable as presented, the reviewer believes, not related to the incremental nature).

---

> ### Author Response · Authors · 2022-08-02
> **Response to Reviewer wA2n**
>
> We thank the reviewer for the comments.
>
> _Weakness: application to attention based structures and implicit models._
>
> Thanks for bringing up this interesting question! We are aware that there is some research analyzing the transition to linearity of attention based structures, but it's out of the scope of our work. We will leave it as the future work.
>
> _Q1. How does the results apply to CNN, ResNets and DenseNets? Experiments?_
>
> Our DAG architecture strictly includes DenseNets architectures (example in Appendix A) and can be easily extended ResNets, as discussion in Appendix D. Therefore, our theoretical analysis prove the property of transition to linearity for FCNs and ResNets. For CNNs, we believe our results can handle them with minor modification of our analysis, to accommodate the shared weights in CNNs.
>
> For the experiment, we have added an experiment in Appendix J in revision. More experiments will be added later. Also please see the top of the page as a highlight of our experiment.
>
> _Q2. the constant in (12) depend on c in Assumption 3.5_
>
> The constant $c$ in Assumption 3.5 puts a constraint on the largest relative in-degree. Our bound of the norm of the Hessian relies on the minimum in-degree hence the bound will not rely on the constant $c$. The $c$ appears at the probability, which is introduced by the tail bound of the matrix norm and matrix Gaussian series (Lemma I4-I8). Specifically, the probability in THM 3.6 will take form of $1- \exp( \Omega(c\log^2 m))$,

---

### Official Review · Reviewer_vWFG · 2022-07-21

**Rating:** 4
**Confidence:** 4
**Soundness:** 3 good
**Presentation:** 3 good
**Contribution:** 2 fair

**Summary:**

In the "NTK" regime, the behavior of a neural network becomes similar to a linear model (i.e., the first-order approximation at initialization) when the network width approaches infinity. This has been established by many authors under standard architectures such as fully-connected and convolutional neural networks.

This paper considers feed-forward networks that can be represented as directed acyclic graphs (DAGs), which is an extension of existing feed-forward architectures such as fully-connected networks, convolutional networks, densenet, etc.

Defining the minimum in-degree (# of edges coming into a node) as the "width" $m$, the paper shows that the Hessian of the network function decays as $O(1/\sqrt{m})$ under the standard NTK parameterization, which implies that the network becomes similar to a linear approximation around its initialization as $m \to \infty$.

Such an observation leads to consequences in optimization: If the minimum eigenvalue of the NTK kernel matrix is bounded away from zero at initialization, then linear convergence to a global minimum can be implied.


**Questions:**

Q1. In the main results, there is the radius constant $R > 0$. The result seems to hold for any $R$, so it is natural to expect to see some dependence on $R$ appearing in the theorem statement. However, as it is stated in the paper, $R$ does not show up both in the spectral norm and the probability. This seems unnatural to me and I think there should be some $R$ hiding in the $O$ notation. What is the exact dependency on $R$ in the theorem statements?

Q2. In (8), why do you take the maximum up to $L-1$? Does it mean that we ignore the in-degrees of output neurons? However, in Line 164 it is stated that "except input and first layers".

Q3. In line 193, it is stated that: "Here we assume the output dimension of the network function is one. The same analysis can be applied to multiple outputs". However, the main theorems are handling each of the multiple output neurons albeit separately. This is confusing.

Q4. I don't understand the equations below line 256. Here, $\ell'$ is supposed to be anything between $\ell_2$ and $L$ inclusive. However, it seems to me that in the 2nd terms of RHS (the ones that involve $\sigma''$), $\ell'$ is acting as if $\ell' = L-1$, the last hidden layer? Correct me if I'm mistaken.


**Limitations:**

I did not sense any problem with honesty regarding limitations of this paper or potential ethical harms related to this paper.

**Strengths And Weaknesses:**

S1. This paper extends the well-known NTK phenomenon to general feed-forward networks which can be represented by DAGs. This is a nontrivial thing to do, and they have apparently used advanced mathematical tools to achieve this (although I didn't have the time to check the proof).

W1. While I value the amount of effort put into proving these results, I have to say that the main results and their implications are somewhat exactly as expected. In [4], it was already proved for general smoothly parameterized models that increasing the scale $\alpha$ of the model leads to lazy training (i.e., NTK regime) where the training dynamics becomes similar to the first-order approximation at initialization. As mentioned in [4], for neural networks with NTK parameterization, increasing width $m$ corresponds to increasing the scale $\alpha$, hence we can expect a similar "linearization" behavior regardless of the network's architecture (except that the output layer has to have identity activation). For this reason, my impression of this paper is that the new insights provided by this paper are somewhat limited. Please correct me if I am mistaken.

W2. While the authors claim that the DAG formulation includes many popular architectures like CNN, ResNet, and Dropout, there is a caveat; I believe that DAG formulation as described in Section 2.1 does NOT include any of them properly. Again, please correct me if I am wrong, I would be happy to change my view.

First, it doesn't include CNN because the formulation does not carefully handle shared weights. The authors claim in Appendix A that their results continue to hold for such shared weights and I think this would be the case, but such dependencies must be handled with great care.

Second, since the formulation assigns weights to each edge, the formulation cannot include connections without tunable weights (i.e., identity skip connections in ResNets). The authors claim in Appendix D that they handle skip connections by considering a network of the form
$f_v = \sigma_v(\tilde f_v) + \tilde f_v$, but this is nothing but redefining the activation function $\sigma_v \to \sigma_v + Id$! A proper "ResNet" must be of the form $f_v = \Phi(f_u) + f_u$ where $\Phi$ contains some neural network layers, NOT a single pass through an activation function.

Third, Dropout drops some of the neurons during training, which can be perfectly modeled with this DAG formulation. However, what I'm concerned about is its dynamic nature; throughout training, different sets of neurons get dropped and put back to the architecture. I wonder if this DAG formulation is capable of modeling and handling such a stochastic and dynamic nature of Dropout layers.

W3. Partly due to the DAG structure, the paper has very confusing notation. My overall impression is that we are not gaining much out of this additional confusion.

Throughout, I got very confused about the output layer index $\ell$ or $L$. Stated in line 138, the output layer is layer $\ell$, but $\ell$ is also frequently used to index different lines. Also, if the network has depth $L$, shouldn't the output layer be $L$, the last layer? This already suggests that the paper could use more polishing, in terms of its clarity/writing.


A Minor point. Typo in Line 57: $\tilde O(1\sqrt m)$

---

> ### Author Response · Authors · 2022-08-02
> **Response to Reviewer vWFG**
>
> We thank the reviewer for the detailed and insightful comments.
>
> _W1: How's our results different from lazy training_
>
>
>
> We'd like to point out that the "lazy training" in [4] does not  describe neural networks with NTK parameterization, and does not explain its transition to linearity. Please note that, for neural networks with NTK parameterization [Jacot et al. 2018], increasing the width does NOT change the scale of the model itself. Instead, as shown in [LZB 2020], it decreases the scale of the Hessian matrix of the model (second derivative), while keeping the model scale unchanged ($=\Theta(1)$). We'd like refer the reviewer to paragraph "Can the transition to linearity be explained by model rescaling?" at the end of introduction in [LZB 2020] for a detailed discussion regarding the relation of transition to linearity of neural networks with the ``lazy training'' concept.
>
>
> _W1: new insights provided by our paper_
>
> This paper focuses on the relationship between the neural network architecture and the property of transition to linearity. Although this property has been found on standard neural network architectures, it was far from clear that whether these architectures are special or not. Our contributions in this paper include generalizing this property to a much more general architectural setting, and shedding light on understanding why those counterexamples (e.g. bottleneck neural networks analyzed in [LZB 2020]) cannot transition to linearity. With these results one can believe that transition to linearity, which is often counter-intuitive, is actually a very common property happening for many neural networks.
>
>
>
> _W2: caveat of DAG_
>
>
> We thank the reviewer for the helpful feedback! We will answer each point in the following.
>
> **CNN.** We agree with your point that our current analysis did not handle shared weights in CNN properly. In the revision, we removed our claims about CNN, though we still believe similar results should still hold with a bit additional treatment in the analysis. Specifically, when the weight is shared at most a constant $d$ times (for example in CNN), the bound on Hessian's spectral norm should still hold, as it is a union bound over $d$ similar terms.
>
>
> **ResNets.** About the formulation of ResNets, thanks for pointing out the error. We have revised the definition of ResNets in Appendix D and show that transition to linearity still hold for ResNets.
>
> Now the formulation is as follows:
>
>
> $$ g_{i,res}^{(\ell)} =\sigma_i^{(\ell)}\left(f_{i,res}^{(\ell)}\right) + f_{j,res}^{(\ell')}, ~~~ g_{i,res}^{(\ell)}= \frac{1}{\sqrt{m_{i}^{(\ell)}}} \left(w_i^{(\ell)}\right)^T f_{\mathcal{S}^{(\ell)}_i,res}$$
>
> where $0\leq \ell' < \ell < L$ and $j\in[d_{\ell'}]$. For the output layer $\ell = L$,
>
> $$    g_{i,res}^{(L)}= \frac{1}{\sqrt{m_{i}^{(L)}}}\left(w_i^{(L)}\right)^T f_{\mathcal{S}^{(L)}_i,res}.$$
>
> Here $g$ is the preactivation corresponding to $f$, i.e., tilde{f}.
>
>
> **Dropout.** We agree that our current analysis does not deal with the dynamic change of architecture during training. Thanks to your point, we realized that the architecture we analyzed in the paper is not exactly the same as the dropout layer proposed in [Srivastava et al. 2014], although a lot of shared similarities. Specifically, we consider a fixed (or static) architecture which is generated by randomly dropping edges from a fully-connected network. We have revised our paper not calling it dropout network to avoid unnecessary confusion.

---

> > ### Author Response · Authors · 2022-08-02
> > **Response to Reviewer vWFG  (continued)**
> >
> > _Q1: What is the dependence of the bound on $R$?_
> >
> > Thanks for your suggestion and we agree that showing this dependence is important! We have explicitly written out the dependency on $R$ in our revision. Please see Theorem 3.6 and its proof in the revised version.
> >
> > The reason we did not take $R$ into the consideration in the bound is that the length of the optimization path of the wide neural networks is of the order $O(1)$ in general ([Liu et al. 2022, Du et al. 2018, 2019, Allen-Zhu 2019]). Therefore it is sufficient to assume $R$ to be a constant.
> >
> > The factor of R mainly comes from the bound on the scale of each individual neuron (Lemma I.2) and application of matrix Gaussian series bound (Lemma I.10). For a DAG with depth $L$, the dependency of the Hessian bound on $R$ is $(R+\log m)^{L^2}$. Therefore, the bound will hold as long as R is of the order $O(\log m)$.
> >
> > _Q2: Why do we take the maximum up to $L-1$_
> >
> > Thanks for pointing this out. This is a typo. The maximum is taken to $L$. We have corrected it in the revision.
> >
> > _Q3: It is confusing that output dimension is assumed to be one in L193._
> >
> > Thanks for pointing this out. The assumption on one output dimension is specific to the Taylor expansion Eq.(9). At L193 we show transition to linearity holds for each output of the network by the taylor expansion, and the results can be extended to multiple output networks. We have revised it to make it clear.
> >
> >
> > _Q4: How to understand the indices in the equations below L256_
> >
> > You are correct that $\ell'$ is anything between $\ell_2$ and $L$. As shown in Eq. (13), the second derivative ${\partial^2 f_k}/{\partial w^{(\ell_1)}\partial w^{(\ell_2)}}$ equals the sum of $G_k^{L,\ell'}$ over indices $\ell'$. For each $\ell'$, we only need to consider those $f_i^{(\ell')} \in \mathcal{F}_{\mathcal{S}_k^{(L)}}$
> >
> >  ( the subscript of the sum in the 2nd terms of RHS) where $\mathcal{F}_{\mathcal{S}_k^{(L)}}$ is the set of incoming neurons of $f_k^{(L)}$. Then $f_i^{(\ell')}$ serves as the incoming neurons of $f_k^{(L)}$ hence acting as ``$\ell' =L-1$''.
> >
> > References:
> >
> >  [LZB 2020]  Liu, Chaoyue, Libin Zhu, and Misha Belkin. "On the linearity of large non-linear models: when and why the tangent kernel is constant." Advances in Neural Information Processing Systems 33 (2020): 15954-15964.
> >
> > [Liu et al. 2022]  Liu, Chaoyue, Libin Zhu, and Misha Belkin.  "Loss landscapes and optimization in over-parameterized non-linear systems and neural networks." Applied and Computational Harmonic Analysis 59 (2022): 85-116.
> >
> > [Du et al. 2018] Du, Simon S., et al. "Gradient descent provably optimizes over-parameterized neural networks." arXiv preprint arXiv:1810.02054 (2018).
> >
> > [Du et al. 2019] Du, Simon, et al. "Gradient descent finds global minima of deep neural networks." International conference on machine learning. PMLR, 2019.
> >
> > [Allen-Zhu et al. 2018] Allen-Zhu, Zeyuan, Yuanzhi Li, and Zhao Song. "A convergence theory for deep learning via over-parameterization." International Conference on Machine Learning. PMLR, 2019.

---

> > > ### Comment · Reviewer_vWFG · 2022-08-08
> > > **Thanks for the response---a follow-up**
> > >
> > > Dear authors,
> > >
> > > Thanks for the detailed response, it sure helped me understand the paper better. Allow me to comment more on some of the points made by the authors, and also on the revised manuscript.
> > >
> > > 1. Re: W1. "lazy training" in [4]
> > >
> > > Thanks for the pointer to [LZB 2020]. I know that many formal theorems in [4] rely on the model at initialization being zero ($h(w_0) = 0$), but if you take a look at Theorem 2.3 they show that even in the case where $h(w_0) \neq 0$ the distance of the model to its linearization decreases with $\alpha$ when $\alpha$ is above some threshold. Also, their Appendix A.2 provides justification that for 2-layer networks with an $\frac{1}{\sqrt{\textup{width}}}$ scaling factor in front (i.e., the NTK parameterization), increasing the width leads to the lazy regime, which is in effect similar to increasing the scale $\alpha$. Indeed, their 2-layer net argument in Appendix A.2 uses the fact that $\mathcal A = \Theta(1)$ and $\mathcal B \ll 1$, as per the notation in [LZB 2020]. Having said that, I now understand that the argument in [4] does not readily extend beyond 2-layer fully-connected networks. Thanks for the clarification.
> > >
> > > 2. Re: W2. "Caveats"
> > >
> > > Thank you for revising the paper by adjusting your claims on CNNs and dropout layers. I also checked Appendix D and the formulation now reflects the usual ResNet architecture. It is interesting that the analysis indeed extends to models with parameter-free skip connections.
> > >
> > > 3. Re: Q1. Dependence on $R$.
> > >
> > > Thanks for clarifying the dependence on $R$ in the theorem statements. Unfortunately, it seems that $R$ has a worrisome exponent $L^2$ in the upper bound; even though $R$ can be thought as a constant in existing papers, leading factors of $2$ vs. $2^{L^2}$ are completely different! While the "transition to linearity" still holds true for sufficiently large $m$, this exponent seems to suggest that the minimum width $m$ required to achieve some benign algorithmic consequence scales exponentially in $L^2$, which is clearly quite unwelcome if we compare against convergences results with $m = \Omega (poly(L))$ (e.g., [2]). Do the authors have any comments on this? Is there any possibility that we can show algorithmic convergence results without an exponential dependence on depth?
> > >
> > >
> > > 4. Lastly, typo in line 298: $\mathcal N(I,0)$.

---

> > > > ### Author Response · Authors · 2022-08-09
> > > > **Response to Reviewer vWFG**
> > > >
> > > > Thanks for your response. We are glad we clarified your concerns.
> > > >
> > > > As for the exponent $L^2$ on the radius $R$ dependence, we agree that it is an important question and would like to address it as follows:
> > > >
> > > > In the current submission, we mainly focused on proving the transition to linearity for DAG architectures, and did not put much effort on optimizing the dependence on the depth $L$. We also don't think the exponential dependence $R^{L^2}$ is optimal and satisfactory. Note that we began to explicitly write out this $R$ dependence after receiving the reviews.  We are actively working on optimizing our analysis to obtain a better $R$ dependence, and will update it to the revision.
> > > >
> > > >
> > > > We would like to provide some highlights of our current attempts/ideas regarding the optimization of our analysis. In the current version of proofs, we simply take the worst case individually in each layer (i.e., the weight change of each layer takes the max of magnitude $R$ and takes the direction that always increases the Hessian norm). This brute force analysis makes the results become an aggregation of the worst case bound of each layer and finally becomes exponentially depending on $L$.   We believe this dependence can be relaxed by analyzing weight changes of different layers altogether, considering that the worst case does not happen to all the layers at the same time.
> > > >
> > > > On the other hand,
> > > > we also noticed that in many prior works, for example, in Theorem 5.1 in [Du et al. 2019] and Theorem 4 in [Liu et al. 2022], the depth dependence is also exponential. One exception is the work [Allen-Zhu et al. 2018], where the dependence is polynomial, but still far from satisfactory, because it has a large polynomial degree $L^{12}$. Also, note that [Allen-Zhu et al. 2018] only considers the gradient descent path, but we consider the ball which contains it.
> > > >
> > > > As to the reason why our result is $L^2$ instead of $L$, it comes from the DAG architecture. Since our result holds as long as the smallest in-degree is sufficiently large, we  put almost no constraint on the number of neurons in each layer. As a result, to get a bound on the norm of neurons in each layer with high probability, we have to introduce $\log(m)$ as well as an extra $R$ (Lemma I.2 and I.3). For standard architectures, e.g., FCNs, the number of neurons in each layer is sufficiently large hence no extra $\log m$ and $R$ will be introduced. We believe if we assume the number of neurons in each layer is sufficiently large, i.e. $\Omega(m)$,  the dependency of the depth can be reduced from $R^{L^2}$ to $R^{L}$.
> > > >
> > > > References:
> > > >
> > > > [Du et al. 2019] Du, Simon, et al. "Gradient descent finds global minima of deep neural networks." International conference on machine learning. PMLR, 2019.
> > > >
> > > > [Allen-Zhu et al. 2018] Allen-Zhu, Zeyuan, Yuanzhi Li, and Zhao Song. "A convergence theory for deep learning via over-parameterization." International Conference on Machine Learning. PMLR, 2019.
> > > >
> > > > [Liu et al. 2022] "Loss landscapes and optimization in over-parameterized non-linear systems and neural networks." Applied and Computational Harmonic Analysis 59 (2022): 85-116.

---

### Official Review · Reviewer_gBNf · 2022-07-23

**Rating:** 6
**Confidence:** 3
**Soundness:** 3 good
**Presentation:** 3 good
**Contribution:** 2 fair

**Summary:**

This paper shows that feedforward neural networks, defined as those whose neurons and their corresponding connections form a Directed Acyclic Graph (DAG) satisfy the transition to linearity property, as their width increases. The authors prove this property for this class of neural networks and show how it can be used to demonstrate exponential convergence of GD and SGD.

**Questions:**

I have the following questions and suggestions to make to the authors:

- In the paper, it is stated that ResNets do not fall under the examined architecture, but I am having difficulty understanding why (given that they only correspond to skipping some layers and do not add circles in the architecture). I would appreciate if the authors could clarify this.

- Given the great number of notation used in this work, I would suggest the authors gather the symbols used in a table in the appendix, so it can be referenced easily.


**Limitations:**

The authors adequately adress the limitations of their work (namely, the fact that they consider only a particular class of networks, and that the all of the neurons need to have similar in-degree), and I do not think further discussion on this is necessary.

**Strengths And Weaknesses:**

This is a purely theoretical work, demonstrating some interesting properties for a very general class of neural networks. Its main strengths lie in the following:

- The authors clearly present the main theoretical contributions of the work, as well as their corresponding limitations. Reading the paper, it is very clear what the authors aim to prove (namely, that the class of neural networks they examine becomes arbitrarily close to linear functions, as their width increases).

- The authors also prove how the above property can be used to show that GD and SGD converge fast for this class of networks, which is a useful guarantee from an optimization perspective.

- The paper is also theoretically sound. The proof sketches in the main paper are clear, and I did not find any major issues with the full proofs included in the appendix. As such, the main claims of the paper are properly supported.

However, I can identify the following weaknesses in the paper, which I believe need to be addressed by the authors.

- First, I am somewhat unsure about the novelty of the setting examined by the paper. The properties proposed are similar to those obtained in a previous paper cited by the authors [A]. The major difference lies in that this work considers a general DAG architecture, while the previous one examines a more constrained fully connected architecture (which constrains the edges between the neurons to be strictly from a previous layer to a later one, while in this work the more general DAG structure allows for back-edges, as long as no cycle is formed). This is because the neurons are arranged in layers based on their distance from the input, so there can be a longer path to a neuron in an early layer which also goes through a later layer. However, it seems to me that the DAG architecture can be converted to that of the previous work by splitting the edges of the network, increasing the distance of neurons from the input, and given that there are no cycles in the graph this process must terminate with a more traditional feedforward network (where all edges go from early layers to later ones). Hence, it seems to me that graphs with a general DAG structure can be converted to an equivalent network corresponding to a regular feedforward network. I would appreciate it if the authors could comment on whether the above is a correct understanding of the architecture they examine, as well as further elaborate on the differences of their work from [A].

- Secondly, the results of the authors consider R (the radius of the ball within which the linear approximation holds) to be a constant, and thus omit it from their results. I believe that this should be included in the result – the quality of the linear approximation with respect to the distance from the origin of the ball is important, in my opinion, to fully understand differences from simple linear approximation via Taylor expansion.

- Finally, while the theoretical part of the paper is strong, I feel that in its current state the authors do not adequately motivate its ramifications. There is a direct application to GD and SGD, but this is made clear only in Section 4. I believe that furthering the discussion on the application of these results would help the paper. In a similar vein (while obviously not necessary, given the theoretical nature of the paper), I believe that a simple visual example of the transition to linearity would help with the presentation of the paper.

Overall, I believe this is a good theoretical work, and I lean towards acceptance, especially if the authors further clarify the relationship of their work with [A].

Reference:
[A]: Liu, C., Zhu, L., & Belkin, M. (2022). Loss landscapes and optimization in over-parameterized non-linear systems and neural networks. Applied and Computational Harmonic Analysis, 59, 85-116.

---

> ### Author Response · Authors · 2022-08-02
> **Response to Reviewer gBNf**
>
> We thank the reviewer for the positive feedback! We address your concerns and questions one by one below:
>
> _W1: Can the DAG structure be converted to FCNs?_
>
>  If we understand correctly, you mean  to replace some of the edges $w$ by a chain of edges $w_1w_2\cdots w_k$ such that each path from the input to a given neuron has an equal length. In this case, a general DAG structure still cannot be converted to that of the previous works. Note that by replacing one edge by a chain of edges, we are introducing new neurons which are the joints of the new edges. (e.g., there are $k-1$ new neurons if we let $w \Rightarrow w_1w_2\cdots w_k$.) These new neurons can *only* be connected with one neuron from the next layer and one neuron from the previous layer. This special restriction on the edge connections does not present in the architectures in previous works and has never been analyzed. Please correct us if we do not understand you correctly. We would appreciate if the reviewer can provide a more detailed explanation on how the converting works, then we can discuss further.
>
>
> _W1: Elaborate the difference of our work from [A]_
>
> We made two major progress, compared to the work [A]. First, we extend the "transition to linearity" from a special architecture, i.e., fully-connected neural networks, to a more general structure i.e., DAG.
> This tells us that "transition to linearity" is not a property specific to a special kind of architecture, but a general property for a much more broad class of structures; hence, it indicates that ``transition to linearity'' is a fundamental phenomenon which has not been fully understood. Secondly, the analytical tools in [A] do not apply to the general DAG architecture setting, since we consider arbitrary in-degrees.
>
>
>
>
> _W2: the radius $R$ should be included in the results._
>
> Thanks for your suggestion and we agree that showing this dependence is important! We have explicitly written out the dependency on $R$ in our revision. Please see Theorem 3.6 and its proof in the revised version.
>
> The reason we did not take $R$ into the consideration in the bound is that the length of the optimization path of the wide neural networks is of the order $O(1)$ in general ([A, Du et al. 2018, 2019, Allen-Zhu 2019]). Therefore it is sufficient to assume $R$ to be a constant.
>
> The factor of R mainly comes from the bound on the scale of each individual neuron (Lemma I.2) and application of matrix Gaussian series bound (Lemma I.10). For a DAG with depth $L$, the dependency of the Hessian bound on $R$ is $(R+\log m)^{L^2}$. Therefore, the bound will hold as long as R is of the order $O(\log m)$.
>
>
> _W3: ramifications of our results; visual example_
>
> Thanks for bringing up interesting ramifications! While our main focus is about the property of transition to linearity of DAGs, we totally agree that a discussion about optimization is useful. As we have shown in Section 4, transition to linearity is a very useful tool which provides a path toward showing the convergence of GD/SGD. We believe We believe a similar result of SGD to Theorem 7 in [A] can be worked out for DAG networks. We will leave it as the future work.
>
> For the experiment, we have added an experiment in Appendix J in revision. More experiments will be added later. Also please see the top of the page as a highlight of our experiment.

---

> > ### Author Response · Authors · 2022-08-02
> > **Response to Reviewer gBNf (continued)**
> >
> > _Q1: Why do ResNets not fall under DAG architecture?_
> >
> > We say that ResNets do not fall in DAG networks because there is no trainable weight  on the skip connection.  By our definition (Definition 2.2), each edge is assigned to a trainable weight. Therefore ResNets do not strictly satisfy our definition, though the DAG structure includes the ResNets structure when not considering the trainability of the edges.
> >
> > We want to point out that, even without satisfying the DAG network definition, our analysis can be easily extended to handle ResNets.
> > Consider for example,  for a DAG network, adding a skip connection from $(\ell-2)$-th layer to $(\ell-1)$-th layer neurons $f_{res}^{(\ell-1)}$, we get $g_{res}^{(\ell)} =<W^{(l)}, f_{res}^{(\ell-1)} + f_{res}^{(\ell-2)}>$ where $g$ denotes the preativation corresponding to $f$ i.e., tilde{f}.
> >
> > Compared to our standard DAG definition which is $g^{(\ell)} =\langle W^{(\ell)}, f^{(\ell-1)}\rangle$, there is an extra term $f_{res}^{(\ell-2)}$. When considering the Hessian of $g_{res}^{(\ell)}$, there will be an extra term related to $f_{res}^{(\ell-2)}$ as an addition, but has already been taken care of in the recursive analysis, since it is from $(\ell-2)$-th layer. Therefore, the scale of Hessian will not be affected. We have a detailed explanation in Appendix D.
> >
> >
> >
> > _Q2: a table of notations_
> >
> > Thanks for the suggestion! We have added the table in our appendix. Please see the Table 1 in revision.
> >
> > References:
> >
> > [A] "Loss landscapes and optimization in over-parameterized non-linear systems and neural networks." Applied and Computational Harmonic Analysis 59 (2022): 85-116.
> >
> > [Du et al. 2018] Du, Simon S., et al. "Gradient descent provably optimizes over-parameterized neural networks." arXiv preprint arXiv:1810.02054 (2018).
> >
> > [Du et al. 2019] Du, Simon, et al. "Gradient descent finds global minima of deep neural networks." International conference on machine learning. PMLR, 2019.
> >
> > [Allen-Zhu et al. 2018] Allen-Zhu, Zeyuan, Yuanzhi Li, and Zhao Song. "A convergence theory for deep learning via over-parameterization." International Conference on Machine Learning. PMLR, 2019.

---

> > > ### Comment · Reviewer_gBNf · 2022-08-08
> > > **Thank you for your comments.**
> > >
> > > I'd like to thank the authors for addressing my comments/suggestions.
> > >
> > > Regarding my first point, the construction you are describing is exactly the one I meant - this will indeed result in the new architecture having a single path inside the graph. I understand now that this is going to pose an issue with the analysis provided by [A], since this is an extra constraint imposed on the network, and I thank the authors for clarifying this point.
> > >
> > > I'll keep my score for now, pending discussion on the applicability of the DAG formulation on CNNs/other architectures, raised by another reviewer.

---

### Official Review · Reviewer_wXUj · 2022-07-23

**Rating:** 5
**Confidence:** 5
**Soundness:** 3 good
**Presentation:** 3 good
**Contribution:** 2 fair

**Summary:**

This paper extend the results on transition of neural networks to linear regime as width grows to Directed Acyclic Graph (DAG) architectures that include almost all of the feedforward networks and networks with dropout. Then based on properties of the linear regime, the authors prove such wide neural networks satisfy the Polyak-Lojasiewicz conditions, that guarantees the convergence of GD/SGD to global minima.

**Questions:**

Is your results in section 4 ( Relation to optimization) only for MSE loss? can it be extended to other looses?

**Ethics Review Area:**

["I don’t know"]

**Limitations:**

Unfortunately, the authors do no provide any experiments to support their theorems. I believe this should be added to the paper.

**Strengths And Weaknesses:**

The paper extend the current results on linearity of wide MLPs to almost all feedforward neural networks. The idea is not original, but the techniques and theorems that are provided in this paper can be interesting to the Neurips community. Also the paper is very well-written.

---

> ### Author Response · Authors · 2022-08-02
> **Response to Reviewer wXUj**
>
> We thank the reviewer for the comments.
>
> _Q: the authors do no provide any experiments to support their theorems. I believe this should be added to the paper._
>
> After receiving the reviews, we conducted a simple experiment to verify our results and added it to the revision (see Appendix J). We also have a discussion/highlight of the experimental results above in this page (see the comments with title "Experimental Results").
>
> _Q: Is the results in Section 4 only for MSE loss? Can it be extended to other losses?_
>
>
> Yes, we assumed MSE loss in this paper. We agree that extending to other losses, e.g., cross-entropy loss, is a very interesting problem. We believe it is possible to extend our results to other losses. For example, in [Frei \& Gu, 2021], they show wide neural networks with cross entropy loss satisfies proxy PL inequality (Eq. (3.3)), which provides certain convergence results of optimization. We believe we can build an optimization theory for DAG neural networks with other losses, using such variants of PL condition.
>
>
> Reference:
>
> [Frei \& Gu, 2021] Frei, Spencer, and Quanquan Gu. "Proxy convexity: A unified framework for the analysis of neural networks trained by gradient descent." Advances in Neural Information Processing Systems 34 (2021): 7937-7949.

---

### Author Response · Authors · 2022-08-02
**Experimental results**

We thank all the reviewers for the helpful feedback!

After receiving the suggestion of adding a visual example/experiment from several reviewers, we conducted a simple experiment and added it to section J in the Appendix. The following is a highlight of the experimental setting and results:

We trained a  DAG network built from a 3-layer DenseNet with each weight removed i.i.d. with prob. $0.5$,  on $10$ data points of CIFAR-2 using GD.

We observe the convergence of loss for all widths $\{2,2^2,...,2^{12}\}$, and the following table shows the scaling of the tangent kernel change with width follows close to the theoretical prediction of $\Theta(1/\sqrt{m})$.

| Width  | $2^1$ |$2^2$ |$2^3$ |$2^4$ |$2^5$ |$2^6$ |$2^7$ |$2^8$ |$2^9$ |$2^{10}$ |$2^{11}$ |$2^{12}$ |
| --- | -----|------|------|------|------|------|------|------|------|------|------|------|
| **Maximum change of NTK during training** |0.2268       |0.1788        |0.1531       |0.1266        |0.0933       |0.0792        |0.0605       |0.0495        |0.0333       |0.0209        |0.0129       |0.0107        |

---

### Meta-Review · Area_Chair_CAWM · 2022-08-26

**Recommendation:** Accept
**Confidence:** Less certain

**Metareview:**

This is a challenging manuscript to make a final recommendation on accept or reject as there is a clear consensus amongst the reviewers that the manuscript is borderline between accept and reject.  The main concern is the incremental nature of the results, extending the prior results of Reference: [A]: Liu, C., Zhu, L., & Belkin, M. (2022). Loss landscapes and optimization in over-parameterized non-linear systems and neural networks. Applied and Computational Harmonic Analysis, 59, 85-116. from fully connected networks to the more general setting of DAGs.  The authors and reviewers point out that the DAG setting requires substantial adaptation of the technique used to prove the results in [A] which is the reason I have selected Accept over Reject.  That said, the architecture the reviewers are most interested in is CNNs which were pointed out don't fall within the definition of the DAG and have been removed from the manuscript.  Inclusion of shared weights, CNNs, would be a great addition to the manuscript which would make the decision for acceptance clearer and would also make the manuscript more compelling for readers who otherwise are unsure the benefit of the DAG setting.

**Award:**

No

---

### Decision · Program_Chairs · 2022-09-14

Accept